# FRACTAL-INSPIRED MESSAGE PASSING NEURAL NETWORKS WITH FRACTAL NODES

## ABSTRACT

Graph Neural Networks (GNNs) have emerged as powerful tools for learning on graph-structured data, but they struggle to balance local and global information processing. While graph Transformers aim to address these issues, they often neglect the inherent locality of Message Passing Neural Networks (MPNNs). Inspired by the fractal nature of real-world networks, we propose a novel concept, '*fractal nodes*', that addresses the limitations of both MPNN and graph Transformer. The approach draws insights from renormalization techniques to design a message-passing scheme that captures both local and global structural information. Our method enforces feature self-similarity into nodes by creating fractal nodes that coexist with the original nodes. Fractal nodes adaptively summarize subgraph information and are integrated into MPNN. We show that fractal nodes alleviate an over-squashing problem by providing direct shortcuts to pass fractal information over long distances. Experiments show that our method achieves comparable or better performance to the graph Transformers while maintaining the computational efficiency of MPNN by improving the long-range dependencies of MPNN.

## 1 INTRODUCTION

GNNs have emerged as powerful tools for learning on graph-structured data, in various domains such as social network analysis, molecular property prediction, and recommendation systems (Defferrard et al., 2016; Veličković et al., 2018; Chen et al., 2020a; Chamberlain et al., 2021). At the core of this field lies the MPNN (Gilmer et al., 2017), which iteratively propagates information between neighboring nodes. Recent research has focused on addressing the limitations of MPNN, such as over-smoothing (Nt & Maehara, 2019) and over-squashing (Alon & Yahav, 2021). To overcome these challenges, Transformer architectures (Vaswani et al., 2017) have been introduced to the graph learning community, applying self-attention mechanisms to enable long-range interactions by treating all nodes as tokens (Dwivedi & Bresson, 2021; Wu et al., 2021; Kreuzer et al., 2021b). While graph Transformers have shown promise in capturing global information, they often neglect the inherent locality of MPNNs (Xing et al., 2024). Although approaches such as GraphGPS (Rampášek et al., 2022) attempt to combine MPNN and Transformer node representations to balance local and global information, the computational complexity of Transformers remains a challenge.

**Motivation.** The limitations of both MPNN and graph Transformers motivate us to seek a novel approach that balances local and global information processing while maintaining computational efficiency. Our inspiration comes from the fractal nature (Mandelbrot, 1983) of real-world networks (Dill et al., 2002; Kim & Kahng, 2010; Chen et al., 2020b). This fractality exhibits self-similarity over different scales, meaning that parts of the network resemble the whole. We approach this self-similarity from two perspectives – the structural aspect, where structural patterns repeat across scales, and the feature aspect, where we aim to enable consistent feature patterns on different network scales. In fractal network analysis, a popular technique is renormalization (see Fig. 1(a)), which involves replacing groups of nodes with "super-nodes" to study how network properties change in scales (Song et al., 2005). The fractality properties and the concept of renormalization motivate us to ask: "*Can we design a message passing scheme inspired by fractal geometry and renormalization that effectively captures both local and global structural information in graphs?*" Our answer is "yes," and we introduce our main idea.

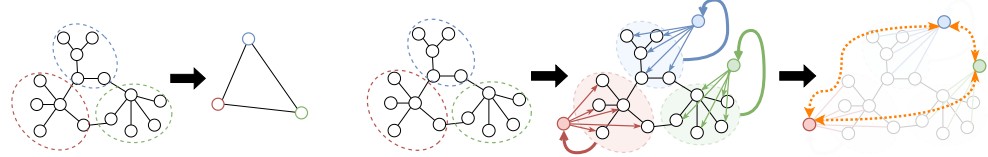

(a) Coarsened (renormalized) graph        (b) Graph where our fractal nodes are connected.

Figure 1: Heuristic comparison of renormalization and our fractal node process. (a) In renormalization, the original graph is replaced by a single node according to each box-covering method, resulting in a coarsened network. (b) After partitioning the original graph into subgraphs, we aggregate the low and high-frequency information of each subgraph to create *fractal nodes* (🔴, 🔵, 🟢). Then, we propagate the information to the original nodes (see our proposed FN). We also support the long-distance interactions (orange dashed lines) between fractal nodes (see our proposed FN$_M$).

**Main idea: *fractal nodes* for enforcing self-similarity.**  We propose a novel concept called a '*fractal node*', inspired by fractal nature and the renormalization process. Drawing on our perspective of self-similarity, this approach aims to reflect the characteristics of larger structures in individual network nodes while enforcing feature self-similarity, thus promoting efficient information flow. Unlike renormalization, which replaces node groups with single nodes, our method partitions the given graph into multiple subgraphs and creates fractal nodes for each subgraph that coexist with the original nodes (see Fig. 1(b)). These fractal nodes represent the information of each subgraph while maintaining connections to the original structure.

By incorporating subgraph features into each node within the given subgraph through direct connections with fractal nodes, our approach enables smaller units (nodes) to reflect the properties of larger units (subgraphs), effectively enforcing feature self-similarity into the nodes. Specifically, we achieve this by adaptively combining low-frequency (global) and high-frequency (local) components of node features within each subgraph, where the low-frequency component captures common subgraph features while a learnable parameter controls the contribution of high-frequency variations. This process of combining node-specific features with subgraph characteristics enables the seamless integration of fractal nodes into existing MPNNs. In addition, we ensure that the hidden vectors of fractal nodes and original nodes are in the same latent space using the same MPNN layer. This approach allows for the simultaneous consideration of local and global information while maintaining computational efficiency. Each fractal node adaptively summarizes the information of nodes within its corresponding subgraph, going beyond mean pooling to capture subgraph-level characteristics.

Assigning fractal node to each subgraph contributes to mitigate over-squashing problem. Each fractal node has direct first-order connections to every node within the corresponding subgraph, while preserving the rich node features aggregated across all nodes. This direct connection between the fractal node and the orignal nodes serves as a shortcut pathway to facilitate the propagation of the information across multi-hop distances, which has been considered as the primary cause of over-squashing (Alon & Yahav, 2021). Additionally, we apply an MLP-Mixer (Tolstikhin et al., 2021) at the last layer to flexibly mix the representations of fractal nodes. This enables inter-subgraph long range interactions to globally exchange the subgraph context without passing through multiple edges with potential risk of singal degradation as depth grows.

**Contributions.**  We introduce a novel paradigm, *fractal nodes*[1], for better propagation by *enforcing self-similarity* at the subgraph level into individual nodes. Our main contributions are as follows:

- We propose *fractal nodes*, which can be integrated into MPNNs, inspired by the fractal nature of networks (Section 3) and discuss the properties of our fractal nodes (Section 4).

- We theoretically and empirically show that fractal nodes alleviate the over-squashing problem (Section 5.1) and improve the expressive power over MPNN (Section 5.2).

- Our experiments on various benchmark datasets demonstrate that MPNNs augmented with fractal nodes achieve performance comparable to or better than state-of-the-art graph Transformer-based models (Section 5.3), while maintaining computational efficiency (Section 5.4).

---

[1]Our source code is available here: https://sites.google.com/view/fractalnode/

## 2 BACKGROUND & RELATED WORK

In this section, we discuss MPNNs, their limitations, graph Transformers, augmented MPNNs and discuss fractality and self-similarity in networks.

**Message passing neural network.** Given a graph $\mathcal{G} = (\mathcal{V}, \mathcal{E})$, we use $\mathcal{V}$ and $\mathcal{E}$ to denote its nodes and edges, respectively. The nodes are indexed by $v$ and $u$ such that $v, u \in \mathcal{V}$, and an edge connecting nodes $v$ and $u$ is denoted by $(v, u) \in \mathcal{E}$. We consider the case where each node $v$ has a hidden vector $h_v^{(\ell)} \in \mathbb{R}^d$, where $d$ is the size of the hidden dimension, and $\ell$ is the number of layers. MPNNs iteratively update node representations using the following equation:

$$h_v^{(\ell+1)} = \varphi(h_v^{(\ell)}, \psi^{(\ell)}(\{h_u^{(\ell)} : u \in \mathcal{N}(v)\})), \tag{1}$$

where $\psi^{(\ell)}$ and $\varphi^{(\ell)}$ are aggregation function and update function. Their different definitions result in different architectures (Kipf & Welling, 2017; Xu et al., 2019a; Bresson & Laurent, 2017).

**Limitations of MPNNs.** In several studies, MPNN has been investigated for its expressive power limitations and over-squashing problems. Simple MPNN is only as powerful as the 1-Weisfeler-Leman graph isomorphism test (Xu et al., 2019b). The over-squashing problem occurs when MPNNs struggle to propagate information along long paths, resulting in substantial loss of information when aggregating from too many neighbors into a fixed-sized node feature vector (Alon & Yahav, 2021; Di Giovanni et al., 2023). In such scenarios, local information spreading along the natural graph circuits is insufficient to fully capture the local and global context of the graph. This leads to the emergence of graph Transformers that use self-attention, thereby solving the over-squashing problem of self-attention with its "everything is connected to everything".

**Graph Transformers.** Because of successes of Transformers in natural language processing (Vaswani et al., 2017), and computer vision (Zhou et al., 2021; Touvron et al., 2021), many previous works have attempted to bring Transformer architecture to the graph domain (Dwivedi & Bresson, 2021; Müller et al., 2023). Dwivedi & Bresson (2021) proposed the use of graph Laplacian eigenvectors as node positional encodings. Subsequent research has explored various strategies to enhance graph Transformer performance. Rampášek et al. (2022) proposed a general framework, GraphGPS, that combine MPNN and graph Transformer including self-attentions and positional or structure encoding. Ying et al. (2021) proposed Graphormer that uses attention mechanisms to estimate several types of encoding, such as centrality, spatial, and edge endodings. Wu et al. (2021) applies the MPNN directly to all nodes and then applies a Transformer, which is computationally intensive. He et al. (2023) generalize ViT Dosovitskiy et al. (2021) to graphs and Ma et al. (2023) show that adding inductive biases to graph Transformers removes the need for MPNN modules in GraphGPS. Exphormer improves GraphGPS by using self-attention on expander graphs (Shirzad et al., 2023).

One common belief of the advantage of the graph Transformer over MPNN is its capacity in capturing long-range interactions while alleviating over-squashing in MPNN (Alon & Yahav, 2021; Di Giovanni et al., 2023). While graph Transformers have shown promise in addressing the limitations of MPNNs, they often come at the cost of increased computational complexity, typically scaling from $\mathcal{O}(|\mathcal{E}|)$ to $\mathcal{O}(|\mathcal{N}|^2)$, where $|\mathcal{E}|$ is the number of edges and $|\mathcal{N}|$ is the number of nodes. This computational burden calls for more efficient architectures that can capture global information without the full quadratic cost of attention mechanisms.

**Augmented MPNNs.** To improve information flow and address the limitations of standard MPNNs, various strategies have been proposed (Di Giovanni et al., 2023; Shi et al., 2023; Choi et al., 2024). One approach involves incorporating additional global graph features during the representation learning process (Gilmer et al., 2017; Hu et al., 2020). Another effective method is rewiring the input graph to enhance connectivity and alleviate structural bottlenecks (Gasteiger et al., 2019; Black et al., 2023; Karhadkar et al., 2023; Nguyen et al., 2023). These adjustments allow for more effective information flow within the network. Another example of graph augmentation is the virtual node, which adds a new node to the graph to enhance information exchange between all pairs of nodes. This heuristic, introduced by Gilmer et al. (2017), has been observed to improve performance on various tasks. Further analysis by Hwang et al. (2022) and Cai et al. (2023) has explored the role of virtual nodes in mitigating under-reaching and over-smoothing issues.

**Subgraphs in graph learning.** Several works introduce hierarchical clustering and coarsening for learning on graphs (Dong et al., 2023). Chiang et al. (2019) use graph clustering to identify well-connected subgraphs on large graphs. HC-GNN (Zhong et al., 2023) shows competitive performance in node classification on large-scale graphs, using hierarchical community structures for message passing. In graph Transformers, several hierarchical models (Zhao et al., 2022; Gao et al., 2022; Zhu et al., 2023; He et al., 2023) attempt to manage computational complexity, though they still face challenges with scalability as all nodes remain within the computational burden of the Transformer architecture. However, our approach, the incorporation of fractal nodes to MPNN, can reduce this computational cost while preserving structural information.

**Fractality and self-similarity in networks.** The concept of fractals, introduced by Mandelbrot (1983), transformed our understanding of complex, irregular structures in nature by revealing self-similarity across different scales. This insight has since been applied to various fields, including network science, where many real-world networks have been shown to exhibit fractal structures and scale-free properties (Song et al., 2005; Kim et al., 2007; Fronczak et al., 2024). For instance, social networks, the World Wide Web, and even protein interaction networks have been found to have fractal properties (Chen et al., 2020b).

In our work, we define fractality as the degree to which subgraph properties resemble those of the entire graph when consistently partitioned. While traditional fractal analysis (e.g., renormalization techniques) commonly uses box-covering algorithms (Kim et al., 2007), we bring this concept to the constraints of benchmark datasets where absolute node positions are unknown. Instead, we construct subgraphs through graph partitioning.

## 3 FRACTAL-INSPIRED MESSAGE PASSING WITH FRACTAL NODES

In this section, we propose our fractal nodes and explain how they contribute to overcome limitations of existing MPNNs. We describe how to enfore self-similarity to a graph by assigning fractal nodes and how to implement intra and inter-subgraph local and global interactions guided by fractal nodes.

**Notaion.** Let $\{\mathcal{V}_1, \ldots, \mathcal{V}_C\}$ be the set of node subsets corresponding to $C$ subgraphs, where $C$ is the number of subgraphs. $\mathcal{G}_c = (\mathcal{V}_c, \mathcal{E}_c)$ is the induced subgraph of $\mathcal{G}$. We define $h_{v,c}^{(\ell)}$ as the hidden vector of node $v$ of the $c$-th subgraph in layer $\ell$, and $f_c^{(\ell)}$ as the hidden vector of the fractal node of the $c$-th subgraph in the $\ell$-th layer.

**Message passing with *fractal nodes*.** We first introduce the message passing process, including fractal nodes. The message passing process for both the node-level and fractal node-level representations proceeds as follows:

$$\widetilde{h}_{v,c}^{(\ell+1)} = \varphi^{(\ell)}(h_{v,c}^{(\ell)}, \psi^{(\ell)}(h_{u,c}^{(\ell)} : u \in \mathcal{N}_v)), \tag{2}$$

$$f_c^{(\ell+1)} = \varphi_{\mathsf{FN}}^{(\ell)}(f_c^{(\ell)}, \psi_{\mathsf{FN}}^{(\ell)}(\widetilde{h}_{u,c}^{(\ell+1)} : u \in \mathcal{N}_v)), \tag{3}$$

$$h_{v,c}^{(\ell+1)} = \widetilde{\varphi}^{(\ell)}(\widetilde{h}_{v,c}^{(\ell+1)}, f_c^{(\ell+1)}), \tag{4}$$

where $\mathcal{N}(v)$ is the set of neighbors of node $v$. Equation (2) performs standard message passing at the node level. If the graph is not partitioned into subgraphs, Equation (2) alone is equivalent to standard MPNN. Equation (3) updates the fractal node representations. It aggregates hidden vectors from all nodes in the subgraph, $\mathcal{V}_c$, using the $\widetilde{h}_{u,c}^{(\ell+1)}$, and then updates the fractal node representation. $\psi_{\mathsf{FN}}^{(\ell)}$ and $\varphi_{\mathsf{FN}}^{(\ell)}$ are aggregate and update functions for fractal nodes, which will be explained in more detail. The update function $\widetilde{\varphi}^{(\ell)}$ is the step that shows that the message $f_c^{(\ell+1)}$ is propagated to $h_{v,c}^{(\ell)}$.

**How to create fractal nodes.** As shown in Fig. 1(b), fractal nodes are created from partitioned subgraphs. To partition into subgraphs, we consider the METIS (Karypis & Kumar, 1998) algorithm for its computational efficiency. How we use METIS is discussed in more detail in Appendix B.1. Following our dual perspective of self-similarity, each fractal node serves two purposes: (1) representing structural patterns of a subgraph that potentially mirror the whole graph's topology and (2) enabling feature self-similarity by integrating low and high-frequency components from the node

features within the subgraph. While graph partitioning preserves structural patterns, we focus on achieving feature self-similarity by adaptively combining low-pass filtering (LPF) and high-pass filtering (HPF). We first show that mean pooling captures only the direct current (DC) component (i.e., the lowest frequency component) of the signal.

**Theorem 3.1** (Mean pooling as a low-pass filter capturing the DC component). *Let $h_v$ represent the hidden state of node $v$ in subgraph $\mathcal{V}_c$ and let $H_c = [h_1, h_2, \ldots, h_n] \in \mathbb{R}^{n \times d}$ be the matrix of node features for all nodes in $\mathcal{V}_c$ where $n = |\mathcal{V}_c|$ is the number of nodes in the subgraph. The mean pooling operation applied to the node features is equivalent to extracting the DC or the lowest frequency component of the signal in the frequency domain.*

As shown in Theorem 3.1, mean pooling corresponds to extracting the lowest frequency component — also known as the DC component — in the Fourier domain. This DC component capture the global characteristic of the subgraph, but it ignores higher-frequency variations that represent local details. A formal proof of Theorem 3.1 is provided in Appendix A.

While Theorem 3.1 shows that mean pooling only captures the DC component, fractal nodes go beyond this limitation by using LPF and HPF. We adaptively rescale the high-frequency component, and combine LPF and HPF together to form fractal nodes:

$$f_c^{(\ell+1)} = \mathsf{LPF}(h_{v,c}^{(\ell+1)}) + \omega_c^{(\ell)}\mathsf{HPF}(h_{v,c}^{(\ell+1)}), \tag{5}$$

where $\omega_c^{(\ell)}$ is a learnable parameter controlling the contribution of high-frequency components. We use a learnable scalar parameter, $\omega_c^{(\ell)} \in \mathbb{R}^1$, or a learnable vector parameter, $\boldsymbol{\omega}_c^{(\ell)} \in \mathbb{R}^d$. The LPF is computed by averaging the node features within the subgraph, so it can capture global information:

$$\mathsf{LPF}(h_{v,c}^{(\ell+1)}) = \frac{1}{|\mathcal{V}_c|} \sum_{v \in \mathcal{V}_c} h_{v,c}^{(\ell+1)}. \tag{6}$$

Equation (6) is analogous to mean pooling and represents the global, low-frequency component of the subgraph. To capture the finer details, the HPF is applied by subtracting the low-pass filtered output from the original node hidden vector. This allows the model to retain the local variations that would otherwise be lost:

$$\mathsf{HPF}(h_{v,c}^{(\ell+1)}) = h_{v,c}^{(\ell+1)} - \mathsf{LPF}(h_{v,c}^{(\ell+1)}). \tag{7}$$

**Fractal Nodes mixing with MLP-Mixer.** We can also allow fractal nodes to exchange messages, as the coarsened network in Fig. 1(a) takes advantage of long-distance interactions. To do this, we can apply the MLP-Mixer layer (Tolstikhin et al., 2021) to the fractal nodes in the last layer. This means that we do not need to create a coarsened network, and the MLP-Mixer flexibly mix the representations of fractal nodes:

$$\widetilde{F} = \mathsf{MLPMixer}(F^{(L)}), \ F^{(L)} = [f_1^{(L)}, f_2^{(L)}, ..., f_C^{(L)}], \tag{8}$$

where $F^{(L)}$ is the matrix of all fractal node representations at final layer $L$. The MLP-Mixer layer consists of token-mixing and channel-mixing steps:

$$U = F^{(L)} + (W_2\rho(W_1\mathsf{LayerNorm}(F^{(L)}))) \in \mathbb{R}^{C \times d} \tag{9}$$

$$\widetilde{F}^{(L)} = U + (W_4\rho(W_3\mathsf{LayerNorm}(U^T)^T) \in \mathbb{R}^{C \times d}, \tag{10}$$

where $\rho$ is a GELU nonlinearity, $\mathsf{LayerNorm}(\cdot)$ is layer normalization, and matrices $W_1 \in \mathbb{R}^{d_1 \times C}, W_2 \in \mathbb{R}^{C \times d_1}, W_3 \in \mathbb{R}^{d_2 \times d}, W_4 \in \mathbb{R}^{d \times d_2}$ are learnable weight matrices, where $d_1$ and $d_2$ are the tunable hidden widths in the token-mixing and channel-mixing MLPs.

**Instance of our framework.** To better understand our framework, we show how to integrate fractal nodes into MPNNs: GCN (Kipf & Welling, 2017), GINE (Xu et al., 2019a), and GatedGCN (Bresson & Laurent, 2017). We will use these MPNNs for our experiments. The update equation for GCN + FN is the following:

$$\widetilde{h}_{v,c}^{(\ell+1)} = \sigma\Big(h_{v,c}^{(\ell)} + \sum_{u \in N(v)} \frac{1}{\sqrt{\deg_v \deg_u}} h_{u,c}^{(\ell)} W^{(\ell)}\Big),$$

$$f_c^{(\ell+1)} = \mathsf{LPF}(\widetilde{h}_{v,c}^{(\ell+1)}) + \omega_c^{(\ell)} \cdot \mathsf{HPF}(\widetilde{h}_{v,c}^{(\ell+1)}), \tag{11}$$

$$h_{v,c}^{(\ell+1)} = \widetilde{h}_{v,c}^{(\ell+1)} + f_c^{(\ell+1)},$$

where $\sigma$ a ReLU activation function, and $\deg_v$ and $\deg_u$ are their node degrees. Due to space constraints, the update equations of GINE and GatedGCN can be found in Appendix B.2 and we provide implementation details in Appendix B.

The method of applying the fractal nodes as in Equation (11) is called FN, and the method of using the fractal nodes of the last layer by mixing (see Equation (8)) is called $\mathsf{FN}_M$ from now on.

**The output layer.** Once the final representation $h_G$ is derived, we use a multi-layer perceptron (MLP) as an output layer to predict graph-level outputs:

$$y_G = \mathsf{MLP}(h_G), \quad h_G = \mathsf{MeanPool}(H^{(L)} \text{ for FN}, \ \widetilde{F}^{(L)} \text{ for } \mathsf{FN}_M) \in \mathbb{R}^d,$$

where $y_G$ is either a scalar for regression tasks or a vector for classification tasks, and $H^{(L)} = [h_1^{(L)}, ..., h_{|V|}^{(L)}]$ is the matrix of node representations at the final layer $L$ for all nodes in the graph.

## 4 PROPERTIES OF FRACTAL NODES

In this section, we analyze why fractal nodes are effective and what properties they have, discuss the model complexity, and compare them with previous work.

### 4.1 WHY FRACTAL NODES WORK?

**Theoretical analysis.** We provide theoretical analysis showing that fractal nodes help mitigate oversquashing by reducing the effective resistance between nodes.

**Theorem 4.1** (Resistance reduction). *Let $\mathcal{G}$ be the original graph and $\mathcal{G}_f$ be the augmented graph with fractal nodes. For any nodes $u, v \in \mathcal{G}$, the effective resistance in $\mathcal{G}_f$ satisfies:*

$$R_f(u, v) \leq R(u, v), \tag{12}$$

*where $R_f(u, v)$ is the effective resistance in $\mathcal{G}_f$ and $R(u, v)$ is the original effective resistance in $\mathcal{G}$.*

This reduction in effective resistance directly improves signal propagation between distant nodes:

**Theorem 4.2** (Signal propagation with fractal nodes). *For a MPNN with fractal nodes, the signal propagation between nodes $u, v$ after $\ell$ layers satisfies:*

$$\|h_u^{(\ell)} - h_v^{(\ell)}\| \leq \exp(-\ell/R_f(u, v))\|h_u^{(0)} - h_v^{(0)}\|, \tag{13}$$

*where $R_f(u, v)$ is the effective resistance in the augmented graph with fractal nodes.*

Since $R_f(u, v) \leq R(u, v)$, fractal nodes improve the worst-case signal propagation bound compared to the original graph. *The proofs and detailed analysis can be found in* Appendices L.2 and L.3.

**Frequency response analysis.** We analyze the frequency response of node representations to understand the information encoding properties of fractal nodes. Fig. 2 shows the normalized frequency response for GCN, self-attention, mean pooling, and fractal nodes. Self-attention shows a prominent response in low and high frequencies but with a potential overemphasis on global information. Mean pooling shows a minimal response, primarily in the low-frequency domain, which suggests an oversimplification of node representations by losing local details. In contrast, fractal nodes show a distinctive response for low and high frequencies. The prominent low-frequency response captures the global context of subgraphs, while the elevated high-frequency response ensures the retention of fine-grained, local de-

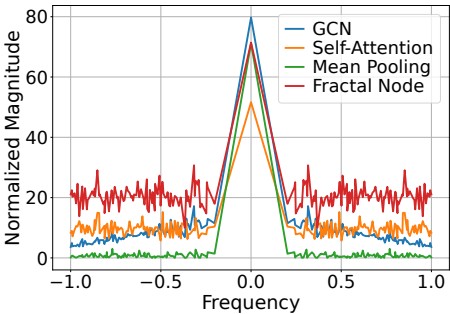

Figure 2: Normalized frequency response on PEPTIDES-STRUCT.

tails. This unique combination highlights the ability of fractal nodes to encode subgraph-level patterns while preserving node-level distinctions.

**Self-similarity in structural patterns and feature representations.** Our fractal nodes work by using structural and feature self-similarity. In structural perspective, we observe it through node centrality distributions at various scales. We use betweenness centrality (Freeman, 1977) as it captures local and global structural importance, particularly in networks where even low-degree nodes can be critical bridges (Kitsak et al., 2007). We partition the graphs into different numbers of subgraphs and compare the distributions between the original graph and its subgraphs (See Appendix C for more details). As shown in Fig. 3, the structural similarity increases with the number of subgraphs.

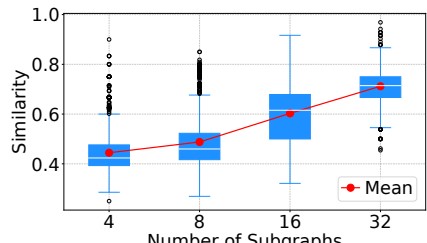

Figure 3: Structural similarity of node centrality distribution in PEPTIDES.

From the feature perspective, our fractal nodes go beyond structural patterns by adaptively combining LPF and HPF to represent both global and local features. By using the learnable parameter $\omega_c^{(\ell)}$, fractal nodes can represent multi-scale feature effectively. While mean pooling only retains global information through DC components, our approach preserves global patterns and local variations in the feature space. This dual consideration allows our method to better capture the inherent self-similarity of real-world networks.

**Expressive power of fractal nodes.** The expressive power of fractal nodes can be understood through the lens of existing theoretical results on subgraph-based approaches. The methods have been shown to increase expressive power beyond MPNNs. Encoding local subgraphs is stronger than 1-WL and 2-WL tests (Zhao et al., 2022, Theorem 4.3). In the context of subgraph WL (SWL) test (Zhang et al., 2023), fractal nodes achieve expressive power comparable to SWL with additional single-point aggregation and potentially approach SWL with additional global aggregation (Zhang et al., 2023, Theorem 4.4), as the fractal nodes implicitly perform a form of global aggregation within each subgraph. We will empirically verify expressive power in Section 5.2.

## 4.2 MODEL COMPLEXITY

Our fractal nodes show improvements in computational efficiency compared to Transformer-based models such as graph Transformers (Dwivedi & Bresson, 2021) and GraphGPS (Rampášek et al., 2022). The time complexity of our FN method is $\mathcal{O}(L(|\mathcal{V}| + |\mathcal{E}|))$, where $L$ is the number of layers, $|\mathcal{V}|$ is the number of nodes, and $|\mathcal{E}|$ is the number of edges. The $\mathsf{FN}_M$ introduces an additional mixing step through the MLP-Mixer, leading to a time complexity of $\mathcal{O}(L(|\mathcal{V}| + |\mathcal{E}|) + Cd^2)$. $C$ is the number of subgraphs and $d$ is the hidden dimension. Given that $C$ is much smaller than $|\mathcal{V}|$, this term does not dominate the overall complexity, preserving the efficiency of the model. In contrast, graph Transformers incur a time complexity of $\mathcal{O}(L(|\mathcal{V}|^2))$, due to the quadratic cost of computing self-attention over all node pairs, which is expensive for large graphs. Similary, GraphGPS combines MPNNs with self-attention, resulting in comparable quadratic complexity $\mathcal{O}(L(|\mathcal{V}|^2))$. Thus, fractal nodes offer a computational advantage over graph Transformer-based methods.

## 4.3 COMPARISON WITH PRIOR WORK

**Comparison to graph coarsening methods.** Coarformer (Kuang et al., 2022) tries to use coarsened and original graphs as separate views, where the coarsened graph is input to the Transformer, while ANS-GT (Cai et al., 2021) feeds a sequence of node representations to the graph Transformer by combining original, global, and coarsened node representations formed via adaptive sampling. Our method, on the other hand, incorporates fractal nodes representing subgraph information into the MPNN and enables fractal nodes to exchange messages with the original nodes and exchange information between fractal nodes via MLP-Mixer.

**Comparison to virtual node.** If we do not split into subgraphs, there will be only one fractal node. This can be compared to a virtual node (Gilmer et al., 2017; Hwang et al., 2022; Cai et al., 2023), which is known to have the information of a global node. While both approaches facilitate global or subgraph-level global information exchange, the key difference lies in how they process information. Virtual nodes aggregate global information from the entire graph, whereas fractal nodes operate at a subgraph level. A virtual node has its own update and aggregation functions that process messages

from all graph nodes, while regular nodes incorporate both their local neighborhood messages and the virtual node's message. In contrast, our fractal nodes adaptively decompose and process both low and high frequency components of subgraph features. This allows fractal nodes to capture richer information at the subgraph level compared to virtual node implementations that typically aggregate global information.

## 5 EXPERIMENTS

To evaluate the effectiveness of our proposed fractal nodes, we conduct extensive experiments on various tasks. We aim to answer the following key questions: **(Q1.)** Can fractal nodes mitigate over-squashing compared to MPNNs? **(Q2.)** Do fractal nodes improve expressiveness compared to MPNNs? **(Q3.)** How do fractal nodes compare to MPNNs and other graph Transformers in terms of performance on various benchmark datasets? **(Q4.)** Does the lower theoretical complexity of fractal nodes lead to faster run time? Through this experiment, we aim to determine if fractal nodes provide meaningful benefits. Afterwards, we perform a series of ablation and sensitivity analyses.

### 5.1 ANALYSIS ON OVER-SQUASHING (**Q1.**)

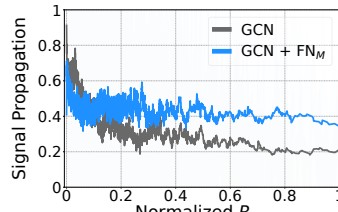

**Singal propagation and effective resistance.** The signal propagation of MPNNs is inversely proportional to the total effective resistance $R_{tot}$ (Di Giovanni et al., 2023). Consistent with our theoretical analysis in Theorems 4.1 and 4.2, this motivates us to check if adding fractal nodes help maintain signal flow across a graph with high $R_{tot}$. $R_{tot}$ is the total effective resistance between all pairs of nodes in a graph. The theoretical details of effective resistance and signal propagation are provided in Appendix F. The results in Section 5.1 validate our theoretical predictions – GCN+FN mitigates the decay of signal propagation with higher $R_{tot}$ compared to GCN. GCN fails to maintain the magnitude of signal flow under severe bottleneck structure, indicated as higher

Figure 4: The amount of signal propagated across the graphs w.r.t. the normalized $R_{tot}$ in PEPTIDES-FUNC. More results are in Appendix F.

total effective resistance. In contrast, GCN+FN$_M$ demonstrates resilience to over-squashing and maintains higher levels of signal propagation even under the highest $R_{tot}$. This improvement can be attributed to fractal nodes, which serve as single-hop shortcuts to connect all nodes and enable efficient long-range interactions by exchanging the features across them through MLP-Mixer layer.

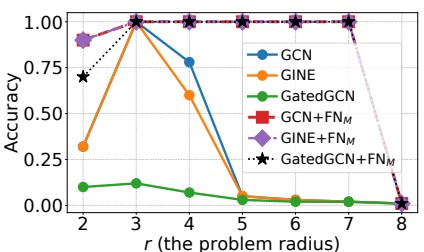

**Fractal nodes alleviates over-squashing.** We evaluate our fractal nodes on the TREENEIGHBOURSMATCH proposed by Alon & Yahav (2021), which has tree structures that show fractal-like properties. The dataset helps evaluate over-squashing. In this dataset, each example consists of a binary tree of depth $r$, with the task of predicting the label for target node by matching its degree of neighbors with a leaf node. As shown in Fig. 5, standard MPNNs (i.e., GCN, GINE, GatedGCN) fail to generalize for $r > 4$, while our fractal nodes mitigate over-squashing and generalize well up to $r = 7$. We empirically show that MPNNs augmented with fractal nodes can directly propagate long-distance information, avoiding the over-squashing problem.

Figure 5: Test accuracy in the TREENEIGHBOURMATCH problem.

### 5.2 EXPRESSIVE POWER OF FRACTAL NODES (**Q2.**)

We experimentally evaluate the expressive power of fractal nodes on 3 simulated datasets: CSL (Murphy et al., 2019), EXP (Abboud et al., 2021), and SR25 (Balcilar et al., 2021). Each dataset contains graphs that are indistinguishable by the 1 to 3-WL test, and detailed descriptions are provided in Appendix D.1. Table 1 shows that our model

Table 1: Synthetic results (Accuracy ↑)

| Method | CSL | SR25 | EXP |
|---|---|---|---|
| GCN | 10.00 | 6.67 | 52.17 |
| GINE | 10.00 | 6.67 | 51.35 |
| GatedGCN | 10.00 | 6.67 | 51.25 |
| GCN + FN$_M$ | 39.67 | 100.0 | 86.40 |
| GINE + FN$_M$ | 47.33 | 100.0 | 95.58 |
| GatedGCN + FN$_M$ | 49.67 | 100.0 | 96.50 |

Table 2: Test performance on two peptide datasets from LRGB (Dwivedi et al., 2022) and four other benchmark datasets (Hu et al., 2020; Dwivedi et al., 2023). ↑ denotes the higher the better and ↓ denotes the lower the better. Top three models are colored by **first**, **second**, **third**.

| Method | PEPTIDES-FUNC | PEPTIDES-STRUCT | MNIST | CIFAR10 | MOLHIV | MOLTOX21 |
|---|---|---|---|---|---|---|
| | AP ↑ | MAE ↓ | Accuracy ↑ | Accuracy ↑ | ROCAUC ↑ | ROCAUC ↑ |
| GCN | 0.6328±0.0023 | 0.2758±0.0012 | 0.9269±0.0023 | 0.5423±0.0056 | 0.7529±0.0098 | 0.7525±0.0031 |
| GINE | 0.6405±0.0086 | 0.2780±0.0021 | 0.9705±0.0023 | 0.6131±0.0035 | 0.7885±0.0034 | 0.7730±0.0064 |
| GatedGCN | 0.6300±0.0029 | 0.2778±0.0017 | 0.9776±0.0017 | 0.6628±0.0017 | 0.7874±0.0119 | 0.7641±0.0057 |
| GT | - | - | 0.9083±0.0016 | 0.5975±0.0029 | 0.7350±0.0040 | 0.7500±0.0060 |
| GraphiT | - | - | - | - | 0.7460±0.0100 | 0.7180±0.0130 |
| Graphormer | - | - | - | - | 0.7930±0.0040 | 0.7730±0.0800 |
| Transformer + LapPE | 0.6326±0.0126 | 0.2529±0.0016 | 0.9083±0.0016 | 0.5975±0.0029 | - | 0.7323±0.0057 |
| SAN + LapPE | 0.6384±0.0121 | 0.2683±0.0043 | - | - | 0.7775±0.0061 | 0.7130±0.0080 |
| EGT | - | - | 0.9817±0.0009 | 0.6870±0.0041 | - | - |
| GraphGPS | 0.6534±0.0091 | 0.2509±0.0014 | 0.9805±0.0013 | 0.7230±0.0036 | 0.7880±0.0101 | 0.7570±0.0040 |
| GRIT | **0.6988**±0.0082 | 0.2460±0.0012 | 0.9811±0.0011 | **0.7647**±0.0089 | - | - |
| Graph-ViT/MLP-Mixer | 0.6970±0.0080 | **0.2449**±0.0016 | **0.9846**±0.0009 | 0.7158±0.0009 | 0.7997±0.0102 | **0.7910**±0.0040 |
| Exphormer | 0.6527±0.0043 | 0.2481±0.0007 | **0.9841**±0.0035 | **0.7469**±0.0013 | - | - |
| GECO | **0.6975**±0.0025 | 0.2464±0.0009 | - | - | 0.7980±0.0200 | - |
| CRaWl | 0.6963±0.0079 | 0.2506±0.0022 | 0.9794±0.0050 | 0.6901±0.0026 | 0.7707±0.1490 | - |
| PNA | - | - | 0.9794±0.0012 | 0.7035±0.0063 | 0.7905±0.0132 | - |
| GNN-AK+ | 0.6480±0.0075 | 0.2736±0.0012 | - | 0.7219±0.0013 | 0.7961±0.0119 | - |
| SUN | 0.6730±0.0115 | 0.2498±0.0008 | - | - | 0.8003±0.0055 | - |
| CIN | - | - | - | - | **0.8094**±0.0057 | - |
| GCN + FN | 0.6802±0.0043 | 0.2530±0.0004 | 0.9393±0.0084 | 0.6006±0.0070 | 0.7564±0.0059 | 0.7670±0.0073 |
| GINE + FN | 0.6815±0.0059 | 0.2515±0.0020 | 0.9790±0.0012 | 0.6584±0.0069 | 0.7890±0.0104 | 0.7751±0.0029 |
| GatedGCN + FN | 0.6778±0.0056 | 0.2536±0.0019 | 0.9826±0.0012 | 0.7125±0.0035 | 0.7967±0.0098 | 0.7759±0.0054 |
| GCN + FN$_M$ | 0.6787±0.0048 | 0.2464±0.0014 | 0.9455±0.0004 | 0.6413±0.0068 | 0.7866±0.0034 | 0.7882±0.0041 |
| GINE + FN$_M$ | **0.7018**±0.0074 | **0.2446**±0.0018 | 0.9786±0.0004 | 0.6672±0.0068 | **0.8127**±0.0076 | **0.7926**±0.0021 |
| GatedGCN + FN$_M$ | 0.6950±0.0047 | **0.2453**±0.0014 | **0.9848**±0.0005 | **0.7526**±0.0033 | **0.8097**±0.0047 | **0.7922**±0.0054 |

achieves perfect accuracy on all 3 datasets while MPNNs fail (see detailed result in Appendix M). Our results are empirical but align with our discussion in Section 4.1.

## 5.3 EXPERIMENTS ON GRAPH BENCHMARKS (Q3.)

**Experimetnal setting and baselines.** We evaluate our method on two different types of tasks: graph-level prediction and large-scale node classification. For graph-level tasks, we use six benchmark datasets: two peptide datasets from LRGB (Dwivedi et al., 2022), two graph-level super-pixel image datasets from Benchmarking GNNs (Dwivedi et al., 2023), and two molecular datasets from OGB dataset (Hu et al., 2020). We compare our fractal nodes to MPNNs, graph Transformer-based models, and other state-of-the-art models: GCN (Kipf & Welling, 2017), GINE (Xu et al., 2019a), GatedGCN (Bresson & Laurent, 2017), GT (Dwivedi & Bresson, 2021), GraphiT (Mialon et al., 2021), Graphormer (Ying et al., 2021), Transformer + LapPE, SAN (Kreuzer et al., 2021a), EGT (Hussain et al., 2022), GraphGPS (Rampášek et al., 2022), GRIT (Ma et al., 2023), GraphViT/MLPMixer (He et al., 2023), Exphormer (Shirzad et al., 2023), GECO (Sancak et al., 2024), GNN-AK+ (Zhao et al., 2022), SUN (Frasca et al., 2022), CIN (Bodnar et al., 2021), CraWl (Tönshoff et al., 2023), and PNA (Corso et al., 2020). Detailed experimental settings for graph-level tasks are provided in Appendix D, while the setup and baseline comparisons for the large-scale node classification experiments are described separately in Appendix K.

**Results on graph-level tasks.** Our proposed fractal nodes (FN and FN$_M$) consistently enhance the performance of baseline MPNNs on all benchmark datasets, often surpassing graph Transformer models. In Table 2, for instance, on PEPTIDES-FUNC dataset, GINE+FN$_M$ achieves an average precision (AP) of 0.7018, outperforming both Exphormer and GraphGPS. The capabilities of base MPNN impact performance outcomes. Our fractal nodes framework is model-agnostic and augments various MPNNs. Our fractal nodes capture global information at the subgraph level through low and high-pass filtering and enable long-range interactions without self-attention layers. The superior performance of GRIT on CIFAR10 stems from its self-attention, positional encoding, and degree scalers. Our comparable performance with Graph-ViT and Exphormer on MNIST shows that fractal nodes can effectively capture local and global information without self-attention layer.

**Results on large-scale graphs.** The effectiveness of our method is particularly evident in large-scale graph experiments in Table 16 of Appendix K. On ogbn-arxiv, GCN+FN improves accuracy from 71.74% to 73.03%, while on ogbn-product, GraphSAGE+FN$_M$ demonstrates a substantial improvement from 78.29% to 83.11%. These improvements are achieved while maintaining the computational efficiency of MPNNs, offering a more practical alternative to graph Transformers for large-scale graph learning tasks.

### 5.4 Runtime Comparison (Q4.)

As we discussed in Section 4.2, our fractal nodes provide benefits in capturing long-range dependencies without increasing computational complexity. As shown in Table 3, GCN+FN results in only a slight runtime increase compared to base MPNNs. This efficiency extends to large-scale graphs (see Appendix J.2) — on ogbn-arxiv, GCN+FN maintains identical computational requirements to GCN. Even with FN$_M$, the overhead remains minimal and far below graph Transformers such as GraphGPS and Exphormer. Our empirical analysis of graph partitioning algorithms (detailed in Table 14) shows that using METIS with $\mathcal{O}(|\mathcal{E}|)$ complexity enables efficient fractal node creation even for large graphs such as ogbn-arxiv and ogbn-products. Given these results shown in Table 16, we believe our method achieves a balance between accuracy and computational efficiency.

Table 3: Runtime and memory consumption on PETIDES-FUNC.

|  | Time/epoch | Memory |
|---|---|---|
| GCN | 4.04 s | 250 MB |
| Trans.+LapPE | 10.01 s | 6,661 MB |
| GraphGPS | 12.01 s | 6,904 MB |
| GCN + FN | 5.03 s | 512 MB |
| GCN + FN$_M$ | 6.17 s | 667 MB |

### 5.5 Ablation, Sensitivity, and Additional Studies

We report ablation studies for $\omega_c^{(\ell)}$ and HPF in Appendices E.1 and E.2. We report results when $\omega_c^{(\ell)}$ is zero, that is, without HPF, and when we use either a scalar parameter (denoted 'SC') or a learnable vector parameter (denoted 'VC'). We also report sensitivity studies on $C$, i.e., the number of fractal nodes, and additional analyses on a variant of message passing between fractal nodes in Appendices E.5 and E.6. Analysis of the use of partitioning algorithms other than METIS is reported in Appendix I.

**Fractal nodes vs. augmented MPNNs.** We compare our fractal nodes to 6 augmented MPNNs including graph rewiring methods: DIGL (Gasteiger et al., 2019), SDRF (Topping et al., 2022), FoSR (Karhadkar et al., 2023), BORF (Nguyen et al., 2023), GTR (Black et al., 2023), PANDA (Choi et al., 2024), and LASER (Barbero et al., 2023) (see Appendix E.3 for detail setup). If there is only one fractal node and no subgraph is created, our method can be reduced to the virtual node method, so we compare our fractal nodes and virtual nodes in Appendix E.4.

Table 4: Comparison to rewiring methods

| Method | PEPTIDES-FUNC | PEPTIDES-STRUCT |
|---|---|---|
|  | AP ↑ | MAE ↓ |
| GCN | $0.5930_{\pm 0.0023}$ | $0.3496_{\pm 0.0013}$ |
| + FoSR | $0.5947_{\pm 0.0035}$ | $0.3473_{\pm 0.0007}$ |
| + GTR | $0.5075_{\pm 0.0029}$ | $0.3618_{\pm 0.0010}$ |
| + SDRF | $0.5947_{\pm 0.0126}$ | $0.3478_{\pm 0.0013}$ |
| + BORF | $0.5994_{\pm 0.0037}$ | $0.3514_{\pm 0.0009}$ |
| + PANDA | $\mathbf{0.6028_{\pm 0.0031}}$ | $\mathbf{0.3272_{\pm 0.0001}}$ |
| + LASER | $\mathbf{0.6440_{\pm 0.0010}}$ | $\mathbf{0.3043_{\pm 0.0019}}$ |
| + FN | $\mathbf{0.6445_{\pm 0.0057}}$ | $\mathbf{0.2535_{\pm 0.0012}}$ |

## 6 Concluding Remark

We introduced the fractal nodes to enforce self-similarity into MPNNs, inspired by the fractal nature of real-world networks. Our method effectively combines local and global graph information, addressing limitations of both MPNNs and graph Transformers. Experimental results on 6 benchmark datasets show the superiority of our approach, consistently improving the performance of MPNNs and competing advantageously with state-of-the-art graph Transformers-based methods.

**Limitations and future directions.** While fractal nodes are effective, they are currently designed to extend MPNN architectures. Although efficient and widely used, the use of METIS for subgraph partitioning may not be optimal for all types of graphs. While alternative partitioning methods could be used for large-scale graphs, the computational efficiency of METIS limits our options for more computationally intensive partitioning approaches. Future work could explore better ways to construct subgraphs at scale, and it may be worthwhile to investigate extending our fractal nodes in ways better suited for graph Transformers.

## ETHICAL STATEMENTS

In terms of the broader impact of this research on society, we do not see the very negative impacts that might be expected.

## REPRODUCIBILITY STATEMENT

To ensure reproducibility and completeness, we have included appendices in this paper. Appendix A provides a proof of Theorem 3.1. We provide details of our experiments presented in the paper in Appendix D. Only a part of the source code that reproduces the experiments is available at https://sites.google.com/view/fractalnode/. We plan to make all the code available after acceptance.

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

# Supplementary Materials for "Fractal-Inspired Message Passing Neural Networks with Fractal Nodes"

# Table of Contents

# A  PROOF OF THEOREM 3.1

**Theorem 3.1** (Mean pooling as a low-pass filter capturing the DC component). *Let $h_v$ represent the hidden state of node $v$ in subgraph $\mathcal{V}_c$ and let $H_c = [h_1, h_2, \ldots, h_n] \in \mathbb{R}^{n \times d}$ be the matrix of node features for all nodes in $\mathcal{V}_c$ where $n = |\mathcal{V}_c|$ is the number of nodes in the subgraph. The mean pooling operation applied to the node features is equivalent to extracting the DC or the lowest frequency component of the signal in the frequency domain.*

*Proof.* The mean pooling operation aggreagated the features of all nodes in the subgraph or graph by computing the average,

$$f_c^{mean} = \frac{1}{n} \sum_{v \in \mathcal{C}_c} h_v. \tag{14}$$

To understand this operation in the frequency domain, we use discrete Fourier transform (DFT), which transforms the node feature matrix $H_c$ into its frequency domain. The DFT of a signal $h_v$ is represented as:

$$\mathcal{F}(h_v) = \mathsf{DFT} \cdot h_v, \tag{15}$$

where $\mathsf{DFT} \in \mathbb{C}^{n \times n}$ is the Fourier matrix. The rows of the Fourier matrix are given by the Fourier basis vectors, which are complex exponential functions. These basis vectors represent different frequencies, and each row in the DFT corresponds to a specific frequency component. The first row of the Fourier matrix $\mathsf{DFT}$ corresponds to the DC component, which is the lowest frequency component of the signal. This row is a vector of ones:

$$\mathsf{DFT}[1, :] = \frac{1}{\sqrt{n}} \cdot [1, 1, \ldots, 1]. \tag{16}$$

This row corresponds to the mean or average of the signal. Therefore, when we project the input signal onto this basis vector, we are effectively extracting the global, smooth structure of the signal.

The DC component of the DFT is then expressed as:

$$DC[x] = \mathsf{DFT}^{-1} \mathrm{diag}(1, 0, \ldots, 0) \mathsf{DFT} x = \frac{1}{n} 11^\intercal x. \tag{17}$$

This operation corresponds to projecting the input signal $x$ onto the vector of ones, effectively averaging all elements of $x$, which is exatly the result of mean pooling:

$$f_c^{DC} = \frac{1}{n} 11^\intercal H_c = \frac{1}{n} \sum_{v \in \mathcal{C}_c} h_v. \tag{18}$$

$\square$

Therefore, mean pooling captures the DC component of the signal, which is the lowest frequency component. This corresponds to extracting the global, smooth node features of the subgraph, but it does not retain higher-frequency variations, which represent the local details.

Thus, mean pooling is equivalent to applying a low-pass filter that only retains the DC component of the signal.

# B  IMPLEMENTATION DETAIL

## B.1  METIS PARTITIONING FOR FRACTAL NODE CREATION

To create fractal nodes, we employ METIS (Karypis & Kumar, 1998), a graph clustering algorithm known for its excellent balance between accuracy and computational efficiency. METIS partitions a graph into a pre-defined number of clusters, maximizing within-cluster connections while minimizing between-cluster links. This approach effectively captures the community structure of the graph.

However, using non-overlapping partitions could result in the loss of important edge information, particularly at the boundaries between partitions. To address this issue and retain all original edges,

we introduce overlapping subgraph. After the initial METIS partitioning, we expand each partition to include nodes from neighboring partitions.

Formally, we first apply METIS to partition a graph $\mathcal{G}$ into $C$ non-overlapping subgraphs: $\{\mathcal{V}_1, \ldots, \mathcal{V}_C\}$ such that $\mathcal{V} = \{\mathcal{V}_1 \cup \ldots \cup \mathcal{V}_C\}$ and $\mathcal{V}_i \cap \mathcal{V}_j = \varnothing, \forall i \neq j$, where $C$ is the number of fractal nodes or subgraphs. Then, we expand these subgraphs to include $k$-hop neighborhoods:

$$\mathcal{V}_i \leftarrow \mathcal{V}_i \cup \{\mathcal{N}_k(j) | j \in \mathcal{V}_i\}, \tag{19}$$

where $\mathcal{N}_k(j)$ defines the $k$-hop neighbourhood of node $j$. This expansion ensures that each subgraph retains information about its immediate surroundings. The choice of $k$ allows us to control the degree of overlap between subgraphs. A larger $k$ value increases the overlap, potentially capturing more global information but at the cost of increased computational complexity. This overlapping subgraph approach allows our fractal nodes to capture both local structural details and broader subgraph-level information, enhancing the model's ability to learn multi-scale representations of the graph structure.

### B.2 INSTANCE OF OUR FRAMEWORK

We describe update equations for how our fractal node is applied to MPNN. The update equation for GatedGCN + FN is the following:

$$\begin{aligned}
\widetilde{h}_{v,c}^{(\ell+1)} &= \sigma\Big(\Omega^{(\ell)} h_{v,c}^{(\ell)} + \sum_{u \in N(v)} \mathsf{gate}^{(\ell)}(h_{v,c}^{(\ell)}, h_{u,c}^{(\ell)}) \odot h_{u,c}^{(\ell)} W_1^{(\ell)}\Big), \\
f_c^{(\ell+1)} &= \mathsf{LPF}(\widetilde{h}_{v,c}^{(\ell+1)}) + \omega^{(\ell)} \cdot \mathsf{HPF}(\widetilde{h}_{v,c}^{(\ell+1)}), \\
h_{v,c}^{(\ell+1)} &= \widetilde{h}_{v,c}^{(\ell+1)} + f_c^{(\ell+1)}, \\
\mathsf{gate}^{(\ell)}(h_{v,c}^{(\ell)}, h_{u,c}^{(\ell)}) &= \mathsf{sigmoid}(W2^{(\ell)} h_{v,c}^{(\ell)} + W_3^{(\ell)}) h_{u,c}^{(\ell)},
\end{aligned} \tag{20}$$

where $\sigma$ is a ReLU activation function, $W_0^{(\ell)}$, $W_1^{(\ell)}$, $W_2^{(\ell)}$, $W_3^{(\ell)}$ are learnable weight matrices, $\mathsf{gate}^{(\ell)}$ is a gating mechanism that controls the information flow between nodes.

The update equation for GINE + FN is the following:

$$\begin{aligned}
\widetilde{h}_{v,c}^{(\ell+1)} &= \mathsf{MLP}^{(\ell)}\Big((1 + \epsilon^{(\ell)}) \cdot h_{v,c}^{(\ell)} + \sum_{u \in N(v)} \sigma(h_{u,c}^{(\ell)} + e_{uv}^{(\ell)})\Big), \\
f_c^{(\ell+1)} &= \mathsf{LPF}(\widetilde{h}_{v,c}^{(\ell+1)}) + \omega_c^{(\ell)} \cdot \mathsf{HPF}(\widetilde{h}v, c^{(\ell+1)}), \\
h_{v,c}^{(\ell+1)} &= \widetilde{h}_{v,c}^{(\ell+1)} + f_c^{(\ell+1)},
\end{aligned} \tag{21}$$

where $\epsilon^{(\ell)}$ is a learnable scalar parameter, and $e_{uv}^{(\ell)}$ is a edge hidden vector between node $u$ and $v$.

Note that the positional encoding scheme and readout function schemes can also be applied to MPNNs with fractal nodes.

### B.3 POSITIONAL ENCODING

When we integrate our fractal node to MPNN, we incorporate two distinct positional encodings (PE): an absolute PE for individual nodes and a relative PE for fractal nodes.

For node-level encoding, we consider dataset-specific approaches. We utilize random-walk structural encoding (RWSE) for molecular graphs and Laplacian eigenvector encodings for super-pixel image-based tasks. To enhance robustness, we randomly flip the sign of Laplacian eigenvectors during training.

Let $M \in \{0, 1\}^{C \times |\mathcal{V}|}$ be a binary matrix where each row corresponds to a fractal node and each column to an original graph node. $M_{ij} = 1$ if node $j$ belongs to fractal node $i$, and 0 otherwise. Then, the coarsened adjacency matrix is computed as $A^C = MM^\top$. This operation effectively counts the number of connections between fractal nodes, where $A_{ij}^C$ represents the number of edges between fractal nodes $i$ and $j$ in the original graph. We then derive a positional encoding $p_v \in \mathbb{R}^{d_p}$ for each

fractal node from this coarsened adjacency matrix. This encoding is incorporated into the fractal node representation through a linear transformation:

$$f_v^{(L)} = Tp_v + Of_v^{(L)} + b \in \mathbb{R}^d, \tag{22}$$

where $T \in \mathbb{R}^{d \times d_p}$ and $O \in \mathbb{R}^{d \times d}$ are learnable transformation matrices, and $b \in \mathbb{R}^d$ is a learnable bias vector.

By incorporating relative positional information between fractal nodes, we enable the $\mathsf{FN}_M$ variant to better use the hierarchical structure of the graph.

## C  STRUCTURAL SELF-SIMILARITY AND NODE CENTRALITY

In this section, we describe how we calculate the self-similarity of a network by comparing the node centrality distributions between the original graph and its subgraphs using betweenness centrality. Specifically, we use the Kolmogorov-Smirnov (KS) test to measure the similarity between these distributions.

**Fractality definition.**  We define the fractality of a graph as the degree to which the properties of the subgraphs resemble those of the original graph when the graph is partitioned consistently. In this work, we focus on how the betweenness centrality distribution of the original graph compares to those of its subgraphs.

Let $\Psi(x)$ represent the node centrality distribution function for the original graph, and let $\Psi_0(x), \ldots, \Psi_{32}(x)$ represent the centrality distributions for each of the subgraphs obtained by partitioning the original graph into 32 subgraphs. We aim to quantify the similarity between $\Psi(x)$ and the subgraph distributions using the KS test.

**Kolmogorov-Smirnov test.**  The KS test is a non-parametric test that compares the empirical cumulative distribution function (CDF) $\Psi_n(x)$ of the sample (subgraph centrality) with the CDF $\Psi(x)$ of the reference distribution (original graph centrality). The KS test statistic $D$ is defined as:

$$D = \sup_x |\Psi_n(x) - \Psi(x)|, \tag{23}$$

where $D$ represents the maximum distance between the two CDFs. A smaller $D$ value indicates higher similarity between the two distributions.

**Similarity metric.**  We define the similarity between the original graph and a subgraph as $1 - D$, where $D$ is the KS test statistic. Therefore, a higher $1 - D$ value implies greater similarity. For each graph, we compute the similarity for all $C$ subgraphs, yielding $C$ similarity values.

**Fractality calculation.**  In our fractality evaluation, it is sufficient to identify the subgraph whose centrality distribution is most similar to that of the original graph. This is because not all subgraphs need to exhibit self-similarity for the graph to be considered fractal-like; the presence of one or more highly similar subgraphs is indicative of fractality. Thus, we take the maximum of the $C$ similarity values $(1 - D)$ as the self-similarity score for the graph:

$$\text{Self-Similarity Score} = \max_i (1 - D_i), \tag{24}$$

where $D_i$ is the KS test statistic for the $i$-th subgraph. This approach allows us to compute a self-similarity score for a single graph based on betweenness centrality. The comparison according to the number of subgraphs is shown in Fig. 6.

## D  EXPERIMENTAL DETAILS

In this section, we provide further details about our experiments.

### D.1  DATASET DESCRIPTION

We provide the descriptions and statistics of all datasets used in our experiments.

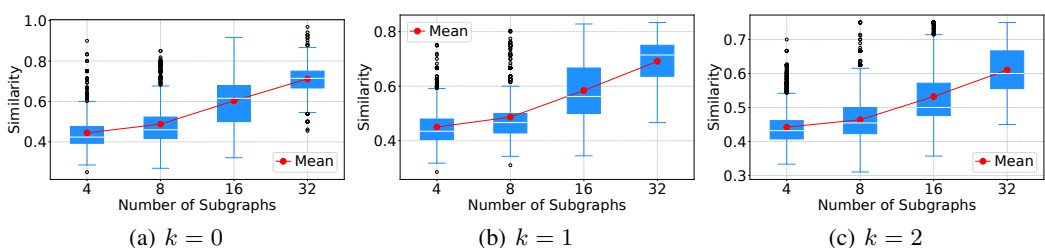

(a) $k = 0$        (b) $k = 1$        (c) $k = 2$

Figure 6: Similarity of node centrality distribution in PEPTIDE-STRUCT.

**PEPTIDES-FUNC & PEPTIDES-STRUCT.** (CC BY-NC 4.0 License) (Dwivedi et al., 2022): These datasets comprise 16K atomic peptide graphs from SAT-Pdb, with residues as nodes. They differ in their graph-level tasks: PETIDES-FUNC is a multi-label classification task with 10 nonexclusive functional classes, while PEPTIDES-STRUCT involves regression on 11 3D structural properties. Dataset splitting utilizes meta-class holdout based on original peptide labels.

**MNIST & CIFAR10.** (CC BY-SA 3.0 and MIT License): These datasets adapt popular image classification tasks to graph classification. Images are converted to graphs using super-pixels, representing homogeneous intensity regions. Both are 10-class classification tasks following standard splits: 55K/5K/10K for MNIST and 45K/5K/10K for CIFAR10 (train/validation/test).

**MOLHIV & MOLTOX21.** (MIT License) (Hu et al., 2020): These molecular property prediction datasets use common node and edge features representing chemophysical properties, pre-processed with RDKit (RDKit, online). Molecules are represented as graphs with atoms as nodes and chemical bonds as edges. Node features are 9-dimensional, including atomic number, chirality, and other properties. Predefined scaffold partitions are used: MOLTOX21 6K/0.78K/0.78K and MOLHIV 32K/4K/4K for training/validation/test.

**CSL.** CSL (Murphy et al., 2019) is a synthetic dataset testing GNN expressivity, containing 150 4-regular graphs in 10 isomorphism classes. These graphs, indistinguishable by 1-WL tests, form cycles with skip-links. The task is to classify them into their respective isomorphism classes.

**EXP.** EXP (Abboud et al., 2021) consists of 600 graph pairs that 1&2-WL tests fail to distinguish, aiming to classify these into two categories.

**SR25.** SR25 (Balcilar et al., 2021) consists of 15 strongly regular graphs (3-WL indistinguishable) with 25 nodes each, forming a 15-way classification problem.

**TREENEIGHBOURMATCH.** Proposed by Alon & Yahav (2021), this synthetic dataset highlights over-squashing in MPNNs. It uses binary trees of depth $r$ (problem radius), requiring information propagation from leaves to a target node for label prediction, thus demonstrating over-squashing issues.

### D.2 HARDWARE SPECIFICATIONS AND LIBRARIES

We have implemented our method using PYTORCH-GEOMETRIC, and built on the source code of Rampášek et al. (2022)[2] and He et al. (2023)[3]. All experiments were performed using the following software and hardware environments: UBUNTU 18.04 LTS, PYTHON 3.7.13, PYTORCH 1.12.1, PYTORCH GEOMETRIC 2.5.2, , TORCH-SCATTER 2.1.0, TORCH-SPARSE 0.6.16, NUMPY 1.24.3, METIS 0.2a5, CUDA 11.3, NVIDIA Driver 465.19, i9 CPU, NVIDIA RTX 3090/A6000.

---

[2] https://github.com/rampasek/GraphGPS
[3] https://github.com/XiaoxinHe/Graph-ViT-MLPMixer

## D.3 SETUP & HYPERPARAMETERS

We use the same learning rates and weight decay to GCN, GINE, and GatedGCN, and the hyperparameters we considered are shown in Tables 5 to 7. The experimental results of MPNN are the same as the results using positional encoding, and we use the setup of He et al. (2023).

In Tables 5 to 7, we report the hyperparameters used in our experiments.

Table 5: Hyperparameter search space of fractal nodes for benchmark datasets

| Hyperparameters | Search Space |
|---|---|
| $\omega_c^{(\ell)}$ | {SC, VC} |
| $C$ | {4, 8, 16, 32} |
| HPF | {True, False} |
| $k$-hop | {0, 1, 2} |
| $L$ | {2, 3, 4, 5, 6, 7, 8} |
| $L_M$ | {1, 2, 4} |

Table 6: Best hyperparameter of FN for PEPTIDES-FUNC, PEPTIDES-STRUCT, MNIST, CIFAR10, MOLHIV, and MOLTOX21.

| Hyperparameter | Method | PEPTIDES-FUNC | PEPTIDES-STRUCT | MNIST | CIFAR10 | MOLHIV | MOLTOX21 |
|---|---|---|---|---|---|---|---|
| $\omega_c^{(\ell)}$ | GCN | VC | SC | VC | VC | VC | SC |
| | GINE | SC | VC | VC | VC | SC | SC |
| | GatedGCN | SC | VC | VC | VC | VC | SC |
| $C$ | GCN | 32 | 32 | 32 | 32 | 32 | 32 |
| | GINE | 32 | 32 | 32 | 32 | 32 | 32 |
| | GatedGCN | 32 | 32 | 32 | 32 | 32 | 32 |
| HPF | GCN | True | True | True | True | True | True |
| | GINE | True | True | True | True | True | True |
| | GatedGCN | True | True | True | True | True | True |
| $k$-hop | GCN | 1 | 1 | 1 | 1 | 1 | 1 |
| | GINE | 1 | 1 | 1 | 1 | 1 | 1 |
| | GatedGCN | 1 | 1 | 1 | 1 | 1 | 1 |
| $L$ | GCN | 4 | 4 | 4 | 7 | 2 | 4 |
| | GINE | 4 | 4 | 4 | 7 | 2 | 4 |
| | GatedGCN | 4 | 4 | 4 | 7 | 2 | 4 |

Table 7: Best hyperparameter of $\text{FN}_M$ for PEPTIDES-FUNC, PEPTIDES-STRUCT, MNIST, CIFAR10, MOLHIV, and MOLTOX21.

| Hyperparameter | Method | PEPTIDES-FUNC | PEPTIDES-STRUCT | MNIST | CIFAR10 | MOLHIV | MOLTOX21 |
|---|---|---|---|---|---|---|---|
| $\omega_c^{(\ell)}$ | GCN | VC | SC | VC | VC | VC | VC |
| | GINE | SC | VC | VC | VC | VC | SC |
| | GatedGCN | VC | SC | VC | VC | VC | VC |
| $C$ | GCN | 32 | 32 | 32 | 32 | 32 | 32 |
| | GINE | 32 | 16 | 32 | 32 | 32 | 32 |
| | GatedGCN | 32 | 32 | 4 | 4 | 32 | 32 |
| HPF | GCN | True | True | True | True | True | True |
| | GINE | True | True | True | True | True | True |
| | GatedGCN | True | True | False | True | True | True |
| $k$-hop | GCN | 1 | 1 | 1 | 1 | 2 | 1 |
| | GINE | 1 | 1 | 1 | 1 | 2 | 1 |
| | GatedGCN | 1 | 1 | 1 | 1 | 2 | 1 |
| $L$ | GCN | 4 | 4 | 4 | 7 | 2 | 5 |
| | GINE | 4 | 4 | 4 | 7 | 2 | 4 |
| | GatedGCN | 4 | 4 | 4 | 8 | 2 | 5 |
| $L_{\text{M}}$ | GCN | 2 | 2 | 4 | 1 | 2 | 4 |
| | GINE | 2 | 2 | 4 | 1 | 2 | 4 |
| | GatedGCN | 2 | 2 | 4 | 1 | 2 | 4 |

# E  ABLATION, SENSITIVITY AND ADDITIONAL STUDIES

## E.1  IMPACT OF HPF

We use both LPF and HPF to create fractal nodes, as shown in Equation (5). We analyze the cases when $\omega_c^{(\ell)}$ is 0, i.e., with and without HPF. Our results are reported in Table 8, and we obtain the best performance when using HPF in almost all cases.

Table 8: Ablation study on HPF

| Method | HPF | PEPTIDES-FUNC | PEPTIDES-STRUCT | MNIST | CIFAR10 | MOLHIV | MOLTOX21 |
| | | AP ↑ | MAE ↓ | Accuracy ↑ | Accuracy ↑ | ROCAUC ↑ | ROCAUC ↑ |
|---|---|---|---|---|---|---|---|
| GCN + FN | True | **0.6802**±**0.0043** | **0.2530**±**0.0004** | **0.9393**±**0.0084** | **0.6006**±**0.0070** | **0.7564**±**0.0059** | **0.7670**±**0.0073** |
| | False | 0.6768±0.0016 | 0.2547±0.0023 | 0.9383±0.0102 | 0.5993±0.0081 | 0.7551±0.0084 | 0.7608±0.0093 |
| GINE + FN | True | **0.6815**±**0.0059** | **0.2515**±**0.0020** | **0.9790**±**0.0012** | **0.6584**±**0.0069** | **0.7882**±**0.0050** | **0.7751**±**0.0029** |
| | False | 0.6749±0.0111 | 0.2524±0.0021 | 0.9788±0.0008 | **0.6584**±**0.0069** | 0.7861±0.0054 | 0.7702±0.0045 |
| GatedGCN + FN | True | **0.6778**±**0.0071** | **0.2536**±**0.0019** | **0.9826**±**0.0012** | **0.7125**±**0.0035** | **0.7967**±**0.0098** | **0.7759**±**0.0054** |
| | False | 0.6661±0.0103 | 0.2609±0.0016 | 0.9801±0.0015 | 0.7010±0.0031 | 0.7908±0.0084 | 0.7674±0.0024 |
| GCN + FN$_M$ | True | **0.6787**±**0.0048** | 0.2464±0.0014 | **0.9455**±**0.0004** | **0.6413**±**0.0070** | **0.7866**±**0.0034** | **0.7882**±**0.0041** |
| | False | 0.6778±0.0056 | **0.2461**±**0.0022** | 0.9448±0.0007 | 0.6130±0.0080 | 0.7689±0.0124 | 0.7874±0.0080 |
| GINE + FN$_M$ | True | **0.7018**±**0.0074** | **0.2446**±**0.0018** | **0.9786**±**0.0004** | **0.6672**±**0.0068** | **0.8127**±**0.0076** | **0.7926**±**0.0021** |
| | False | 0.6647±0.0052 | 0.2484±0.0018 | 0.9744±0.0007 | 0.6670±0.0056 | 0.7959±0.0079 | 0.7895±0.0067 |
| GatedGCN + FN$_M$ | True | **0.6950**±**0.0047** | **0.2453**±**0.0014** | 0.9836±0.0010 | **0.7526**±**0.0033** | **0.8097**±**0.0047** | **0.7922**±**0.0054** |
| | False | 0.6900±0.0055 | 0.2477±0.0005 | **0.9848**±**0.0005** | 0.7501±0.0042 | 0.7930±0.0057 | 0.7883±0.0067 |

## E.2  IMPACT OF TYPE OF $\omega_c^{(\ell)}$

When creating a fractal node, we can use a learnable scalar parameter (denoted as 'SC') or a learnable vector parameter (denoted as 'VC') to make the contribution of high frequency components. We report the results in Table 9.

Table 9: Sensitivity study on $\omega_c^{(\ell)}$

| Method | $\omega_c^{(\ell)}$ | PEPTIDES-FUNC | PEPTIDES-STRUCT | MNIST | CIFAR10 | MOLHIV | MOLTOX21 |
| | | AP ↑ | MAE ↓ | Accuracy ↑ | Accuracy ↑ | ROCAUC ↑ | ROCAUC ↑ |
|---|---|---|---|---|---|---|---|
| GCN + FN | SC | 0.6797±0.0056 | **0.2530**±**0.0004** | 0.9377±0.0080 | 0.6003±0.0075 | 0.7553±0.0061 | **0.7670**±**0.0073** |
| | VC | **0.6802**±**0.0043** | 0.2535±0.0033 | **0.9393**±**0.0084** | **0.6006**±**0.0070** | **0.7564**±**0.0059** | 0.7667±0.0045 |
| GINE + FN | SC | **0.6815**±**0.0059** | 0.2534±0.0016 | 0.9784±0.0010 | 0.6548±0.0088 | **0.7882**±**0.0050** | **0.7751**±**0.0029** |
| | VC | 0.6796±0.0024 | **0.2515**±**0.0020** | **0.9790**±**0.0012** | **0.6584**±**0.0069** | 0.7849±0.0047 | 0.7672±0.0009 |
| GatedGCN + FN | SC | **0.6778**±**0.0071** | 0.2546±0.0020 | 0.9813±0.0018 | 0.7083±0.0032 | 0.7910±0.0090 | **0.7759**±**0.0054** |
| | VC | 0.6647±0.0052 | **0.2536**±**0.0019** | **0.9826**±**0.0012** | **0.7125**±**0.0035** | **0.7967**±**0.0098** | 0.7662±0.0090 |
| GCN + FN$_M$ | SC | 0.6773±0.0039 | **0.2464**±**0.0014** | 0.9444±0.0008 | 0.6405±0.0065 | 0.7762±0.0089 | **0.7882**±**0.0041** |
| | VC | **0.6787**±**0.0048** | 0.2485±0.0016 | **0.9455**±**0.0004** | **0.6413**±**0.0070** | **0.7866**±**0.0034** | 0.7862±0.0037 |
| GINE + FN$_M$ | SC | **0.7018**±**0.0074** | 0.2451±0.0011 | 0.9735±0.0009 | 0.6655±0.0066 | 0.8070±0.0084 | 0.7924±0.0019 |
| | VC | 0.6926±0.0105 | **0.2446**±**0.0018** | **0.9786**±**0.0004** | **0.6672**±**0.0068** | **0.8127**±**0.0076** | **0.7926**±**0.0021** |
| GatedGCN + FN$_M$ | SC | 0.6932±0.0056 | **0.2453**±**0.0014** | 0.9836±0.0010 | 0.7495±0.0051 | **0.8097**±**0.0047** | **0.7922**±**0.0054** |
| | VC | **0.6950**±**0.0047** | 0.2461±0.0009 | **0.9836**±**0.0009** | **0.7526**±**0.0033** | 0.8025±0.0087 | 0.7885±0.0043 |

## E.3  COMAPRISON TO GRAPH REWIRING METHODS

We compare our fractal nodes to no graph rewiring and 4 other state-of-the-art rewiring methods: DIGL (Gasteiger et al., 2019), SDRF (Topping et al., 2022), FoSR (Karhadkar et al., 2023), and BORF (Karhadkar et al., 2023). We also add the recent method, PANDA (Choi et al., 2024) to alleviate over-squashing without rewiring and the state-of-the-art method, LASER (Barbero et al., 2023). We replicate the experimental settings of Dwivedi et al. (2022) and use the results from Barbero et al. (2023). We choose the hidden dimension to respect the 500k parameter budget. In our fractal node, we opt out the positional encodings for a fair comparison.

### E.4 COMPARISON TO VIRTUAL NODE METHODS

To provide a comprehensive comparison with existing virtual node method, we compare with the two virtual node methods by Hu et al. (2020) (denoted as 'virtual node') and Rosenbluth et al. (2024) (denoted as 'VN'). As shwon in Table 10, both FN and $\text{FN}_M$ outperform the GCN and GIN models augmented with virtual nodes from Hu et al. (2020) on MOLHIV and MOLTOX21. On the Peptides datasets, our methods show competitive results with the VN method of Rosenbluth et al. (2024).

Table 10: Comparison to virtual node methods.

| Method | PEPTIDES-FUNC AP ↑ | PEPTIDES-STRUCT MAE ↓ | MOLHIV ROCAUC ↑ | MOLTOX21 ROCAUC ↑ |
|---|---|---|---|---|
| GCN + virtual node | - | - | $0.7599_{\pm 0.0119}$ | $0.7551_{\pm 0.0100}$ |
| GIN + virtual node | - | - | $0.7707_{\pm 0.0149}$ | $0.7621_{\pm 0.0062}$ |
| GCN + VN | $0.6732_{\pm 0.0066}$ | $0.2505_{\pm 0.0022}$ | - | - |
| GatedGCN + VN | $\mathbf{0.6823}_{\pm 0.0069}$ | $0.2475_{\pm 0.0018}$ | - | - |
| GCN + FN | $0.6802_{\pm 0.0043}$ | $0.2530_{\pm 0.0004}$ | $0.7564_{\pm 0.0059}$ | $0.7670_{\pm 0.0073}$ |
| GINE + FN | $0.6815_{\pm 0.0059}$ | $0.2515_{\pm 0.0020}$ | $0.7890_{\pm 0.0104}$ | $0.7751_{\pm 0.0029}$ |
| GatedGCN + FN | $0.6778_{\pm 0.0056}$ | $0.2536_{\pm 0.0019}$ | $\mathbf{0.7967}_{\pm 0.0098}$ | $0.7759_{\pm 0.0054}$ |
| GCN + $\text{FN}_M$ | $0.6787_{\pm 0.0048}$ | $\mathbf{0.2464}_{\pm 0.0014}$ | $0.7866_{\pm 0.0034}$ | $\mathbf{0.7882}_{\pm 0.0041}$ |
| GINE + $\text{FN}_M$ | $\mathbf{0.7018}_{\pm 0.0074}$ | $\mathbf{0.2446}_{\pm 0.0018}$ | $\mathbf{0.8127}_{\pm 0.0076}$ | $\mathbf{0.7926}_{\pm 0.0021}$ |
| GatedGCN + $\text{FN}_M$ | $\mathbf{0.6950}_{\pm 0.0047}$ | $\mathbf{0.2453}_{\pm 0.0014}$ | $\mathbf{0.8097}_{\pm 0.0047}$ | $\mathbf{0.7922}_{\pm 0.0054}$ |

### E.5 SENSITIVITY TO $C$

The analysis of sensitivity to the number of fractal nodes ($C$) reveals distinct performance patterns in various datasets. As shown in Fig. 7, for PEPTIDES-FUNC and PEPTIDES-STRUCT, there is relatively stable performance across different $C$ values, with GINE+$\text{FN}_M$ consistently outperforming the baseline GINE+FN. In MNIST, both GINE variants show an upward trend as $C$ increases, with GINE+$\text{FN}_M$ achieving peak accuracy at $C = 32$.

The optimal results are typically achieved at $C = 32$, which indicates that graph tasks benefit from finer-grained subgraph partitioning and additional mixing operations in $\text{FN}_M$. Overall, the results indicate that larger $C$ values (16 or 32) generally yield better performance for most datasets.

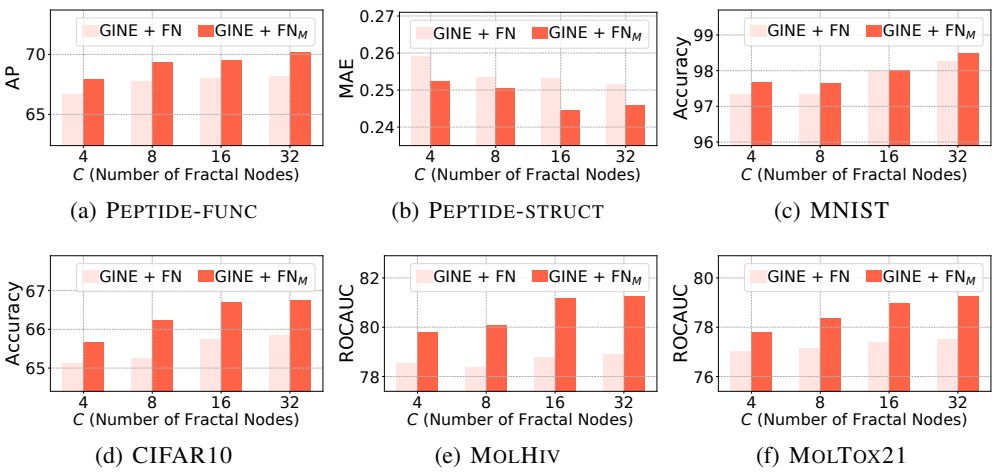

Figure 7: Sensitivity to $C$ with GINE.

### E.6 ADDITIONAL RESULTS ON ALL-LAYER FRATAL NODE MESSAGE PASSING

While our main $FN_M$ design uses an MLP-Mixer in the final layer for fractal node interactions, we also explored an alternative approach with message passing between fractal nodes across all layers (denoted as $FN_A$). This analysis aims to empirically validate our architectural choice.

Table 11 compares 3 variants: i) FN: is a base MPNN with no explicit fractal node interactions; ii) $FN_A$ is an all-layer message passing between fractal nodes; and iii) $FN_M$ is MLP-Mixer in final layer only (our proposed approach). The results show that while $FN_A$ shows some improvements over the base FN model in certain cases (e.g., MOLHIV accuracy improves from 0.7564 to 0.7783 for GCN), it consistently underperforms compared to our proposed $FN_M$ design. This pattern holds across different base architectures (GCN, GINE, GatedGCN) and datasets.

These empirical results validate our design choice of using MLP-Mixer in the final layer rather than implementing message passing between fractal nodes throughout all layers. This result indicates that the flexible mixing capabilities of the MLP-Mixer provide more effective fractal node interactions compared to explicit message passing approaches.

Table 11: Comparison on FN, $FN_A$ and $FN_M$

| Method | PEPTIDES-FUNC | PEPTIDES-STRUCT | MOLHIV | MOLTOX21 |
|---|---|---|---|---|
|  | AP ↑ | MAE ↓ | ROCAUC ↑ | ROCAUC ↑ |
| GCN + FN | $0.6802_{\pm 0.0043}$ | $0.2530_{\pm 0.0004}$ | $0.7564_{\pm 0.0059}$ | $0.7670_{\pm 0.0073}$ |
| GCN + $FN_A$ | $0.6582_{\pm 0.0032}$ | $0.2531_{\pm 0.0008}$ | $0.7783_{\pm 0.0164}$ | $0.7600_{\pm 0.0037}$ |
| GCN + $FN_M$ | $\mathbf{0.6787}_{\pm 0.0048}$ | $\mathbf{0.2464}_{\pm 0.0014}$ | $\mathbf{0.7866}_{\pm 0.0034}$ | $\mathbf{0.7882}_{\pm 0.0041}$ |
| GINE + FN | $0.6815_{\pm 0.0059}$ | $0.2515_{\pm 0.0020}$ | $0.7890_{\pm 0.0104}$ | $0.7751_{\pm 0.0029}$ |
| GINE + $FN_A$ | $0.6660_{\pm 0.0067}$ | $0.2530_{\pm 0.0011}$ | $0.8025_{\pm 0.0100}$ | $0.7680_{\pm 0.0056}$ |
| GINE + $FN_M$ | $\mathbf{0.7018}_{\pm 0.0074}$ | $\mathbf{0.2446}_{\pm 0.0018}$ | $\mathbf{0.8127}_{\pm 0.0076}$ | $\mathbf{0.7926}_{\pm 0.0021}$ |
| GatedGCN + FN | $0.6778_{\pm 0.0056}$ | $0.2536_{\pm 0.0019}$ | $0.7967_{\pm 0.0098}$ | $0.7759_{\pm 0.0054}$ |
| GatedGCN + $FN_A$ | $0.6658_{\pm 0.0048}$ | $0.2531_{\pm 0.0009}$ | $0.7898_{\pm 0.0065}$ | $0.7642_{\pm 0.0050}$ |
| GatedGCN + $FN_M$ | $\mathbf{0.6950}_{\pm 0.0047}$ | $\mathbf{0.2453}_{\pm 0.0014}$ | $\mathbf{0.8097}_{\pm 0.0047}$ | $\mathbf{0.7922}_{\pm 0.0054}$ |

## F EFFECTIVE RESISTANCE AND SIGNAL PROPAGATION

**Effective resistance and signal propagation.** Derived from the field of electrical engineering, the effective resistance between two nodes $u$ and $v$ in an electrical network is defined as the potential difference induced across the edges when a unit current is injected at one of each end (Ghosh et al., 2008). Intuitively, it provides a physical measure of the ease of signal flow from one end to the other. Rayleigh's monotonicity principle, which says that adding paths or shortening existing paths can only decrease the effective resistance between two nodes (Thomassen, 1990), leads to the following interpretation: more and shorter disjoint paths connecting the nodes $u$ and $v$ lead to a lower resistance between them (Black et al., 2023; Devriendt & Lambiotte, 2022). Therefore, edges with higher effective resistance have fewer alternative paths or shortcuts for signals passing through that edge and thus, struggle to propagate information, causing bottlenecks. The total effective resistance $R_{tot}$, the sum of the effective resistance among all pairs of nodes (see Equation (26)), is a key measure for measuring the overall degree of over-squashing across a graph.

**Total effective resistance.** The resistance between nodes $u$ and $v$ in the graph is given by

$$R_{u,v} = (1_u - 1_v)^\mathsf{T} \mathbf{L}^+ (1_u - 1_v), \tag{25}$$

where $\mathbf{L}$ is a Laplacian matrix, $1_v$ and $1_u$ are indicator vectors for node $u$ and $v$, respectively. Total effective resistance, $R_{tot}$, is defined as the sum of effective resistance between all pairs of nodes (Ghosh et al., 2008; Black et al., 2023):

$$R_{tot} = \sum_{u>v} R_{u,v} = n \cdot \mathrm{Tr}(\mathbf{L}^+) = n \sum_i^n \frac{1}{\lambda_i}, \tag{26}$$

where $\lambda_i$ is the $i$-th eigenvalues of $\mathbf{L}$ and $\mathbf{L}^+$ is the pseudoinverse of $\mathbf{L}$.

**Signal propagation w.r.t. effective resistance.** Here, we outline the experimental details for measuring signal propagation with respect to the normalized total effective resistance of the graphs. First, we randomly select a source node $v \in \mathcal{V}$, an entire node set, and assign $d$-dimensional feature vector to it, while all other nodes are initialized with zero vectors. Then, the amount of signal that has been propagated over the graph by the randomly initialized model with $\ell$ layers is given by

$$h_{\odot}^{(\ell)} = \frac{1}{d \max_{u \neq v} k_{\mathcal{G}}(u, v)} \sum_{t=1}^{d} \sum_{u \neq v} \frac{h_u^{(\ell),t}}{\|h_u^{(\ell),t}\|} k_{\mathcal{G}}(u, v), \tag{27}$$

where $h_u^{(\ell),t}$ is the $t$-th feature of $d$-dimensional feature vector of node $u$ at layer $\ell$ and $k_{\mathcal{G}}(u, v)$ is the distance between two nodes $u$ and $v$, computed as a shortest path. Every unitary signal $h_u^{(\ell),t}/\|h_u^{(\ell),t}\|$ propagated across the graph $G$ from the source node $v$ is weighted by the normalized propagation distance $k_{\mathcal{G}}(u, v)/\max_{u \neq v} d_G(u, v)$ for all nodes $u \neq v$ and then averaged over entire $d$ output channels. To estimate the total effective resistance of the graph, 10 nodes are randomly sampled from each graph and total effective resistance of the graph is estimated for each source node. The final $h_{\odot}^{(\ell)}$ and total resistance of the graph are obtained by averaging across the 10 sampled nodes. The experiment is repeated for every graph in the dataset and the signal propagation measured for each graph is plotted against the normalized total effective resistance of the corresponding graph.

In Figs. 8 to 10, we report the results of this analysis.

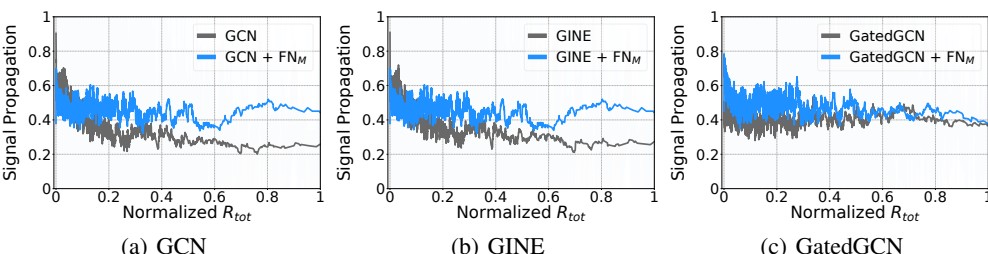

(a) GCN      (b) GINE      (c) GatedGCN

Figure 8: The amount of signal propagated across the graphs w.r.t. the normalized $R_{tot}$ in PEPTIDES-STRUCT.

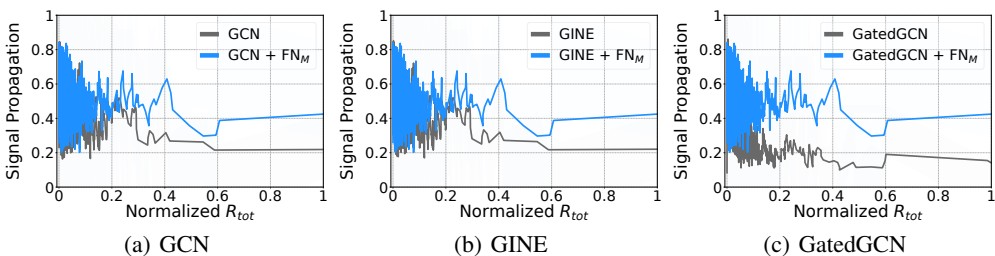

(a) GCN      (b) GINE      (c) GatedGCN

Figure 9: The amount of signal propagated across the graphs w.r.t. the normalized $R_{tot}$ in MOLHIV.

## G   DISTRIBUTION ANALYSIS OF SUBGRAPH SIZE RATIO

We analyze the distribution of subgraph size ratios produced by METIS partitioning across different numbers of partitions ($C$) and datasets.

In generally, as $C$ increases from 2 to 32, the average subgraph size ratio naturally decreases since each partition contains a smaller portion of the original graph. The width of the distributions generally increases with $C$, indicating more variance in partition sizes with finer granularity. Most datasets show roughly normal or slightly skewed distributions around the expected mean ratio of $1/C$.

As shwon in Fig. 11, PEPTIDE-FUNC/STRUCT show relatively tight, symmetric distributions. In indicates that METIS creates balanced partitions for molecular graphs. CIFAR10 and MNIST

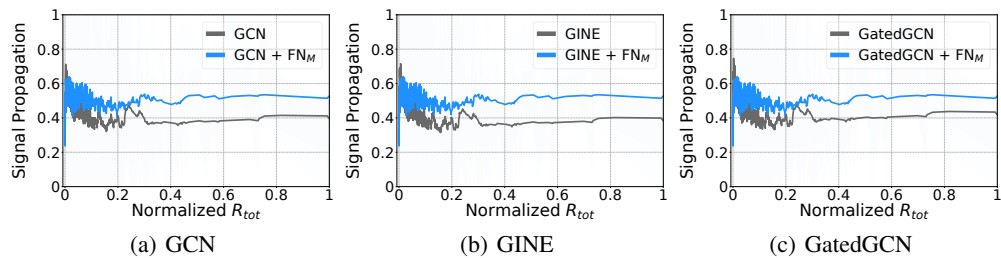

Figure 10: The amount of signal propagated across the graphs w.r.t. the normalized $R_{tot}$ in MOLTOX21.

show distinct bimodal patterns, especially at $C = 16$ and $C = 32$, likely due to the regular grid-like structure of superpixel graphs (See Fig. 12 and Fig. 13). As shown in Fig. 14 and Fig. 15, MOLHIV and MOLTOX21 show broader distributions, particularly at higher $C$ values, reflecting the more heterogeneous nature of these molecular graphs.

The consistent distributions for molecular datasets indicate METIS partitioning is well-suited for these graph types. The bimodal distributions in image-based graphs indicate the natural clustering of superpixels into regions of different sizes. Higher $C$ values (i.e., 16, 32) generally maintain reasonable balance while allowing for more fine-grained capture of graph structure.

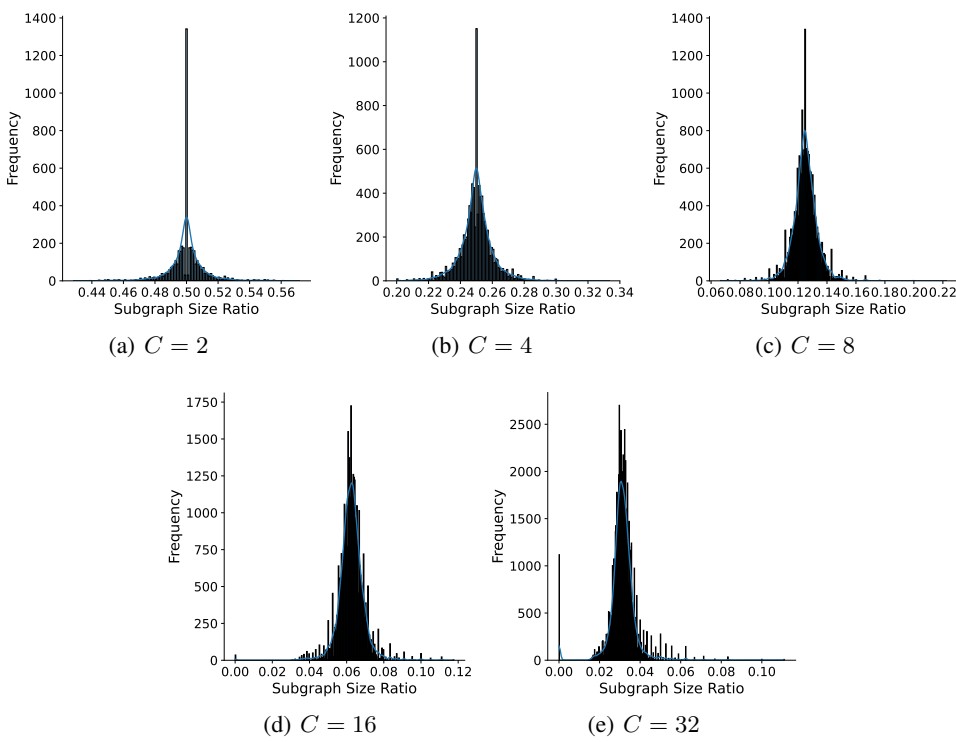

Figure 11: Similarity of node centrality distribution in PEPTIDE-FUNC/STRUCT.

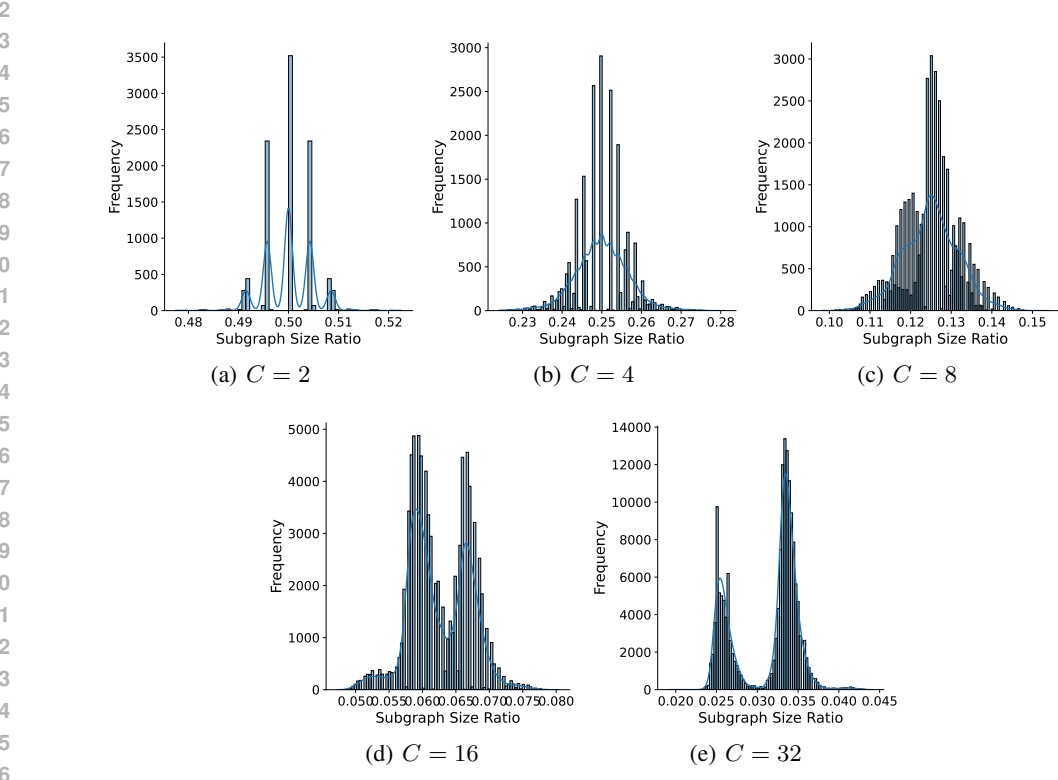

Figure 12: Similarity of node centrality distribution in CIFAR10.

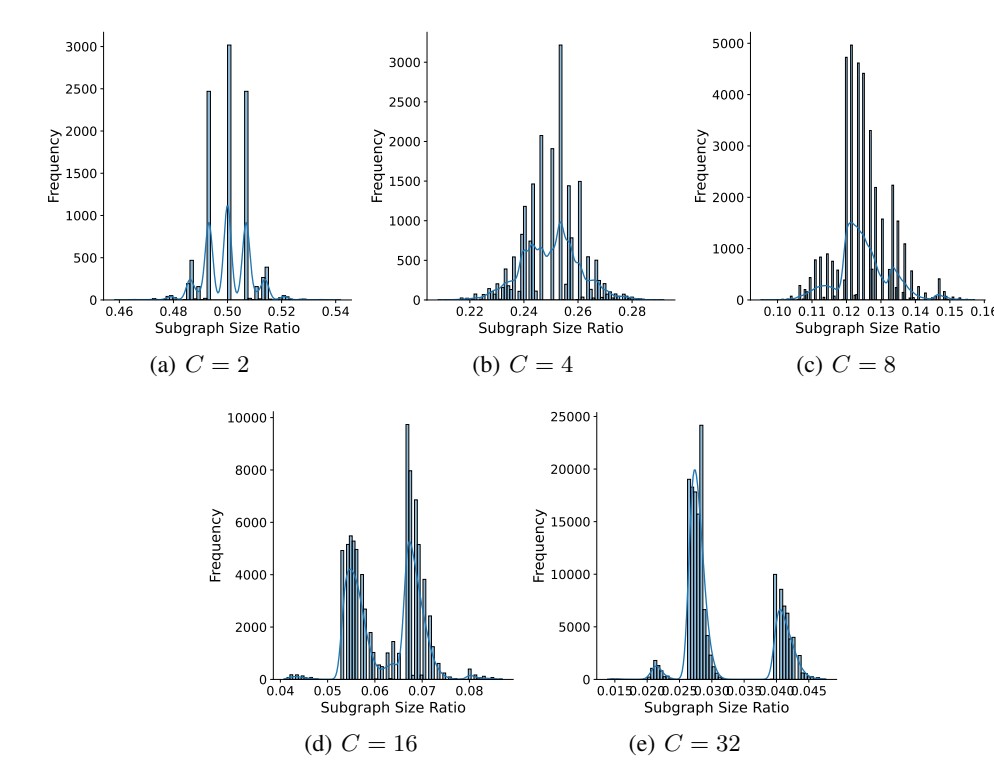

Figure 13: Similarity of node centrality distribution in MNIST.

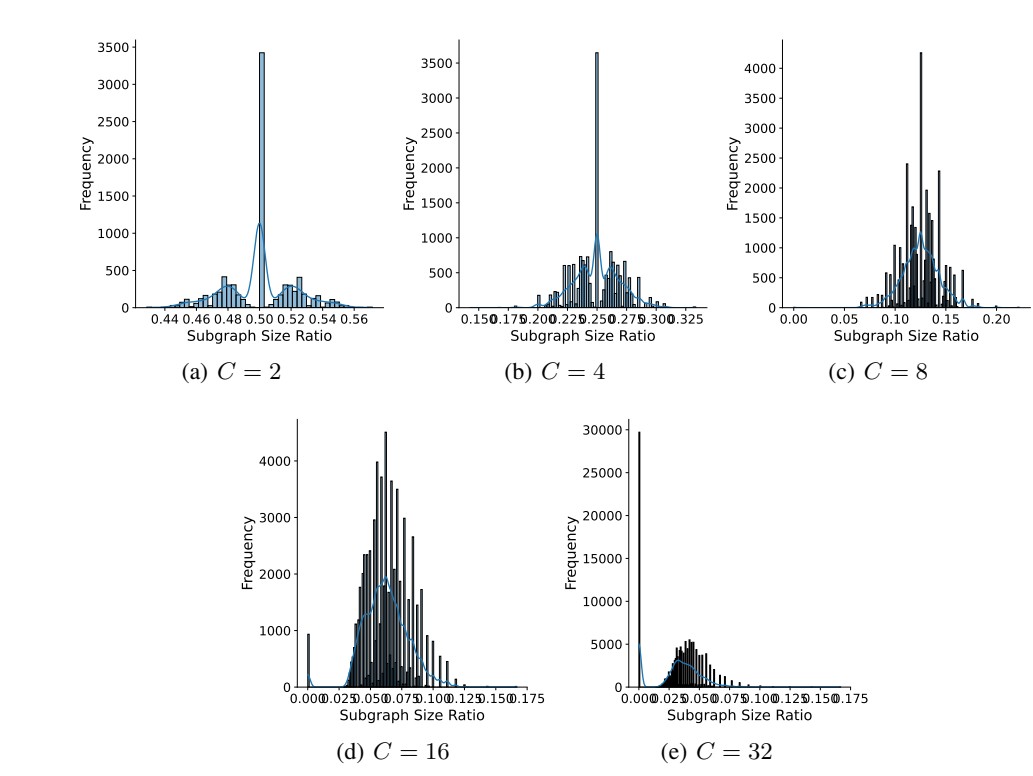

Figure 14: Similarity of node centrality distribution in MOLHIV.

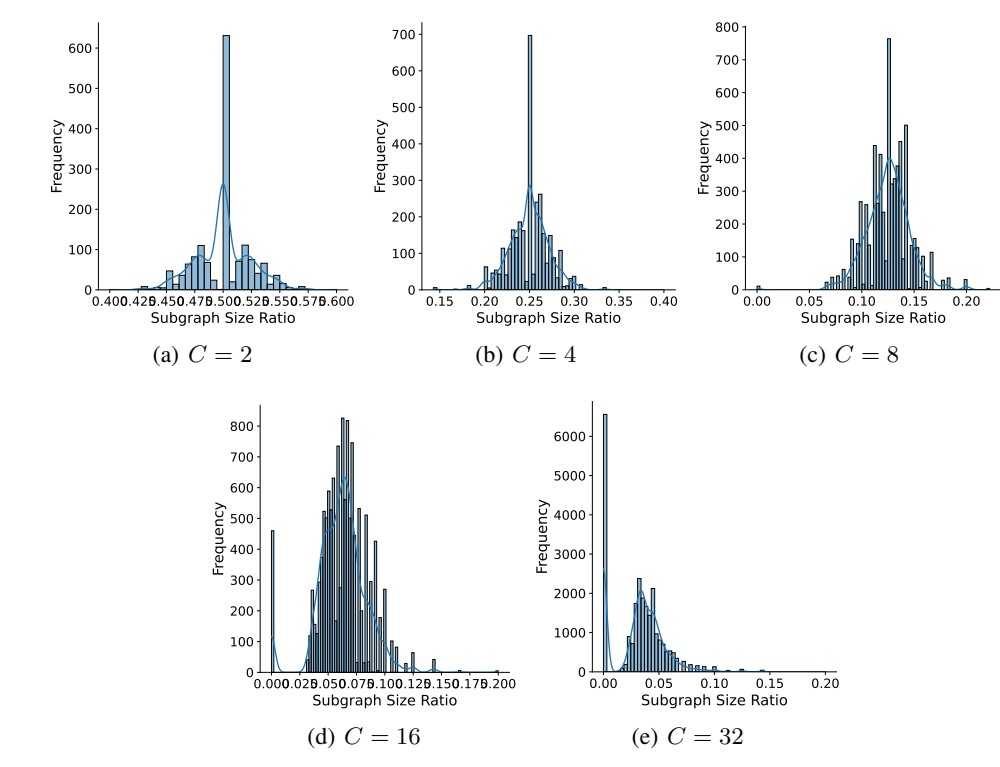

Figure 15: Similarity of node centrality distribution in MOLTOX21.

## H CONNECTION TO RENORMALIZATION TECHNIQUES

Our fractal nodes method draws inspiration from renormalization group techniques in physics, where complex systems are analyzed on different scales. While this connection is conceptual, the fundamental idea of scale transformation provides intuition for our approach. The renormalization involves replacing groups of interacting components with effective units. Similarly, our fractal nodes summarize subgraph information, though we maintain these summary units as fractal nodes alongside the original graph structure rather than replacing them.

From a complex network perspective, fractal nodes facilitate a transition from scale-free fractal networks to small-world networks. Similar to the renormalization techniques described by Wei et al. (2013), our $FN_M$ method introduces long-range interactions between the fractal nodes, giving small-world properties to the network (Albert & Barabási, 2002). We extend beyond renormalization in 3 aspects: (i) preserving the original structure while adding fractal nodes, (ii) enabling adaptive information flow through learned parameters, and (iii) maintaining exchange between local and global scales.

This architecture enables efficient information propagation through several mechanisms. The fractal nodes act as "shortcuts" in the network, reducing the effective distance information must traverse. Maintaining local and summarized representations enables simultaneous processing at multiple scales while preserving local network characteristics. This multi-scale processing capability addresses the over-squashing problem by facilitating efficient global information flow without sacrificing local structural information.

The key differences between our approach and classical renormalization highlight the factors we introduce specifically for graph learning tasks. While traditional renormalization uses fixed transformation rules in a unidirectional manner (fine to coarse), our method learns adaptive representations through trainable parameters and enables bidirectional information exchange. This creates a more flexible framework that captures complex relationships in graph-structured data while maintaining computational efficiency.

## I DIFFERENT PARTITIONING ALGORITHMS

To verify the effectiveness of partitioning other than METIS partitioning, we conduct experiments applying FN and $FN_M$ to GINE on PEPTIDES-FUNC, PEPTIDES-STRUCT, MOLHIV, and MOLTOX21 datasets using random partitioning and Louvain (Blondel et al., 2008) and Girvan-Newman (Girvan & Newman, 2002) partitioning.

In Table 12, our results provide a comprehensive comparison of different graph partitioning methods for GINE with FN and $FN_M$ architectures on multiple molecular and peptide datasets. METIS consistently shows superior or competitive performance on all datasets. It achives the best results in most cases, such as 0.6815 AP on PEPTIDES-FUNC with GINE+FN and 0.7018 AP with GINE+$FN_M$. While random partitioning shows surprisingly competitive performance, particularly on MOLHIV where it achieves 0.8039 ROCAUC with GINE+FN, community detection algorithms such as Louvain and Girvan-Newman generally underperform compared to METIS and random partitioning. The performance gap between different partitioning methods becomes more pronounced when using $FN_M$ compared to FN. METIS shows more stable performance with lower standard deviations across all metrics. For molecular property prediction tasks, the choice of partitioning method appears less critical. However, on PEPTIDES-FUNC and PEPTIDES-STRUCT, METIS shows clear advantages with consistently lower MAE scores. These findings validate our choice of METIS as the default partitioning algorithm while suggesting that the optimal partitioning strategy may depend on the specific graph structure and task requirements.

In Table 13, the analysis of results on ogbn-arxiv provides additional insights into partitioning methods on large scale graph datasets. The performance differences between partitioning methods are relatively small, with scores ranging between 72.46% and 73.03% accuracy. For GCN+FN, METIS achieves the best performance at 73.03%, while random partitioning performs best for GCN+$FN_M$ at 73.01%. GraphSAGE shows slightly lower performance compared to GCN across all partitioning strategies, with Louvain partitioning achieving the best results at 72.76% for GraphSAGE+FN. Interestingly, the Girvan-Newman algorithm consistently times out on this dataset, indicating scalability issues

with larger graphs such as ogbn-arxiv. The standard deviations are generally smaller for GraphSAGE compared to GCN, suggesting more stable performance across different random seeds. These results further support that METIS remains competitive.

Table 12: Comparison of different graph partitioning methods for GINE with FN and $FN_M$ architectures on PEPTIDES-FUNC/STRUCT and molecular property prediction tasks. Best results for each metric are shown in **bold**.

| Method | Partitioning | PEPTIDES-FUNC | PEPTIDES-STRUCT | MOLHIV | MOLTOX21 |
|---|---|---|---|---|---|
| | | AP ↑ | MAE ↓ | ROCAUC ↑ | ROCAUC ↑ |
| GINE + FN | METIS | $\mathbf{0.6815}_{\pm 0.0059}$ | $\mathbf{0.2515}_{\pm 0.0020}$ | $0.7882_{\pm 0.0050}$ | $\mathbf{0.7751}_{\pm 0.0029}$ |
| | Random | $0.6533_{\pm 0.0103}$ | $0.2688_{\pm 0.0014}$ | $\mathbf{0.8039}_{\pm 0.0078}$ | $0.7653_{\pm 0.0065}$ |
| | Louvain | $0.6044_{\pm 0.0068}$ | $0.2799_{\pm 0.0015}$ | $0.7844_{\pm 0.0050}$ | $0.7701_{\pm 0.0026}$ |
| | Girvan-Newman | $0.6528_{\pm 0.0051}$ | $0.2628_{\pm 0.0045}$ | $0.7837_{\pm 0.0078}$ | $0.7630_{\pm 0.0060}$ |
| GINE + $FN_M$ | METIS | $\mathbf{0.7018}_{\pm 0.0074}$ | $\mathbf{0.2446}_{\pm 0.0018}$ | $\mathbf{0.8127}_{\pm 0.0076}$ | $\mathbf{0.7926}_{\pm 0.0021}$ |
| | Random | $0.6680_{\pm 0.0066}$ | $0.2538_{\pm 0.0013}$ | $0.8090_{\pm 0.0061}$ | $0.7867_{\pm 0.0045}$ |
| | Louvain | $0.6164_{\pm 0.0120}$ | $0.2789_{\pm 0.0022}$ | $0.7629_{\pm 0.0164}$ | $0.7510_{\pm 0.0118}$ |
| | Girvan-Newman | $0.6514_{\pm 0.0064}$ | $0.2655_{\pm 0.0037}$ | $0.7763_{\pm 0.0174}$ | $0.7579_{\pm 0.0097}$ |

Table 13: Comparison of different graph partitioning methods for GCN/GraphSAGE with FN and $FN_M$ on ogbn-arxiv dataset. Results show accuracy (%) and best results for each metric are shown in **bold**.

| ogbn-arxiv | GCN + FN | GCN + $FN_M$ | GraphSAGE + FN | GraphSAGE + $FN_M$ |
|---|---|---|---|---|
| METIS | $\mathbf{73.03}_{\pm 0.37}$ | $72.93_{\pm 0.35}$ | $72.70_{\pm 0.11}$ | $72.54_{\pm 0.30}$ |
| Random | $72.79_{\pm 0.37}$ | $\mathbf{73.01}_{\pm 0.41}$ | $72.46_{\pm 0.20}$ | $72.46_{\pm 0.27}$ |
| Louvain | $72.73_{\pm 0.57}$ | $72.95_{\pm 0.26}$ | $\mathbf{72.76}_{\pm 0.15}$ | $\mathbf{72.56}_{\pm 0.58}$ |
| Girvan-Newman | Time-out | Time-out | Time-out | Time-out |

Table 14 demonstrates the empirical runtime performance of different graph partitioning algorithms across various graph-level tasks, providing evidence for the practicality of our approach. While all algorithms show comparable performance on smaller datasets like Peptides (with runtimes in microseconds), noticeable differences emerge starting with medium-sized datasets like MNIST.

The distinction becomes particularly pronounced on large-scale datasets like ogbn-arxiv. We opt for METIS as our default partitioning algorithm due to its theoretical time complexity of $O(|E|)$ and superior empirical performance. METIS efficiently partitions large graphs such as ogbn-arxiv in under 9 seconds, and even handles massive graphs like ogbn-products around 15 minutes.

In contrast, the Louvain algorithm requires over 50 seconds for ogbn-arxiv, while the Girvan-Newman algorithm encounters runtime limitations, making it impractical for large-scale graphs like ogbn-arxiv and ogbn-products. These results validate our choice of METIS as the primary partitioning algorithm, as it provides an effective balance between computational efficiency and partition quality across different graph scales.

Table 14: Empirical runtime of partitioning algorithms.

| Algorithm | PEPTIDES-FUNC/STRUCT | MNIST | MOLHIV | ogbn-arxiv | ogbn-product |
|---|---|---|---|---|---|
| METIS | 0.71 $\mu$s | 0.36 s | 0.71$\mu$s | 8.57 s | 923.27 s |
| Louvain | 1.19 $\mu$s | 0.36 s | 1.19$\mu$s | 52.12s | 119 m |
| Girvan-Newman | 1.19 $\mu$s | 0.36 s | 0.72$\mu$s | Time-out | Time-out |

# J    SCALABILITY ANALYSIS OF OF FRACTAL NODE

## J.1    PROFILING RESULTS ON SYNTHETIC GRAPHS

To evaluate the efficiency and scalability of our FN integrated GCN model, we conducted experiments on synthetic Erdos-Renyi (Erdos et al., 1960) graphs with node counts ranging from 1,000 to 100,000. The edge probability in the Erdos-Renyi network is set to achieve an average node degree of approximately 5, with the node feature dimension fixed at 100.

Fig. 16(a) represents that the GPU memory usage of GCN+FN increases linearly with the graph size and validates its linear space complexity. Fig. 16(b) shows the training time for both GPU and CPU implementations. The GPU training time exhibits a sub-linear growth trend as the graph size increases. This means the ability of fractal nodes to effectively use GPU parallelism for large-scale graph computations. In contrast, the CPU training time grows linearly with the graph size and indicates the sequential nature of CPU computations and its limitations in handling large-scale parallel graph operations.

The results demonstrate that the GPU device (RTX A6000 used in our experiments) efficiently handles the computational workload on varying graph sizes. These observations validate the scalability and practicality of our proposed GCN+FN model, particularly for large-scale graph learning tasks where both memory efficiency and computational speed are critical.

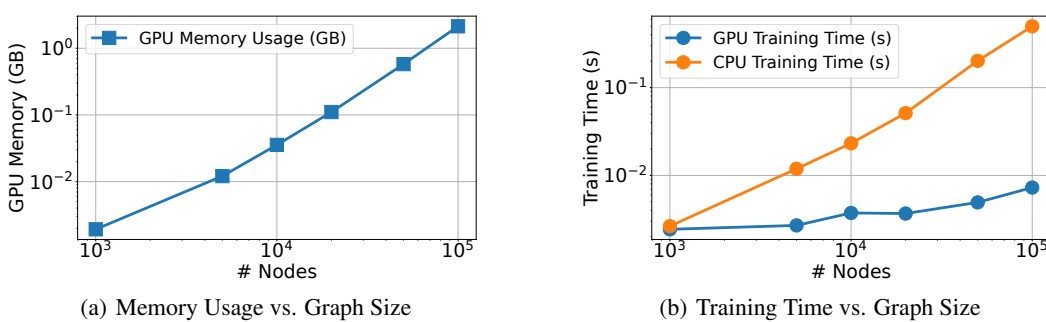

(a) Memory Usage vs. Graph Size        (b) Training Time vs. Graph Size

Figure 16: GPU memory usage and training time of GCN+FN on synthetic graphs.

## J.2    PROFILING RESULTS ON LARGE-SCALE REAL-WORLD GRAPHS

Table 15 shows the profiling results of various models in terms of training time per epoch and memory usage on the large-scale ogbn-arxiv dataset. Note that we perform full batch training for a fair comparison of computational requirements.

The results show that our fractal node approach maintains efficiency. When integrated with base MPNNs, fractal nodes introduce trivial computational overhead — GCN+FN maintains identical training time (1.27s) and memory usage (16.49GB) compared to the vanilla GCN. Similarly, GraphSAGE+FN shows only a marginal increase in computational cost (0.57s vs 0.55s) while preserving the same memory efficiency (7.74GB). Our method uses common MPNN operations without introducing complex additional computations.

In contrast, graph Transformers (e.g., GraphGPS, Exphormer) require substantially more computational resources (38.91GB and 34.04GB memory, respectively) due to their attention mechanisms. This empirical evidence indicates that our fractal node approach achieves a favorable balance between model accuracy and computational efficiency in practice.

# K    LARGE-SCALE NODE CLASSIFICATION

**Large-scale graphs.**    We consider a collection of large graphs released recently by the Open Graph Benchmark (OGB) (Hu et al., 2021): ogbn-arxiv and ogbn-products with node numbers 0.16M and 2.4M, respectively. We maintain all the OGB standard evaluation settings.

Table 15: Training time and GPU memory usage on large graphs

| Model | ogbn-arxiv | |
|---|---|---|
| | Train/Epoch (s) | Mem. (GB) |
| GCN | 1.27 | 16.49 |
| GraphSAGE | 0.55 | 7.74 |
| GraphGPS | 1.32 | 38.91 |
| Exphormer | 0.74 | 34.04 |
| NodeFormer | 1.20 | 16.30 |
| DiffFormer | 0.77 | 24.51 |
| PolyNormer | 0.31 | 16.09 |
| GCN + FN | 1.27 | 16.49 |
| GCN + FN$_M$ | 1.27 | 16.49 |
| GraphSAGE + FN | 0.57 | 7.74 |
| GraphSAGE + FN$_M$ | 0.58 | 7.76 |

**Baselines.** Our main focus lies on classic MPNNs: GCN (Kipf & Welling, 2017), and GraphSAGE (Hamilton et al., 2017); the state-of-the-art scalable graph Transformers: GraphGPS (Rampášek et al., 2022), NAGphormer (Chen et al., 2023), Exphormer (Shirzad et al., 2023), NodeFormer (Wu et al., 2022), DiffFormer (Wu et al., 2023a), PolyNormer (Zakar-Polyák et al., 2023), and SGFormer (Wu et al., 2023b); hierarchical methods: HC-GNN (Zhong et al., 2023), ANS-GT (Cai et al., 2021), and HSGT (Zhu et al., 2023); MLP-based method: LINKX (Lim et al., 2021).

**Setting.** We conduct hyperparameter tuning on classic MPNNs, which is consistent with the hyperparameter search space of Deng et al. (2024). Specifically, we use the Adam optimizer with a learning rate from $\{0.001, 0.005, 0.01\}$ and an epoch limit of 2500. We tune the hidden dimension from $\{64, 256, 512\}$. We consider whether to use batch or layer normalization, residual connections, and dropout rates from $\{0.2, 0.3, 0.5, 0.7\}$, the number of layers from $\{1, 2, 3, 4, 5, 6, 7, 8, 9, 10\}$, and $C$ from $\{32, 64, 128\}$.

**Implementation.** While our main experiment focuses on graph-level tasks, our fractal node method can be naturally extended to node classification tasks. The key distinction lies in how we use the processed fractal node representations from the MLP-Mixer layer to make node-level predictions rather than graph-level ones.

For graph-level tasks, as shown in Equation (8), the fractal nodes are mixed through the MLP-Mixer to produce

$$\tilde{F} = \mathsf{MLPMixer}(F^{(L)}), \quad F^{(L)} = [f_1^{(L)}, f_2^{(L)}, ..., f_C^{(L)}]. \tag{28}$$

These mixed representations are then used directly for graph-level prediction via global pooling.

For node classification, however, we need to propagate this mixed global information back to individual nodes. After the MLP-Mixer processes the $C$ fractal nodes according to Equations (9) and (10), we obtain $\tilde{F}^{(L)} \in \mathbb{R}^{C \times d}$. These processed fractal node representations need to be aligned with all nodes in their respective subgraphs.

Let $\mathcal{V}_c$ be the set of nodes in subgraph $c$. For each node $v \in \mathcal{V}_c$, we update its final representation by combining its current features with the processed fractal node information from its corresponding subgraph:

$$h_v^{(\text{final})} = h_v^{(L)} + \tilde{f}_c^{(L)}, \quad \forall v \in \mathcal{V}_c, \tag{29}$$

where $\tilde{f}_c^{(L)}$ is the $c$-th row of $\tilde{F}^{(L)}$ corresponding to the fractal node of subgraph $c$. This operation ensures that each node receives the processed global context from its subgraph's fractal node and maintains consistency with our method while adapting it for node-level predictions.

In implementation, this process can be efficiently vectorized using a batch membership index that maps each node to its corresponding fractal node representation. This adaptation allows our fractal

node framework to effectively handle both graph-level and node-level tasks while maintaining its computational efficiency and theoretical properties.

**Result.** As shown in Table 16, our experiments on these large-scale benchmarks demonstrate the effectiveness of our fractal node approach. On ogbn-arxiv, GCN+FN achieves 73.03% accuracy, showing substantial improvement over the base GCN (71.74%) and outperforming state-of-the-art graph Transformer models such as Exphormer and GraphGPS. The consistency between GCN+FN and GCN+FN$_M$ indicates the robustness of our approach. The performance gains are even more pronounced on the larger ogbn-products dataset, where GraphSAGE+FN$_M$ demonstrates substantial improvement, achieving state-of-the-art performance of 83.11% accuracy compared to the base GraphSAGE's 78.29%. This surpasses recent advanced models like PolyNormer and other graph Transformer architectures.

A notable advantage of our method becomes apparent when considering scalability. Several Transformer-based models (marked as OOM — Out of Memory in Table 16) fail to scale to ogbn-products due to their quadratic complexity in attention computation. In contrast, our method maintains computational efficiency while achieving superior performance (see Table 15). This highlights not only the effectiveness of fractal nodes in capturing both local and global graph information but also their practical applicability to large-scale graphs.

Table 16: Node classification results on large-scale graphs (%).

| Model | ogbn-arxiv | ogbn-product |
|---|---|---|
| # nodes | 169,343 | 2,449,029 |
| # edges | 1,166,243 | 61,859,140 |
| LINKX | $66.18_{\pm0.33}$ | $71.59_{\pm0.71}$ |
| GraphGPS | $70.97_{\pm0.41}$ | OOM |
| NAGphormer | $70.13_{\pm0.55}$ | $73.55_{\pm0.21}$ |
| Exphormer | $72.44_{\pm0.28}$ | OOM |
| NodeFormer | $69.86_{\pm0.25}$ | $72.93_{\pm0.13}$ |
| DiffFormer | $72.41_{\pm0.40}$ | $74.16_{\pm0.31}$ |
| PolyNormer | $71.82_{\pm0.23}$ | $\textbf{82.97}_{\pm0.28}$ |
| SGFormer | $72.63_{\pm0.13}$ | $74.16_{\pm0.31}$ |
| HC-GNN | $72.79_{\pm0.25}$ | - |
| ANS-GT | $72.34_{\pm0.50}$ | $80.64_{\pm0.29}$ |
| HSGT | $72.58_{\pm0.31}$ | $81.15_{\pm0.13}$ |
| GCN | $71.74_{\pm0.29}$ | $75.64_{\pm0.21}$ |
| GCN + FN | $\textbf{73.03}_{\pm0.37}$ | $81.29_{\pm0.21}$ |
| GCN + FN$_M$ | $\textbf{72.93}_{\pm0.35}$ | $81.33_{\pm0.33}$ |
| GraphSAGE | $71.49_{\pm0.27}$ | $78.29_{\pm0.16}$ |
| GraphSAGE + FN | $\textbf{72.70}_{\pm0.11}$ | $\textbf{83.07}_{\pm0.35}$ |
| GraphSAGE + FN$_M$ | $72.54_{\pm0.30}$ | $\textbf{83.11}_{\pm0.07}$ |

## L  THEORETICAL ANALYSIS

In this section, we provide theoretical analysis of fractal nodes to show how they mitigate oversquashing. Our analysis builds on effective resistance theory to characterize information flow in networks with fractal nodes.

**Preliminaries on effective resistance.**  Following Black et al. (2023) and Appendix F, we recap the effective resistance in graphs. For a connected, non-bipartite graph, the pseudoinverse of the normalized Laplacian can be expressed as:

$$\hat{\mathbf{L}}^+ = \sum_{j=0}^{\infty} \hat{\mathbf{A}}^j, \tag{30}$$

Furthermore, the effective resistance between nodes $u$ and $v$ can be written as:

$$R_{u,v} = \sum_{i=0}^{\infty} \left( \frac{1}{d_u}(\hat{\mathbf{A}}^i)_{uu} + \frac{1}{d_v}(\hat{\mathbf{A}}^i)_{vv} - \frac{2}{\sqrt{\deg_u \deg_v}}(\hat{\mathbf{A}}^i)_{uv} \right), \tag{31}$$

where $(\hat{\mathbf{A}}^i)_{u,v}$ represents the number of paths of length $i$ between nodes $u$ and $v$ (Black et al., 2023). This equation intuitively shows that more shorter and disjoint paths connecting two nodes leads to lower effective resistance.

### L.1  EFFECTIVE RESISTANCE WITH FRACTAL NODES

**Lemma L.1** (Fractal Node Effective Resistance). *Let $\mathcal{G}$ be a connected graph with $\mathcal{C}$ subgraphs and their associated fractal nodes. The effective resistance between any two nodes $u$, $v$ with fractal nodes can be expressed as:*

$$R_f(u,v) = (1_u - 1_v)^T \mathbf{L}_f^+ (1_u - 1_v), \tag{32}$$

*where $\mathbf{L}_f$ is the augmented Laplacian incorporating fractal node connections:*

$$\mathbf{L}_f = \begin{bmatrix} \mathbf{L} + \sum_{i=1}^{\mathcal{C}} \mathbf{F}_i \mathbf{F}_i^T & -[\mathbf{F}_1, \mathbf{F}_2, ..., \mathbf{F}_{\mathcal{C}}] \\ -[\mathbf{F}1, \mathbf{F}2, ..., \mathbf{F}_{\mathcal{C}}]^T & \mathbf{I}_{\mathcal{C}} \end{bmatrix}, \tag{33}$$

*where $\mathbf{L}$ is the original Laplacian matrix, $\mathbf{F}_i$ is the incidence vector for fractal node $i$ indicating its connections to the original nodes.*

Similar to the path-based interpretation in Black et al. (2023), we can express $R_f(u,v)$ in terms of paths:

$$R_f(u,v) = \sum_{i=0}^{\infty} \left( \frac{1}{\deg_u}(\hat{\mathbf{A}}_f^i)_{uu} + \frac{1}{\deg_v}(\hat{\mathbf{A}}_f^i)_{vv} - \frac{2}{\sqrt{\deg_u \deg_v}}(\hat{\mathbf{A}}_f^i)_{uv} \right) \tag{34}$$

where $\hat{\mathbf{A}}_f$ is the normalized adjacency matrix including fractal node connections.

### L.2  PROOF OF THEOREM 4.1

**Theorem 4.1** (Resistance reduction). *Let $\mathcal{G}$ be the original graph and $\mathcal{G}_f$ be the augmented graph with fractal nodes. For any nodes $u, v \in \mathcal{G}$, the effective resistance in $\mathcal{G}_f$ satisfies:*

$$R_f(u,v) \leq R(u,v), \tag{35}$$

*where $R_f(u,v)$ is the effective resistance in $\mathcal{G}_f$ and $R(u,v)$ is the original effective resistance in $\mathcal{G}$.*

*Proof.* Let $\mathcal{G} = (\mathcal{V}, \mathcal{E})$ be the original graph and $\mathcal{G}_f = (\mathcal{V} \cup \mathcal{F}, \mathcal{E} \cup \mathcal{E}_f)$ be the augmented graph with fractal nodes, where $\mathcal{F}$ is the set of fractal nodes and $\mathcal{E}_f$ is the set of edges connecting nodes to fractal nodes.

Following Black et al. (2023), we express the effective resistance in terms of path decomposition:

$$R_f(u,v) = \sum_{i=0}^{\infty} \left( \frac{1}{\deg_u}(\hat{\mathbf{A}}_f^i)_{uu} + \frac{1}{\deg_v}(\hat{\mathbf{A}}_f^i)_{vv} - \frac{2}{\sqrt{\deg_u \deg_v}}(\hat{\mathbf{A}}_f^i)_{uv} \right), \tag{36}$$

where $\hat{\mathbf{A}}_f$ is the normalized adjacency matrix of $\mathcal{G}_f$.

Let $\mathcal{P}_{uv}$ be the set of all paths connecting $u$ and $v$ in $\mathcal{G}_f$. The effective resistance can be expressed as:

$$R_f(u,v) = \min_{p \in \mathcal{P}_{uv}} \sum_{(x,y) \in p} r_{xy}, \tag{37}$$

where $r_{xy}$ is the resistance of edge $(x,y)$.

By Rayleigh's monotonicity principle (Black et al., 2023), since $\mathcal{G}_f$ contains all edges of $\mathcal{G}$ plus additional edges through fractal nodes, adding these edges can only decrease the effective resistance between any pair of nodes. Therefore:

$$R_f(u,v) \leq R(u,v). \tag{38}$$

$\square$

### L.3    PROOF OF THEOREM 4.2

**Theorem 4.2** (Signal propagation with fractal nodes). *For a MPNN with fractal nodes, the signal propagation between nodes $u, v$ after $\ell$ layers satisfies:*

$$\|h_u^{(\ell)} - h_v^{(\ell)}\| \leq \exp(-\ell/R_f(u,v))\|h_u^{(0)} - h_v^{(0)}\|, \tag{39}$$

*where $R_f(u,v)$ is the effective resistance in the augmented graph with fractal nodes.*

*Proof.* First, the message passing process in MPNN (i.e., GCN) with fractal nodes can be expressed as:

$$h_v^{(\ell+1)} = \sigma \left( W h_v^{(\ell)} + \sum_{u \in \mathcal{N}(v)} \frac{1}{\sqrt{\deg_v \deg_u}} W h_u^{(\ell)} + W_f h_f^{(\ell)} \right), \tag{40}$$

where $h_f^{(\ell)}$ is the fractal node representation. To analyze the signal propagation, we consider the continuous-time analog by removing the nonlinearity $\sigma$:

$$\frac{d}{dt} h_v(t) = -\mathbf{L}_f h_v(t), \tag{41}$$

The solution to this differential equation is:

$$h_v(t) = \exp(-t\mathbf{L}_f) h_v(0), \tag{42}$$

The signal difference between two nodes $u, v$ is bounded as follows:

$$||h_u(t) - h_v(t)|| = ||(\exp(-t\mathbf{L}_f))(h_u(0) - h_v(0))|| \tag{43}$$

$$\leq ||exp(-t\mathbf{L}_f)|| \cdot ||h_u(0) - h_v(0)|| \tag{44}$$

$$\leq \exp(-t/R_f(u,v))||h_u(0) - h_v(0)|| \tag{45}$$

The last inequality comes from the spectral bound related to the effective resistance $R_f(u,v)$ in the graph augmented with fractal nodes. Mapping back to the discrete layer steps by setting $t = \ell$, we obtain our desired bound:

$$||h_u^{(\ell)} - h_v^{(\ell)}|| \leq \exp(-\ell/R_f(u,v))||h_u^{(0)} - h_v^{(0)}||, \tag{46}$$

This provides the worst-case signal propagation bound in the graph with fractal nodes. By the previously proven Theorem 4.1, we know that $R_f(u,v) \leq R(u,v)$, thus fractal nodes provide better signal propagation guarantees than the original graph. $\square$

**Corollary L.2** (Improved signal propagation). *Since $R_f(u,v) \leq R(u,v)$ by the Resistance Reduction theorem, fractal nodes improve the worst-case signal propagation bound compared to the original graph:*

$$\exp(-\ell/R_f(u,v)) \leq \exp(-\ell/R(u,v)). \tag{47}$$

### L.4 TOTAL RESISTANCE ANALYSIS

**Theorem L.3** (Total Resistance with Fractal Nodes). *Let $\mathcal{G}_f$ be the graph augmented with $C$ fractal nodes. The total effective resistance satisfies:*

$$R_{tot}^f = n \cdot tr(\mathbf{L}_f^+) = n \cdot \sum_{i=2}^{n+C} \frac{1}{\sigma_i}, \tag{48}$$

*where $\mathbf{L}_f$ is the augmented Laplacian and $\sigma_i$ are its eigenvalues.*

*Proof.* The total resistance can be expressed through the trace of the pseudoinverse of the Laplacian matrix $\mathbf{L}_f$. By construction, $\mathbf{L}_f$ has dimension $(n+C) \times (n+C)$ and its eigendecomposition yields $n + C$ eigenvalues. The pseudoinverse $\mathbf{L}_f^+$ has the same eigenvectors as $\mathbf{L}_f$ with reciprocal non-zero eigenvalues, giving us the stated formula. The factor $n$ appears because we sum over all pairs of the $n$ original nodes. □

**Corollary L.4** (Impact of Fractal Node Count). *For a graph $\mathcal{G}$ augmented with $C$ fractal nodes, the total resistance decreases with $C$ as $R_{tot}^f = n \cdot \sum_{i=2}^{n+C} \frac{1}{\sigma_i}$, where additional eigenvalues from larger $C$ decrease the sum. This leads to improved signal propagation bounds $||h_u^{(\ell)} - h_v^{(\ell)}|| \leq \exp(-\ell / R_f(u, v))$.*

# M   DETAILED DISCUSSION ON SECTION 5.2

To thoroughly analyze the role of positional encodings (PEs) and fractal nodes in model expressivity, we conducted extensive ablation studies analyzing different combinations of structural components. Table 17 shows results across three synthetic datasets (CSL, SR25, EXP) designed to test model expressiveness.

Our ablation study reveals several important insights about the interplay between positional encodings and our method. Without PEs, base MPNNs (GCN, GINE, GatedGCN) consistently show limited expressiveness across all datasets, achieving only 10.00% on CSL, 6.67% on SR25, and approximately 51-52% on EXP. Adding PEs substantially improves base model performance, as evidenced by GCN's significant improvement from 10.00% to 76.17% on CSL and from 52.17% to 100% on EXP.

Notably, even without any positional encodings, our fractal node variants demonstrate significantly enhanced expressivity. GINE+$FN_M$ achieves 47.33% on CSL and 95.58% on EXP without any PE, while GatedGCN+$FN_M$ reaches 49.67% on CSL. All $FN_M$ variants achieve 100% on SR25 regardless of PE configuration, and this indicates that our method provides inherent structural awareness independent of positional encodings.

Table 17: Synthetic results (Accuracy ↑). The gray shaded rows are the results without using PE, and are the fairest to compare against.

| Method | Ablation | | Dataset | | |
|---|---|---|---|---|---|
| | PE (Original Graph) | PE (Coarsened Graph) | CSL | SR25 | EXP |
| GCN | ✗ | N/A | 10.00 | 6.67 | 52.17 |
| GCN | ✓ | N/A | 76.17 | 100.0 | 100.0 |
| GINE | ✗ | N/A | 10.00 | 6.67 | 51.35 |
| GINE | ✓ | N/A | 100.0 | 100.0 | 100.0 |
| GatedGCN | ✗ | N/A | 10.00 | 6.67 | 51.25 |
| GatedGCN | ✓ | N/A | 100.0 | 100.0 | 100.0 |
| GCN + $FN_M$ | ✗ | ✗ | 39.67 | 100.0 | 86.40 |
| GCN + $FN_M$ | ✗ | ✓ | 76.17 | 100.0 | 100.0 |
| GCN + $FN_M$ | ✓ | ✗ | 100.0 | 100.0 | 100.0 |
| GCN + $FN_M$ | ✓ | ✓ | 100.0 | 100.0 | 100.0 |
| GINE + $FN_M$ | ✗ | ✗ | 47.33 | 100.0 | 95.58 |
| GINE + $FN_M$ | ✗ | ✓ | 84.83 | 100.0 | 100.0 |
| GINE + $FN_M$ | ✓ | ✗ | 100.0 | 100.0 | 100.0 |
| GINE + $FN_M$ | ✓ | ✓ | 100.0 | 100.0 | 100.0 |
| GatedGCN + $FN_M$ | ✗ | ✗ | 49.67 | 100.0 | 96.50 |
| GatedGCN + $FN_M$ | ✗ | ✓ | 81.83 | 100.0 | 100.0 |
| GatedGCN + $FN_M$ | ✓ | ✗ | 100.0 | 100.0 | 100.0 |
| GatedGCN + $FN_M$ | ✓ | ✓ | 100.0 | 100.0 | 100.0 |

