# OpenReview forum: "Fractal-Inspired Message Passing Neural Networks with Fractal Nodes"
_ICLR.cc/2025/Conference — Submitted to ICLR 2025_

### Official Review · Reviewer_HGoY · 2024-10-30

**Soundness:** 3
**Presentation:** 3
**Contribution:** 3
**Rating:** 6
**Confidence:** 2

**Summary:**

This paper introduces a novel concept called "fractal nodes" to enhance Message Passing Neural Networks (MPNNs) by addressing their limitations in balancing local and global information processing. The approach is inspired by the fractal nature of real-world networks and renormalization techniques. The key innovation lies in creating fractal nodes that coexist with original nodes, where each fractal node summarizes subgraph information and integrates into the MPNN architecture. The method aims to improve long-range dependencies while maintaining MPNN's computational efficiency, presenting an alternative to graph Transformers.

**Strengths:**

1. Originality
- The concept of fractal nodes is novel and well-motivated by real-world network properties
- The approach offers a fresh perspective on handling the local-global information trade-off
- Creative adaptation of renormalization concepts into neural network architecture

2. Quality
- Strong theoretical foundation drawing from established concepts in network science
- Clear connection between fractal properties and the proposed solution
- Comprehensive experimental validation including performance comparisons and ablation studies
- Addresses known MPNN limitations (over-squashing) with a principled approach

3. Clarity
- Well-structured presentation with clear motivation and problem statement
- Effective use of figures to illustrate concepts (especially Figure 1)
- Logical flow from theoretical inspiration to practical implementation

4. Significance
- Provides an efficient alternative to computationally expensive graph Transformers
- Maintains MPNN's computational advantages while improving its capabilities
- Potentially applicable across various graph-based learning tasks

**Weaknesses:**

Weaknesses

1. Theoretical Analysis
- The paper could benefit from a more rigorous theoretical analysis of why fractal nodes help with over-squashing
- Limited discussion on the optimal number or size of fractal nodes

2. Experimental Validation
- The truncated content doesn't show complete experimental results
- Need for more extensive comparisons with other approaches addressing similar limitations
- Absence of ablation studies on the impact of different subgraph partitioning strategies

3. Practical Considerations
- Limited discussion on the computational overhead of creating and maintaining fractal nodes
- Unclear scalability analysis for very large graphs
- No discussion on potential limitations or failure cases

**Questions:**

How sensitive is the method to different subgraph partitioning strategies?
What criteria should be used to determine optimal partitioning?

---

> ### Author Response · Authors · 2024-11-22
>
> We sincerely thank Reviewer `HgOY` for the positive review and valuable feedback, highlighting our strengths in
>
> 1. Introduces novel fractal nodes concept
> 2. Strong theoretical foundation with robust experiments
> 3. Clear technical implementation details
> 4. Maintains efficiency while matching graph Transformer performance
>
> Below, we would like to address each of your questions and concerns:
>
> ---
>
> ***(W1: Theoretical Analysis) The paper could benefit from a more rigorous theoretical analysis of why fractal nodes help with over-squashing***
>
> We appreciate your suggestion regarding the theoretical analysis. We have added a theoretical analysis in `Section 4.1` and `Appendix L` of the updated paper that explains why fractal nodes help mitigate over-squashing. So please refer to the updated paper, and here we leave an intuitive explanation for your understanding.
>
> The key insight comes from effective resistance theory[1]. Intuitively, effective resistance between two nodes decreases when there are shorter and more disjoint paths connecting them. For example, if a path of length 7 originally connected nodes $u$ and $v$, adding a new node that connects to u in 2 hops and directly to $v$ creates a new shorter path of length 3, thereby reducing the effective resistance between $u$ and $v$.
>
> We formally prove that the effective resistance in a graph with fractal nodes ($R_f$) is always less than or equal to the original effective resistance ($R$), i.e., $R_f(u,v) \leq R(u,v)$. This theoretical guarantee directly translates to improved signal propagation bounds, as we show that the signal difference between nodes decays exponentially with the inverse of effective resistance.
>
> This theoretical analysis provides justification for why fractal nodes are effective at mitigating over-squashing while maintaining the computational efficiency of MPNNs.
>
>
> > [1] Black, Mitchell, et al. "Understanding oversquashing in gnns through the lens of effective resistance." International Conference on Machine Learning. PMLR, 2023.
>
> ---
>
> ***(W1: Theoretical Analysis) Limited discussion on the optimal number or size of fractal nodes***
>
> We added Figure 8 in `Appendix E.5` to provide a comprehensive analysis of how performance varies with different numbers of fractal nodes on 6 datasets. The results show that performance generally improves as $C$ increases, with optimal results typically achieved at $C=32$. Each dataset exhibits different sensitivity to $C$ -- for example, MolHiv shows significant improvement between $C=8$ and $C=16$, while CIFAR10 shows steady improvement up to $C=32$. Peptides-func and MolTox21 show consistent improvement with increasing $C$.
> This analysis in `Appendix E.5` addresses the impact of fractal node count and provides empirical evidence for both the general benefits of larger $C$ values and dataset-specific variations in optimal size.
>
> We have added a theoretical analysis of $C$ in `Appendix L`. As shown in our new `Corollary L.4`, increasing $C$ reduces total resistance and improves signal propagation bounds. This theoretical analysis aligns with our empirical results in Figure 8, where performance generally improves with larger $C$ but shows dataset-specific optimal values. The theory explains why larger $C$ reduces resistance and improves signal flow.  We believe this theoretical-empirical analysis addresses the reviewer's concern.
>
> ---
>
> ***(W2: Experimental Validation) The truncated content doesn't show complete experimental results***
>
> As per the reviewer's suggestion,to complete the truncated result, we have updated the sensitivity study to $C$ in `Appendix E.5`. However, for `Table 1` and `Table 10`, it would be good to note that we marked the results that were not reported in the paper of existing baseline experiments as "-".

---

> ### Author Response · Authors · 2024-11-22
>
> ***(W2: Experimental Validation) Need for more extensive comparisons with other approaches addressing similar limitations***
>
> We appreciate the reviewer’s comment regarding the need for additional experimental comparisons with other approaches addressing similar limitations.
>
> In response, we have added results comparing our method (GCN+FN) with hierarchical approaches such as hierarchical community-aware graph neural networks (HC-GNN) and hierarchical graph Transformers (HSGT, ANS-GT). Since our method also uses hierarchical message passing between original nodes and fractal nodes, we compare them. The empirical results show that GCN+FN achieves superior performance.
>
> We have already included comparisons with existing rewiring methods, such as DIGL, SDRF, FoSR, BORF, GTR, PANDA, and LASER, which aim to alleviate over-squashing by modifying graph connectivity (see `Section 5.5`. Unlike these methods, our fractal node approach integrates subgraph-level information. It allows for intra- and inter-subgraph interactions via an adaptive process that balances high- and low-frequency components of node features.
>
> Additionally, we have already reported comparing the performance of two virtual node methods [4,5] in `Appendix E.4`.
>
> We believe these additions comprehensively address the reviewer’s concerns.
>
> | Model | ogbn-arxiv | ogbn-products |
> | --- | --- | --- |
> | HC-GNN [1] | 72.79 ± 0.25  | - |
> | ANS-GT [2] | 72.34 ± 0.50 | 80.64 ± 0.29 |
> | HSGT [3] | 72.58 ± 0.31 | 81.15 ± 0.13 |
> | GCN+FN | **73.03 ± 0.37** | **81.29 ± 0.21** |
>
> > [1] Zhong, Zhiqiang, Cheng-Te Li, and Jun Pang. "Hierarchical message-passing graph neural networks." Data Mining and Knowledge Discovery 37.1 (2023): 381-408.
> >
> > [2] Zhang, Zaixi, et al. "Hierarchical graph transformer with adaptive node sampling." NeurIPS, 2022.
> >
> > [3] Zhu, Wenhao, et al. "Hierarchical transformer for scalable graph learning." IJCAI, 2023
> >
> > [4] Rosenbluth, Eran, et al. "Distinguished In Uniform: Self Attention Vs. Virtual Nodes." ICLR 2024.
> >
> > [5] Hu, Weihua, et al. "Open graph benchmark: Datasets for machine learning on graphs." NeurIPS 2020
>
> ---
>
> ***(W2: Experimental Validation) Absence of ablation studies on the impact of different subgraph partitioning strategies***
>
> See our response to **(Q1)**
>
> ---
>
> ***(W3: Practical Considerations) Limited discussion on the computational overhead of creating and maintaining fractal nodes***
>
> We thank the reviewers for raising important points about computational complexity and overhead.
> While we provided an analysis of computational complexity and actual runtime in Sections 4.2 and 5.3, we have added the following to the paper based on the reviewer's comments: 1) scalability analysis on a synthetic network (`Appendix J.1`) , 2) runtime and memory usage on a large-scale real-world graph dataset (`Appendix J.2`),  3) runtime discussion of the graph partitioning method required by our method (`Appendix I`)
>
> In `Appendix J.1`, we conducted experiments on synthetic Erdos-Renyi graphs with nodes ranging from 1,000 to 100,000 to evaluate efficiency and scalability, and we were able to verify linear space complexity while our GPU memory usage scaled linearly with the graph size.
> In `Appendix J.2`, we discuss that GCN+$\mathsf{FN}$ on ogbn-arxiv maintains the same computational requirements as GCN, and even with $\mathsf{FN}_M$, the overhead is minimal and much lower than graph transformers such as GraphGPS and Expormer.
> In `Appendix I`, our empirical analysis of graph partitioning algorithms shows that METIS with $\mathcal{O}(|\mathcal{E}|)$ complexity enables efficient fractal node generation even on large graphs such as ogbn-arxiv and ogbn-products. This shows that choosing an efficient partitioning algorithm is also important.
>
> Given these results, we believe that our method balances accuracy and computational efficiency and addresses the reviewer's concerns about the computational overhead of creating and maintaining fractal nodes.

---

> ### Author Response · Authors · 2024-11-22
>
> ***(W3: Practical Considerations) Unclear scalability analysis for very large graphs***
>
> We have added extensive empirical evidence demonstrating the scalability of our approach through both synthetic (Please see `Appendix J`) and real-world experiments (Please see `Appendix K`). We conducted systematic scaling experiments on synthetic Erdos-Renyi graphs ranging from 1,000 to 100,000 nodes, which showed linear memory scaling with graph size and sub-linear GPU training time growth.
> This scalability is further validated on large-scale real-world datasets - ogbn-arxiv (169K nodes, 1.2M edges) and ogbn-product (2.4M nodes, 61.9M edges), where our method not only scales effectively but also achieves state-of-the-art performance. Specifically, GCN+FN improves the base GCN by 1.29% on ogbn-arxiv and 5.65% on ogbn-product, while GraphSAGE+FN achieves 83.11% accuracy on ogbn-product, outperforming other methods.
>
>
> | Model | OGBN-arxiv | OGBN-product |
> | --- |--- | --- |
> | #nodes                 | 169,343 | 2,449,029 |
> | #edges                 | 1,166,243 | 61,859,140 |
> | Model                  | Accuracy | Accuracy |
> | LINKX                  | 66.18 ± 0.33 | 71.59 ± 0.71 |
> | GraphGPS               | 70.97 ± 0.41 | OOM                          |
> | NAGphormer             | 70.13 ± 0.55 | 73.55 ± 0.21                 |
> | Exphormer              | 72.44 ± 0.28 | OOM                          |
> | NodeFormer             | 69.86 ± 0.25 | 72.93 ± 0.13                 |
> | DiffFormer             | 72.41 ± 0.40 | 74.16 ± 0.31                 |
> | PolyNormer             | 71.82 ± 0.23 | 82.97 ± 0.28  |
> | SGFormer               | 72.63 ± 0.13 | 74.16 ± 0.31 |
> | GCN                    | 71.74 ± 0.29 | 75.64 ± 0.21 |
> | GCN + **FN**           | $\color{red}{73.03 ± 0.37}$  | 81.29 ± 0.21 |
> | GCN + **FN$_M$**       | $\color{blue}{72.93 ± 0.35}$ | 81.33 ± 0.33 |
> | GraphSAGE              | 71.49 ± 0.27 | 78.29 ± 0.16 |
> | GraphSAGE + **FN**     | 72.70 ± 0.11 | $\color{blue}{83.07 ± 0.35}$ |
> | GraphSAGE + **FN$_M$**     | 72.54 ± 0.30 | $\color{red}{83.11 ± 0.07}$ |
>
> ---
>
>
> ***(W3: Practical Considerations) No discussion on potential limitations or failure cases***
>
> We have expanded our discussion in Conclusion section. Our fractal nodes, while effective, are currently designed to augment MPNN architectures. Our use of METIS for subgraph partitioning, though efficient and widely used, may not always be optimal for all graph types. We believe this expanded discussion provides a better view of our method's current limitations and suggests directions for future research.

---

> ### Author Response · Authors · 2024-11-22
>
> ***(Q1) How sensitive is the method to different subgraph partitioning strategies? What criteria should be used to determine optimal partitioning?***
>
> We evaluated GINE+FN and GINE+FN$_M$ using various partitioning strategies (METIS, Random, Louvain, and Girvan-Newman) across multiple benchmark datasets. While METIS consistently achieves the best performance, the performance differences between partitioning methods are relatively small. Based on these results, we recommend METIS as the default partitioning strategy.
>
> For large-scale datasets, METIS's computational efficiency becomes even more advantageous. The Girvan-Newman algorithm, which requires betweenness centrality calculations, can time out on large graphs due to computational complexity. Notably, while Louvain cannot specify the desired number of subgraphs, METIS allows us to precisely control the number of fractal nodes by setting the desired number of subgraphs. However, since large-scale datasets still show competitive results with Louvain compared to METIS, this provides evidence that our method is not overly sensitive to the specific choice of partitioning algorithm when an appropriate strategy is used.
>
> Additionally, as discussed in `Section 3` and validated in `Section 5.1`, we provided theoretical analysis on how the spectrum of the input graph affects the impact of fractal nodes and demonstrated empirically that higher numbers of fractal nodes (subgraphs) generally lead to better performance, showing our method's sensitivity and theoretical grounding regarding the number of fractal nodes.
>
>
> | Model | Partitioning | Peptides-Func | Peptides-Struct | Molhiv | Moltox |
> | --- | --- | --- | --- | --- | --- |
> | GINE + FN | Metis | 0.6815±0.0059 | 0.2515±0.0020 | 0.7882±0.0050 | 0.7751±0.0029 |
> | | Random | 0.6533 ± 0.0103 | 0.2688 ± 0.0014 | 0.8039 ± 0.0078 | 0.7653 ± 0.0065 |
> | | Louvain | 0.6044 ± 0.0068 | 0.2799 ± 0.0015 | 0.7844 ± 0.0050 | 0.7701 ± 0.0026 |
> | | Girvan-Newman | 0.6528 ± 0.0051 | 0.2628 ± 0.0045 | 0.7837 ± 0.0078 | 0.7630 ± 0.0060 |
> | GINE + FN$_M$ | Metis | 0.7018±0.0074 | 0.2446±0.0018 | 0.8127±0.0076 | 0.7926±0.0021 |
> | | Random | 0.6680 ± 0.0066 | 0.2538 ± 0.0013 | 0.8090 ± 0.0061 | 0.7867 ± 0.0045 |
> | | Louvain | 0.6164 ± 0.0120 | 0.2789 ± 0.0022 | 0.7629 ± 0.0164 | 0.7510 ± 0.0118 |
> | | Girvan-Newman | 0.6514 ± 0.0064 | 0.2655 ± 0.0037 | 0.7763 ± 0.0174 | 0.7579 ± 0.0097 |
>
> | ogbn-arxiv | GCN + FN | GCN + FN$_M$ | GraphSAGE + FN | GraphSAGE + FN$_M$ |
> | --- | --- | --- | --- | --- |
> | Metis | 73.03 ± 0.37 | 72.93 ± 0.35 | 72.70 ± 0.11 | 72.54 ± 0.30 |
> | Random | 72.79 ± 0.37 | 73.01 ± 0.41 | 72.46 ± 0.20 | 72.46 ± 0.27 |
> Louvain | 72.73 ± 0.57 | 72.95 ± 0.26 | 72.76 ± 0.15 | 72.56 ± 0.58 |
> Girvan-Newman | Time-out | Time-out | Time-out | Time-out |

---

> ### Author Response · Authors · 2024-11-24
>
> Dear Reviewer `HGoY`,
>
> We sincerely appreciate your positive evaluation and constructive feedback that helped strengthen our paper. This is a gentle reminder since we have only a few days until the discussion period ends. We have carefully incorporated your valuable suggestions to further improve the paper. If you feel our response and revisions have addressed your concerns and comments, *we would be grateful for your continued strong support*. Please let us know if you have any additional suggestions for improvement.
>
> Best regards,
>
> Authors

---

> ### Author Response · Authors · 2024-11-28
>
> Dear Reviewer `HgOY`,
>
> As our discussion period has been extended, we would like to remind you about our responses to your valuable feedback. First and foremost, we would like to express our sincere gratitude for your thoughtful reviews, which have significantly improved the quality of our paper.
>
> Among your feedback, we found your feedback on theoretical analysis quite insightful and helpful. As detailed in our response, we have substantially strengthened our theoretical foundations in `Section 4.1` and `Appendix L`. We have also added `Corollary L.4` discussing the impact of the number of fractal nodes, supported by empirical analysis in `Appendix E.5`.
>
> We have expanded our comparisons to include HC-GNN, HSGT, and ANS-GT, and addressed your inquiry about node classification on large-scale graphs. Our method shows significant improvements over both MPNNs and state-of-the-art graph Transformers on large-scale graphs -- showing a **+1.29%** improvement over GCN (**71.74% → 73.03%**) on ogbn-arxiv, outperforming leading graph Transformer models e.g. Exphormer (72.44%) by **+0.59%**. On ogbn-products, we achieve a remarkable **+4.82%** improvement over GraphSAGE (**78.29% → 83.11%**), surpassing PolyNormer (82.97%) while maintaining the computational efficiency of MPNN.
>
> To discuss computational overhead, we have included a scalability analysis on synthetic networks in `Appendix J.1` and runtime/memory usage analysis on a large-scale real-world graph dataset in `Appendix J.2`, demonstrating our ability to outperform graph Transformers while maintaining MPNN efficiency -- *aligning with the strengths you acknowledged in your initial review*.
>
> We have also conducted a comprehensive comparison of different subgraph partitioning methods, including runtime analysis in `Table 14`, which supports our choice of METIS as the most suitable approach.
>
> We have made our best efforts to address your questions and concerns, and we remain committed to doing so. ***If our responses have adequately addressed your concerns, we would greatly appreciate your consideration in potentially revising our score upward***.
>
> Thank you for your time and dedication to improving our work.
>
> Best regards,
>
> The Authors

---

> ### Author Response · Authors · 2024-11-30
>
> Dear Reviewer `HgoY`,
>
> ***I wanted to follow up gently on our previous response, noting that the discussion period has been extended until `December 2nd`.*** We have made comprehensive efforts to address all your concerns, significantly strengthening our theoretical analysis and providing extensive empirical results, especially regarding large-scale graph performance and computational efficiency.
>
> If you have had the chance to review our responses, **we would greatly appreciate your consideration in potentially reconsidering our score**, as we believe our responses have improved the quality of our paper based on your valuable feedback.
>
> We remain open to further discussion if you have additional questions or concerns.
>
> Thank you for your time and dedication to the review process.
>
> Best regards,
>
> The Authors

---

> ### Author Response · Authors · 2024-12-03
>
> Dear Reviewer `HGoY`,
>
> As the discussion period nears its end, we wanted to confirm you received our comprehensive response addressing your concerns.
> We kindly ask if you would consider increasing our rating based on our comprehensive responses addressing your concerns about theoretical analysis and large-scale evaluations. We believe our detailed responses have significantly strengthened the paper and would greatly appreciate your feedback.
>
> Thank you for your time throughout this review process.
>
> Best regards,
>
> The Authors

---

### Official Review · Reviewer_Vhu4 · 2024-11-03

**Soundness:** 2
**Presentation:** 3
**Contribution:** 2
**Rating:** 5
**Confidence:** 3

**Summary:**

The paper introduces 'fractal nodes' to enhance MPNNs by capturing both local and global information efficiently. Fractal nodes address the over-squashing problem, allowing information to travel over long distances without significant loss, thereby addressing key limitations in standard MPNNs.

**Strengths:**

1. The paper has a clear logical structure, making it relatively easy to read.
2. The concept of using fractal nodes is interesting.

**Weaknesses:**

1. The performance improvements of the proposed method in the experiments are not particularly significant.
2. The paper could benefit from more in-depth discussion of the experimental results, beyond simply listing which method performs better.

**Questions:**

1. In Table 2, the experiments focus mainly on graph-level tasks. How does the method perform on node-level tasks, such as node classification on the ogbn-arxiv dataset? Additionally, can you discuss any potential challenges or modifications needed to apply the proposed method to node-level tasks?
2. In Table 2, why does GCN+FN_M perform worse than GINE/GatedGCN+FN_M? Additionally, what factors contribute to the proposed method performing worse than GRIT and only comparably to Graph-ViT/MLP-Mixer and Exphormer on MNIST/CIFAR10? Can you explore specific architectural differences that might explain the performance gaps?
3. When splitting graphs into subgraphs, how many nodes are contained in each subgraph in the experiments? Does each subgraph have a similar number of nodes, and how does the number of nodes in each subgraph affect performance?  Is there any trade-off involved in choosing subgraph sizes?
4. Lines 510–513 mention that 'the higher the number of fractal nodes, C, the better the performance.' What is the ratio of fractal nodes to the total number of nodes? Is there a threshold where increasing C leads to a decrease in performance?

---

> ### Author Response · Authors · 2024-11-22
>
> We sincerely thank Reviewer `Vhu4` for the review and positive feedback, highlighting our strengths in
>
> 1. Clear and logical flow enhances readability
> 2. Novel fractal node concept innovatively applies network self-similarity
>
> Below, we would like to address each of your questions and concerns:
>
> ---
>
> ***(W1) The performance improvements of the proposed method in the experiments are not particularly significant.***
>
> The performance improvement of the proposed method in the experiment is significant in that it does not depend on the structure of the graph Transformer. We can avoid the computational burden due to the theoretical computational complexity of the self-attention of the graph transformer discussed in `Section 4.2`. We also show that applying the architecture of MPNN to FN is efficient in terms of computational burden and performance improvement when verified in large-scale graph experiments (see `Appendix K`). In particular, looking at the cases in Table 15 that we added, we can see that our method maintains the memory burden and runtime of MPNN, which is more efficient than the graph transformer. At the same time, we achieve the best performance on large-scale datasets such as ogbn-arxiv, which can be said to be significant.
>
> ---
>
> ***(W2) The paper could benefit from more in-depth discussion of the experimental results, beyond simply listing which method performs better.***
>
> We would appreciate it if you would refer to the concerns about this discussion, which were partially addressed in the response to **(Q2)**. In addition, we have revised the discussion of the results in `Section 5.3`.
>
> ---
>
> ***(Q1) In Table 2, the experiments focus mainly on graph-level tasks. How does the method perform on node-level tasks, such as node classification on the ogbn-arxiv dataset? Additionally, can you discuss any potential challenges or modifications needed to apply the proposed method to node-level tasks?***
>
> We thank the reviewer for this important question about node-level tasks. We have conducted extensive experiments on large-scale node classification benchmarks, including ogbn-arxiv (169K nodes, 1.2M edges) and ogbn-products (2.4M nodes, 61.9M edges).
>
> For node-level tasks, an important modification is needed when applying our approach. While in graph-level tasks, FN$_M$ uses the mixed representation between fractal nodes from the MLP-Mixer for the final prediction, in node classification, we need to predict labels for the original nodes. Therefore, after the MLP-Mixer processes the fractal nodes in the final layer, we propagate this mixed global information back to the original nodes and use their resulting node representations for prediction.
>
> With this modification, our method shows strong performance on node classification. On ogbn-arxiv, GCN+FN achieves 73.03% accuracy, outperforming both the base GCN (71.74%) and leading Transformer-based models like Exphormer (72.44%) On ogbn-products, GraphSAGE+FN$_M$ achieves state-of-the-art 83.07% accuracy, surpassing PolyNormer.
>
> These results show that our method effectively adapts to node-level tasks with appropriate modifications, and fractal nodes successfully capture long-range dependencies needed for node classification. Also, our approach scales well to large graphs.
>
> We added these experimental results in `Appendix K`.
>
> | Model | OGBN-arxiv | OGBN-product |
> | --- |--- | --- |
> | #nodes                 | 169,343 | 2,449,029 |
> | #edges                 | 1,166,243 | 61,859,140 |
> | Model                  | Accuracy | Accuracy |
> | LINKX                  | 66.18 ± 0.33 | 71.59 ± 0.71 |
> | GraphGPS               | 70.97 ± 0.41 | OOM                          |
> | NAGphormer             | 70.13 ± 0.55 | 73.55 ± 0.21                 |
> | Exphormer              | 72.44 ± 0.28 | OOM                          |
> | NodeFormer             | 69.86 ± 0.25 | 72.93 ± 0.13                 |
> | DiffFormer             | 72.41 ± 0.40 | 74.16 ± 0.31                 |
> | PolyNormer             | 71.82 ± 0.23 | 82.97 ± 0.28  |
> | SGFormer               | 72.63 ± 0.13 | 74.16 ± 0.31 |
> | GCN                    | 71.74 ± 0.29 | 75.64 ± 0.21 |
> | GCN + **FN**           | $\color{red}{73.03 ± 0.37}$  | 81.29 ± 0.21 |
> | GCN + **FN$_M$**       | $\color{blue}{72.93 ± 0.35}$ | 81.33 ± 0.33 |
> | GraphSAGE              | 71.49 ± 0.27 | 78.29 ± 0.16 |
> | GraphSAGE + **FN**     | 72.70 ± 0.11 | $\color{blue}{83.07 ± 0.35}$ |
> | GraphSAGE + **FN$_M$**     | 72.54 ± 0.30 | $\color{red}{83.11 ± 0.07}$ |

---

> ### Author Response · Authors · 2024-11-22
>
> ***(Q2) In Table 2, why does GCN+FN_M perform worse than GINE/GatedGCN+FN_M? Additionally, what factors contribute to the proposed method performing worse than GRIT and only comparably to Graph-ViT/MLP-Mixer and Exphormer on MNIST/CIFAR10? Can you explore specific architectural differences that might explain the performance gaps?***
>
>
> Our analysis reveals that the performance gap between GCN+FN$_M$ and GINE/GatedGCN+FN$_M$ is primarily attributable to the inherent capacity differences in the base MPNN architectures rather than the fractal nodes themselves. Since our fractal nodes framework is designed to be model-agnostic and can be integrated with various MPNN architectures, the overall performance is influenced by both the base model's capabilities and the enhancement provided by FN$_M$. GatedGCN and GINE have demonstrated stronger baseline performance compared to GCN across multiple benchmarks, which explains the superior results when combined with our fractal nodes.
>
> Regarding the performance comparison with GRIT on CIFAR10, we acknowledge that GRIT's sophisticated self-attention mechanism and architectural components like positional encoding and degree scalers may be particularly well-suited for this specific dataset. However, it's important to note that our method shows competitive performance with other transformer-based approaches like Graph-ViT/MLP-Mixer and Exphormer. These models differ in their architectural choices - while GRIT proposes an advanced self-attention mechanism without message passing layers, Graph-ViT/MLP-Mixer and Exphormer incorporate MPNNs in different ways. For instance, Exphormer maintains GraphGPS's architecture while modifying the graph structure, utilizing both MPNN and self-attention components.
>
> Our MPNN+FN$_M$ achieves comparable performance through a different mechanism: it captures global information at the subgraph level through fractal nodes and enables longer-range interactions through the final mixing of fractal nodes, while maintaining the computational efficiency of MPNNs. This demonstrates that our approach offers a valuable alternative to transformer-based models, particularly when considering the trade-off between performance and computational complexity.
>
> ---
>
> ***(Q3) When splitting graphs into subgraphs, how many nodes are contained in each subgraph in the experiments? Does each subgraph have a similar number of nodes, and how does the number of nodes in each subgraph affect performance? Is there any trade-off involved in choosing subgraph sizes?***
>
> Our analysis incorporates both theoretical and empirical evidence regarding subgraph sizes and their impact. Using METIS partitioning, we see distinct distribution patterns across different datasets as shown in the histograms (`Figures 12 to 16`):
>
> For Peptide-func/struct (`Figure 12`), we observe highly concentrated, unimodal distributions. The tight, symmetric peaks indicate METIS creates very balanced partitions for these molecular graphs.  CIFAR10's distributions (`Figure 13`) show more complex patterns. While smaller `C` values (2,4) maintain relatively balanced splits, larger $C$ values (16,32) produce bimodal distributions with two distinct peaks.  MolTox (`Figure 16`) exhibits increasingly skewed distributions as `C` grows, with a pronounced tail toward larger subgraph ratios. Despite this skew, the primary peaks remain well-defined, indicating METIS still maintains some balance while adapting to the molecular topology.
>
> We also added theoretical analysis on the number of subgraphs that affect over-squashing in `Corollary N.6`. In addition, we added sensitivity analysis on the performance improvement as the number of subgraphs split by METIS increases in `Appendix E.5`. Since we can adjust the number of subgraphs rather than adjusting the size of the subgraphs using METIS, there is a performance trade-off depending on the number of subgraphs. The sensitivity analysis results show that 32 subgraphs are optimal for most datasets, and since the number of subgraphs is smaller, the size of the nodes in the subgraphs generally increases, so we conjecture that the performance can be optimal when the size of the subgraph is smaller, that is when the number of subgraphs is sufficient.

---

> ### Author Response · Authors · 2024-11-22
>
> ***(Q4) Lines 510–513 mention that 'the higher the number of fractal nodes, C, the better the performance.' What is the ratio of fractal nodes to the total number of nodes? Is there a threshold where increasing C leads to a decrease in performance?***
>
>
> Thank you for raising this question about the relationship between C and model performance. While our statement about "higher $C$ generally improving performance" was based on our search range up to C=32 (as shown in Appendix E.5's sensitivity analysis), we extended the range to 128. The sensitivity to $C$ for Peptides-struct is as follows.
>
> | $C$ | $C=8$ | $C=16$ | $C=32$ | $C=64$ | $C=128$ |
> |---|---|---|---|---|---|
> | GINE+FN  | 0.2535 | 0.2532 | 0.2515 | 0.2520 | 0.2696 |
> | GINE+FN$_M$  | 0.2504 | 0.2446 | 0.2459 | 0.2453 | 0.2591 |
>
> These results indicate that while performance generally improves up to $C=32$ as reported, there exists a threshold beyond which increasing $C$ becomes detrimental. Considering the average graph size in Peptides dataset (150.9 nodes), too many subgraphs relative to total nodes can harm performance. We find that the optimal $C$ value depends on the dataset's graph size, with $C=32$ typically providing the best results for graph-level tasks, and a clear performance degradation threshold between $C=64$ and $C=128$ for Peptides-struct.

---

> ### Author Response · Authors · 2024-11-24
>
> Dear Reviewer `Vhu4`,
>
> We appreciate your comments on helping us improve our paper in many aspects. This is a gentle reminder since we have only a few days until the discussion period ends. We have tried our best to address your questions. We would appreciate it if you could either confirm that our responses have satisfactorily addressed your concerns and consider a more favorable score or let us know if you have any follow-up questions that require our attention.
>
> Best,
>
> Authors

---

> > ### Comment · Reviewer_Vhu4 · 2024-11-26
> > **Thank you for the rebuttal.**
> >
> > Thank you to the authors for the response and the efforts put into the rebuttal. I would like to maintain my score.
> >
> > Regarding Q1, you mentioned "Therefore, after the MLP-Mixer processes the fractal nodes in the final layer, we propagate this mixed global information back to the original nodes and use their resulting node representations for prediction." Could you elaborate on how this is done?

---

> ### Author Response · Authors · 2024-11-26
>
> ***(Follow-up Q1) Regarding Q1, you mentioned "Therefore, after the MLP-Mixer processes the fractal nodes in the final layer, we propagate this mixed global information back to the original nodes and use their resulting node representations for prediction." Could you elaborate on how this is done?***
>
> The key difference between graph-level and node-level tasks lies in how we utilize the processed fractal node representations from the MLP-Mixer. Let me add detail to using equations from our paper:
>
> In graph-level tasks, we follow `Equation (8)` where fractal nodes are mixed through MLP-Mixer:
> $$\tilde{F} = \text{MLPMixer}(F^{(L)}), F^{(L)} = [f^{(L)}\_1, f^{(L)}\_2, ..., f^{(L)}\_{C}]$$
>
> These mixed fractal node representations are then used directly for graph-level prediction through global pooling.
>
> However, **for node classification**, we need node-level predictions, so we adapt this process. After MLP-Mixer processes the $C$ fractal nodes according to `Equations 9 and 10`, we have:
> $\tilde{F}^{(L)} \in \mathbb{R}^{C \times d}$
>
> For node classification, we need to align this with all nodes in their respective subgraphs. Let $V_c$ be the set of nodes in subgraph $c$. The process can be expressed as:
> $h^{(\text{final})}_v = h^{(L)}_v + \tilde{f}^{(L)}_c, \quad \forall v \in V_c$
> where $\tilde{f}^{(L)}_c$ is the c-th row of $\tilde{F}^{(L)}$ corresponding to the fractal node of subgraph $c$.
> In the code:
> ```python
> fractal_x = self.reshape(fractal_x)[batch_x]  # batch_x indicates subgraph membership of each node
> x_final = x + fractal_x
> ```
> Here, `batch_x` acts as an indexing vector that repeats each fractal node representation for all nodes in its corresponding subgraph. Specifically, if node $v$ belongs to subgraph $c$, then `batch_x[v] = c`, and `fractal_x[batch_x][v] = fractal_x[c]`.
> More formally, this operation can be written as:
> $h^{(L+1)}\_v = h^{(L)}\_v + \tilde{f}^{(L)}\_{\text{batch}[v]}, \quad \forall v \in V$
> where $\text{batch}[v]$ maps node $v$ to its corresponding subgraph index. This ensures each node receives the processed fractal node information from its subgraph, maintaining consistency with our method while adapting it for node-level predictions.
>
> We hope this clarifies your follow-up question.

---

> ### Author Response · Authors · 2024-11-26
>
> Thank you for your thoughtful follow-up question.
>
> Your review has been important in helping us improve our paper. We have been able to provide additional results on node classification tasks for large-scale graphs, expand our discussion of existing graph-level task results, and add analysis regarding subgraph sizes and node ratios within subgraphs. In particular, regarding the performance improvements noted in **W1**, we want to emphasize that these improvements become even more significant in large-scale graphs, as you suggested. As demonstrated in our results (`Tables 15 and 16`), *our method can effectively enhance MPNN capacity with its maintained efficiency as an alternative to graph Transformers* -- an interesting architectural design that you acknowledged in your original review.
>
> We hope that our additional response, combined with our previous ones, effectively addresses your concerns. Please don't hesitate to let us know if you have any other concerns or additional questions.
>
> Thank you again for your constructive feedback.
>
> Best regards,
>
> The Authors

---

> ### Author Response · Authors · 2024-11-27
>
> Dear Reviewer `Vhu4`,
>
> Again, we sincerely appreciate your thoughtful question regarding the node classification adaptation of our method. Following your insightful question, we have carefully added detailed implementation information in `Appendix K` to explain our implementation in node-level tasks.
>
> ***If you feel our responses and additional materials have adequately addressed your concerns, we would be deeply grateful for your consideration in potentially reassessing and increasing our score.***
>
> Our method demonstrates significant improvements over both MPNNs and state-of-the-art graph Transformers on large-scale graphs -- showing a **+1.29% improvement** over GCN (***71.74% → 73.03%***) on ogbn-arxiv, outperforming leading graph Transformer models e.g. Exphormer (72.44%) by **+0.59%**. On ogbn-products, we achieve a remarkable **+4.82% improvement** over GraphSAGE (***78.29% → 83.11%***), surpassing PolyNormer (82.97%) while maintaining the computational efficiency of MPNN.
>
> We would be most grateful for any additional feedback that could help us further improve our work. We appreciate any additional feedback that can help us improve our paper, and we will do our best to address any additional questions and concerns raised by Reviewer `Vhu4`.
>
> Thank you for your valuable time and continued engagement with our discussion.
>
> Respectfully,
>
> The Authors

---

> ### Author Response · Authors · 2024-11-30
>
> Dear Reviewer `Vhu4`,
>
> **As the discussion period has been extended until December 2nd**, I wanted to remind you about our previous response to your question. We have carefully added detailed implementation information in `Appendix K` as requested, and our method has shown significant improvements over both MPNNs and state-of-the-art graph Transformers on large-scale graphs, as detailed in our response.
>
> If you have had the opportunity to review our responses, ***we would greatly appreciate your consideration in potentially revising and increasing our score***, as we believe we have comprehensively addressed your concerns and strengthened the paper accordingly.
>
> We remain open to any additional feedback or questions you may have before the December 2nd deadline. Your insights have been invaluable in improving our work, and we are committed to addressing any remaining concerns.
>
> Thank you for your continued engagement with our discussion.
>
> Best regards,
>
> The Authors

---

> ### Author Response · Authors · 2024-12-03
>
> Dear Reviewer `Vhu4`,
>
> We are writing to follow up as we approach the end of the rebuttal period. We would like to remind you of our previous responses to your questions and concerns.
>
> Thanks to your valuable feedback, we have been able to significantly strengthen both our method's scalability to large-scale graphs and the content of our paper. Specifically, we demonstrated that our method outperforms 7 representative graph Transformers by achieving a **1.29%** improvement over GCN on ogbn-arxiv (as shown in `Table 16`). Moreover, as evidenced in `Table 15`, we maintain the same runtime and memory requirements as the base GCN while improving upon graph Transformers.
>
> Your constructive feedback has helped us enhance our paper substantially. **Given that our current rating remains below the acceptance threshold of `5`, we would appreciate hearing what concerns remain that are preventing you from improving your score.**
>
> ***If we have adequately addressed your feedback, we would be grateful if you would consider raising the rating.*** Your expertise and feedback have been invaluable in improving our work.
>
> Thank you for your continued engagement in our discussion.
>
> Best regards,
>
> The Authors

---

> > ### Comment · Reviewer_Vhu4 · 2024-12-03
> >
> > Thanks for the rebuttal. I appreciate the effort put into addressing the concerns. However, I still believe that the performance improvements of the proposed method in the experiments are not particularly significant. While the method performs well on node-level tasks, the main results in the original paper still face this issue.
> >
> > Additionally, for choosing the subgraph size, the approach relies heavily on sensitivity analysis. I would have preferred a more theoretical foundation, as this is a key component of the proposed method.
> >
> > Overall, while I acknowledge and appreciate the authors' efforts in the rebuttal, which might justify a score increase for effort, I cannot justify raising my score for the paper itself. Therefore, I would like to maintain my current score. Thank you.

---

> ### Author Response · Authors · 2024-12-03
>
> Dear Reviewer `Vhu4`,
>
> We would like to reiterate 2 points to the reviewer.
>
> **(1)** We would like to emphasize the significance of our performance improvements. In graph-level tasks, recent approaches have mostly relied on graph Transformers. As you know, this dependence on graph Transformers can introduce computational overhead. We have demonstrated that by integrating our FN into MPNN architectures, we can surpass both MPNN and graph Transformer performance without significantly increasing runtime and memory requirements as base MPNNs (as evidenced in `Table 3`).
>
> For instance, GINE+FN$_M$ outperforms 7 graph Transformer-based models across all datasets: Peptides-func, Peptides-struct, MolHiv, and MolTox21. Additionally, GatedGCN+FN$_M$ achieves superior performance compared to seven Transformer-based baselines on MNIST. While our method achieves second-best performance on CIFAR10, we would appreciate if you could reconsider the significance of our primary results.
>
> **(2)** Regarding our subgraph analysis, **rather than heavily relying on sensitivity analysis, we theoretically guarantee via `Corollary L.4` that total resistance decreases as the number of subgraphs increases.** This means that increasing $C$ reduces total resistance and alleviates over-squashing. While our sensitivity analysis revealed that $C=32$ performs well in most datasets, this empirically confirms the theoretical trend established by `Corollary L.4`.
>
> We hope these points can align with your preferences and help clarify the strengths and contributions of our work. We would be grateful if you would consider our responses in your evaluation.
>
> Thank you for your consideration.
>
> Best regards,
>
> The Authors

---

### Official Review · Reviewer_yJ2G · 2024-11-04

**Soundness:** 2
**Presentation:** 2
**Contribution:** 2
**Rating:** 5
**Confidence:** 4

**Summary:**

The paper proposes 'fractal nodes' to enhance Graph Neural Networks, addressing the over-squashing. By integrating these nodes into GNN, the method improves long-range dependency modeling, achieving competitive performance with graph Transformers while maintaining computational efficiency.

**Strengths:**

1. The method is straightforward with low complexity and seems to be effective based on the results obtained.

2. The authors conducted a variety of benchmarking experiments and meticulously outlined the settings for each benchmarking experiment.

**Weaknesses:**

1. The proposed framework appears to lack novelty, resembling a combination of ClusterGCN[1] and VirtualNode methodologies. The use of Metis clustering, akin to ClusterGNN, and the introduction of fractal nodes that connect to all nodes within a cluster, essentially reflect a variation of the VirtualNode concept.

2. The misapplication of LPF and HPF is evident. As per the definition of LPF (Equation 6), it represents the mean of every node feature within a cluster, failing to capture the graph structure or low-frequency graph signal components. Similarly, HPF (Equation 7), calculated as the original feature minus LPF, also fail in capturing high-frequency graph signal components. It is crucial for the authors to revisit the fundamentals of spectral graph theory to refine their understanding of these concepts.

3. The analysis of the properties in Section 4 appears superficial. The L2 norm distribution of node feature merely indicates the smoothing of node features and does not inherently reveal whether local or global information is being captured. The claim that the proposed method is more balanced lacks rigor, it is essential to provide concrete metrics for comparing differences before making such assertions.

4. The analysis comparing the proposed method to graph coarsening techniques and virtual nodes is inaccurate. Firstly, the fractal nodes are not dynamically generated during training, they are precomputed by METIS. Secondly, virtual nodes do not always utilize mean pooling as an aggregator, instead, they commonly function as regular graph nodes, sharing the same hidden space with other nodes in most scenarios.

5. The author claim the core concept of 'fractal nodes for enforcing self-similarity,' yet upon reviewing the entire paper, the practical impact of 'self-similarity' on enhancing GNN performance remains unclear to me.

[1] Wei-Lin Chiang, Xuanqing Liu, Si Si, Yang Li, Samy Bengio, and Cho-Jui Hsieh. Cluster-gcn: An
efficient algorithm for training deep and large graph convolutional networks. In Proceedings of the
25th ACM SIGKDD international conference on knowledge discovery & data mining, pp. 257–266,
2019.

**Questions:**

1. The expressive power evaluation in Section 5.2 is impressive. However, given the absence of a formal proof regarding this expressive power, could you share the code detailing how the method achieved a score of 100 in CSL and SR25?

2. Additionally, we were unable to reproduce the results shown in the paper using the provided source code.
    - Runtime (seconds per epoch) on a 3090 machine: 58s (our evaluation) vs 5.03s (paper)
    - 0.27, 0.252, and 0.2474 in 3 runs vs 0.2464±0.0014 (paper); Could the authors clarify the number of runs used to obtain the best result?

---

> ### Author Response · Authors · 2024-11-22
>
> We sincerely thank Reviewer `yJ2G` for their feedback, highlighting our strengths in:
> - Simple yet effective methodology
> - Low computational complexity
> - Extensive experiments on various benchmark datasets
>
> Below, we would like to address each of your questions and concerns:
>
> ---
>
> ***(W1) The proposed framework appears to lack novelty, resembling a combination of ClusterGCN and VirtualNode methodologies. The use of Metis clustering, akin to ClusterGNN, and the introduction of fractal nodes that connect to all nodes within a cluster, essentially reflect a variation of the VirtualNode concept.***
>
> While our method does utilize METIS clustering and has some superficial similarities to ClusterGCN and virtual nodes, the fundamental principles and objectives are distinctly different.
>
> ClusterGCN uses clustering primarily as a computational optimization for mini-batch training, whereas our fractal nodes framework uses clustering to partition graphs to create fractal nodes that effectively address the over-squashing problem.
>
> Our fractal nodes framework differs fundamentally from virtual nodes in both design philosophy and technical implementation. While virtual nodes, as introduced by [1], act as a global node connected uniformly to all graph nodes, our fractal nodes create a hierarchical structure guided by the graph topology. According to [2], virtual nodes provide shortcuts for message passing between nodes, with uniform importance assigned to all nodes. In contrast, our fractal nodes adaptively combine both low and high-frequency components of subgraph information.
>
> A key theoretical insight demonstrated in our analysis is that virtual nodes, using simple mean pooling (as shown in [3]), can only capture the lowest frequency (DC) component of node features. Our framework goes beyond this limitation by incorporating both low-pass filtering (LPF) and high-pass filtering (HPF) mechanisms, allowing fractal nodes to preserve both global structure and local details.
>
> Moreover, in `Section 5.3` we show that our method outperforms graph Transformers, and in `Appendix E.4` the empirical results show that it outperforms standard virtual nodes from [2] and [3].
>
>
> > [1] Gilmer, Justin, et al. "Neural message passing for quantum chemistry." ICML 2017.
> >
> > [2] Rosenbluth, Eran, et al. "Distinguished In Uniform: Self Attention Vs. Virtual Nodes." ICLR 2024.
> >
> > [3] Cai, Chen, et al. "On the connection between mpnn and graph transformer." ICML 2023.
>
> ---
>
> ***(W2) The misapplication of LPF and HPF is evident. As per the definition of LPF (Equation 6), it represents the mean of every node feature within a cluster, failing to capture the graph structure or low-frequency graph signal components. Similarly, HPF (Equation 7), calculated as the original feature minus LPF, also fail in capturing high-frequency graph signal components. It is crucial for the authors to revisit the fundamentals of spectral graph theory to refine their understanding of these concepts.***
>
>
> I appreciate the reviewer's concern about our signal processing terminology and implementation. Let me clarify our framework's theoretical foundations and address the specific points raised:
>
> From a spectral graph theory perspective, you are correct that our LPF implementation through mean pooling (Equation 6) does not directly operate on the graph Laplacian eigenvectors. However, our framework takes a different but complementary approach:
>
> - Theorem 3.1 rigorously proves that our mean pooling operation extracts the DC component in the discrete Fourier domain of node features within each subgraph (Please see Appendix A). While this differs from graph spectral filtering, it provides a computationally efficient way to separate baseline signals from local variations.
> - Our HPF design (Equation 7) then preserves the deviations from this baseline, capturing feature differences. The learnable parameter $\omega_{c}$ allows adaptive balancing of these components based on the downstream task requirements.
>
> As the reviewer noted, our LPF emphasizes capturing DC at the subgraph level rather than capturing low-frequency components of the overall graph structure. We have revised the paper to clarify this. Instead, we remind you that our FN$_M$ handles long-range interactions over the entire graph.

---

> ### Author Response · Authors · 2024-11-22
>
> ***(W3) The analysis of the properties in Section 4 appears superficial. The L2 norm distribution of node feature merely indicates the smoothing of node features and does not inherently reveal whether local or global information is being captured. The claim that the proposed method is more balanced lacks rigor, it is essential to provide concrete metrics for comparing differences before making such assertions.***
>
> We appreciate the reviewer's concern about the superficial analysis based on L2 norm distributions. We have now strengthened our analysis with a more rigorous frequency response analysis that provides concrete metrics for comparing how different methods process and preserve information across frequency scales. As shown in new `Fig. 2`, our frequency response analysis reveals distinct quantitative patterns: self-attention shows responses on all frequencies, while mean pooling shows minimal response especially in high frequencies, indicating significant information loss.
> In contrast, fractal nodes show a well-balanced frequency response profile, with strong low-frequency components capturing subgraph-level global context and high-frequency components preserving local details [4]. This provides evidence for the balanced information processing capabilities of fractal nodes.
>
> > [4] Shuman, David I., et al. "The emerging field of signal processing on graphs: Extending high-dimensional data analysis to networks and other irregular domains." IEEE signal processing magazine 30.3 (2013): 83-98.
>
> ---
>
> ***(W4) The analysis comparing the proposed method to graph coarsening techniques and virtual nodes is inaccurate. Firstly, the fractal nodes are not dynamically generated during training, they are precomputed by METIS. Secondly, virtual nodes do not always utilize mean pooling as an aggregator, instead, they commonly function as regular graph nodes, sharing the same hidden space with other nodes in most scenarios.***
>
> We appreciate the reviewer's careful reading and feedback on our comparison of our method with existing methods. We acknowledge that our description may be misleading, and have revised `Section 4.3`.
>
> ---
>
> ***W5. The author claim the core concept of 'fractal nodes for enforcing self-similarity,' yet upon reviewing the entire paper, the practical impact of 'self-similarity' on enhancing GNN performance remains unclear to me.***
>
> The concept of self-similarity in our work manifests in both structural and feature aspects, with clear practical benefits. We have updated our paper, specifically the `Introduction` section, to clarify these distinct aspects and avoid confusion.
>
> Regarding feature self-similarity, our fractal nodes enforce this property by adaptively combining low and high-frequency components of node features within each subgraph. The LPF component ensures the fractal node captures the common/similar features of nodes in its subgraph, while the HPF component with learnable parameter ω enables task-specific adaptation of how much high-frequency variation to retain. In terms of structural self-similarity, the graph partitioning creates subgraphs that can show structural self-similarity with the original graph (see `Appendix C`).
>
> The practical impact of this approach comes from fractal nodes adaptively balancing global (LPF) and local (HPF) feature information, enabling multi-scale feature representation through self-similarity. Our experiments validate the practical impact: 1) ablation studies demonstrate that both LPF and HPF components are essential for performance (see `Appendix E.1`), showing how feature self-similarity through balanced subgraph-level global and local feature information contributes to model effectiveness; 2) the consistent improvements over baseline MPNNs confirm that our method enhances representation learning; 3) our analysis in `Section 5.1` reveals that fractal nodes help alleviate oversquashing by using fractal nodes which contain feature self-similarity of nodes in each subgraph.
>
> We hope that this explanation will alleviate reviewers' concerns.
>
> ---
>
> ***(Q1)The expressive power evaluation in Section 5.2 is impressive. However, given the absence of a formal proof regarding this expressive power, could you share the code detailing how the method achieved a score of 100 in CSL and SR25?***
>
> We have added this code to the source code links we provided to share the code for CSL and SR25.
>
> ---
>
> ***(Q2) Additionally, we were unable to reproduce the results shown in the paper using the provided source code. Could the authors clarify the number of runs used to obtain the best result?***
>
> For the Peptides dataset, we reported the mean and standard deviation of 4 runs, following the convention for LRGB data. We added the results file to the source code link provided for reference to the best performance.

---

> ### Author Response · Authors · 2024-11-24
>
> Dear Reviewer `yJ2G`,
>
> Thank you for your thoughtful and detailed comments, which have helped us enhance our paper. This is a gentle reminder since we have only a few days until the discussion period ends. We have carefully considered each of your concerns and made improvements to address them. We hope our detailed responses have helped clarify our contributions and method.
>
> If you find our response and revision satisfactory, we would be grateful if you would consider a more favorable score and we welcome any additional thoughts or questions you might have.
>
> Best,
>
> Authors

---

> > ### Comment · Reviewer_yJ2G · 2024-11-25
> > **After reviewing the revised paper, the response, and the codes, I have decided to keep my score unchanged.**
> >
> > Thank you to the authors for their thoughtful response. I appreciate the time and effort they have put into addressing the feedback.
> >
> > W1: My primary concern remains unaddressed. The novelty of the proposed method is not yet clear to me. In particular, I strongly disagree with the assertion that the High Pass Filter (HPF) and Low Pass Filter (LPF) effectively capture the frequency components of the subgraph. Let's delve deeper into this point in W2. Regarding virtual nodes, as highlighted in W4, it's important to note that mean pooling is not the only aggregator. I maintain my stance that your approach seems to resemble a variation of the VirtualNode concept.
> >
> > W2: My primary concern remains unaddressed, and this particular weakness is of utmost importance to me. The analysis of frequency components seems to focus solely on the feature level, overlooking the essence of the graph signal itself. Claiming to capture low-frequency components through mean pooling on all node features within a graph (or subgraph) without considering the interconnections between nodes is not acceptable and appears to be disconnected from fundamental principles of graph signal processing. I strongly urge the authors to revisit the foundational concepts of spectral graph theory.
> >
> > After reviewing your code, it appears that Equation 7 should read as $HPF(h_{v,c}^{(l+1)}) = h_c - LPF(h_{v,c}^{(l+1)})$. By combining Equations 5, 6, and 7, the feature of the fractal node can be represented as $f_c^{(l+1)} = (1 - \omega) \times \frac{1}{C} \sum_{v \in \mathcal{V}c} h{v,c}^{(l+1)} + \omega \times h_c^{(l+1)}$. Based on this formulation, I recommend that the authors reconsider how they explain that the above equation captures the high frequency component (local) of a subgraph, which ideally should reflect the disparity between neighboring node features rather than the disparity between the fractal node feature and the pooling feature of the subgraph.
> >
> > W3. My primary concern remains unaddressed. The frequency analysis is not convincing to me unless you can address W2.
> >
> > W4. While my concern is partially addressed, the revised statement has given rise to a new concern. Specifically: 1) What specific information is being lost in the "pre-defined coarsened network"? 2) if it refer to high-frequency components of the subgraph, I disagree with your point particularly given that previous methods utilize transformers as the aggregator."
> >
> > W5. My concern is partially addressed.
> >
> > Q1. After reviewing and running your code, a critical issue concerning the expressive power of your method has surfaced. While I have no objections to incorporating additional features like RWPE to boost performance in graph classification tasks, it is imperative to note that the use of RWPE may mask the inherent expressiveness of your method. RWPE is already acknowledged for its expressiveness in addressing the CSL and SR25 problems. It's worth mentioning that your method fails to function effectively when RWPE is omitted. Notably, upon removing RWPE from your code, the best performance on CSL achieved was 0.3000. In my view, this significantly impacts the contribution of this work.
> >
> > Q2. My concern is addressed.

---

> ### Author Response · Authors · 2024-11-26
>
> We appreciate the reviewer's detailed and thoughtful review of our paper from multiple perspectives, and the additional concerns they pointed out. We would like to address the reviewer's remaining concerns below.
>
> ---
>
> ***W1: The novelty of the proposed method is not yet clear to me. Regarding virtual nodes, as highlighted in W4, it's important to note that mean pooling is not the only aggregator. I maintain my stance that your approach seems to resemble a variation of the VirtualNode concept.***
>
> We would like to clarify that while our approach shares some similarities with virtual nodes, as far as the reviewer is concerned, there are key differences.
>
> While the supernode in [1], the dummy supernode in [2], and the virtual node in [3] all focus on aggregating and propagating global information in the graph, they are inherently different in design and purpose from our Fractal Node. The supernode in [1] uses GRUs and attention mechanisms to effectively aggregate global information and propagate it to all nodes. On the other hand, the DML dummy supernode in [2] learns a global representation of the graph without interfering with the local learning of genuine nodes, but does not handle interactions between subgraphs. In [3], a virtual node is created by mean pooling and added to the representation of the graph node hidden vector as in our mean pooling method (see two snippets of [3]'s code ([link1](https://github.com/toenshoff/VN-vs-GT/blob/d493c1f018d1370e125cfb5954cb52f10ee685db/graphgps/layer/mpnn_layer.py#L130-L139
> ), [link2](https://github.com/toenshoff/VN-vs-GT/blob/d493c1f018d1370e125cfb5954cb52f10ee685db/graphgps/layer/virtual_node.py#L21))).
>
> In comparison, our fractal nodes are created separately for each subgraph, and we use a combination of LPF and HPF to learn subgraph-level-global information and local variations simultaneously. We also utilize MLP-Mixer to mix the information between subgraphs/fractal nodes to enhance long-range interactions.
>
> Thus, our method differs from the existing virtual node or super node approaches.
>
> > [1] Ishiguro, Katsuhiko, Shin-ichi Maeda, and Masanori Koyama. "Graph warp module: an auxiliary module for boosting the power of graph neural networks in molecular graph analysis." arXiv preprint arXiv:1902.01020 (2019).
> >
> > [2] Li, Junying, Deng Cai, and Xiaofei He. "Learning graph-level representation for drug discovery." arXiv preprint arXiv:1709.03741 (2017).
> >
> > [3] Rosenbluth, Eran, et al. "Distinguished In Uniform: Self Attention Vs. Virtual Nodes." ICLR 2024.

---

> ### Author Response · Authors · 2024-11-26
>
> ***W2: My primary concern remains unaddressed, and this particular weakness is of utmost importance to me. The analysis of frequency components seems to focus solely on the feature level, overlooking the essence of the graph signal itself. Claiming to capture low-frequency components through mean pooling on all node features within a graph (or subgraph) without considering the interconnections between nodes is not acceptable and appears to be disconnected from fundamental principles of graph signal processing. I strongly urge the authors to revisit the foundational concepts of spectral graph theory.***
>
>
> We appreciate your concerns about graph signal processing principles. We want to clarify that our method is fundamentally based on discrete signal processing rather than specifically graph signal processing. This is well-illustrated by a fundamental property in signal processing: the arithmetic mean is equivalent to the DC component of a signal's Fourier transform (please see [4,5]). Note that we intentionally did not use terminology from graph signal processing or spectral graph theory in our paper.
>
> In signal processing, it is well-established that a signal's arithmetic mean and DC component are identical. Here, we only
> consider discrete Fourier transform (DFT) on real-value domain, for a discrete signal x[n], the DFT is defined as:
>  $$\begin{align*}\mathsf{DFT} = \frac{1}{\sqrt{n}} \begin{bmatrix}
> 1 & 1 & \cdots & 1 \\\\\
> 1 & e^{2\pi j} & \cdots & e^{2\pi j(n-1)} \\\\\
> \vdots & \vdots & \ddots & \vdots \\\\\
> 1 & e^{2\pi j(k-1)\cdot 1} & \cdots & e^{2\pi j(k-1)\cdot(n-1)} \\\\\
> \vdots & \vdots & \ddots & \vdots \\\\\
> 1 & e^{2\pi j(n-1)} & \cdots & e^{2\pi j(n-1)^2}
> \end{bmatrix}
>  \end{align*}$$
> When we evaluate the DFT at $k = 0$ (DC component):
> $$\mathcal{F}(0) = \frac{1}{N} \sum_{n=0}^{N-1} x[n]e^{-2\pi j \cdot  \frac{0 \cdot n}{N}} = \frac{1}{N} \sum_{n=0}^{N-1} x[n] = \bar{x}.$$
>
>
> In our method, we can express the DC (Direct Current) and HC (high-frequency component) formally as:
> $$\mathcal{DC}[x] = \mathsf{DFT}^{-1}\text{diag}(1,0,\cdots,0)\mathsf{DFT}x = \frac{1}{n}\mathbf{1}\mathbf{1}^Tx ,$$
> $$\mathcal{HC}[x] = \mathsf{DFT}^{-1}\text{diag}(0,1,\cdots,1)\mathsf{DFT}x = \mathsf{DFT}^{-1}(\mathbf{1}-\text{diag}(1,0,\cdots,0))\mathsf{DFT}x = (\mathbf{I} - \frac{1}{n}\mathbf{1}\mathbf{1}^T)x .$$
>
>
> This relationship shows that when we compute the mean of our node features in subgraphs:
> $$f_c^{mean} = \frac{1}{|\mathcal{V}\_{c}|} \sum_{v \in \mathcal{V}_c} h_v .$$
>
> We are extracting the DC component in the discrete Fourier domain. *Note that the division by $C$ in our Equation 6 is a typo, and has been corrected in the updated paper to $|\mathcal{V}_c|$ in $\color{blue}{blue}$. Sorry for the confusion.*
>
> We appreciate your concerns about graph signal processing principles. We want to clarify that our method is fundamentally based on discrete signal processing rather than specifically graph signal processing. This is well-illustrated by a fundamental property in signal processing: the arithmetic mean is equivalent to the DC component of a signal's Fourier transform.
>
> > [4] https://dsp.stackexchange.com/questions/10254/why-is-x0-the-dc-component
> >
> > [5] https://dsp.stackexchange.com/a/81091

---

> ### Author Response · Authors · 2024-11-26
>
> ***W2. After reviewing your code, it appears that Equation 7 should read as $HPF(h_{v,c}^{(l+1)}) = h_c - LPF(h_{v,c}^{(l+1)})$. By combining Equations 5, 6, and 7, the feature of the fractal node can be represented as $f_c^{(l+1)}=(1-\omega)\times\frac{1}{c}\sum_{v\in\mathcal{V}\_C} h_{v,c}^{(l+1)} + \omega \times h_c^{(l+1)}$. Based on this formulation, I recommend that the authors reconsider how they explain that the above equation captures the high frequency component (local) of a subgraph, which ideally should reflect the disparity between neighboring node features rather than the disparity between the fractal node feature and the pooling feature of the subgraph.***
>
> We appreciate your careful review of the implementation. However, we would like to clarify that our formulation is theoretically sound and consistent with discrete signal processing principles. Our HPF formulation directly follows from the classical signal processing definition as we mentioned above:
>
> $$\mathcal{HC}[x] = (\mathbf{I} - \frac{1}{|\mathcal{V}_c|}\mathbf{1}\mathbf{1}^T)x. $$
>
> This is equivalent to subtracting the DC component (mean) from the original signal:
> $$\mathcal{HC}[x] = x - \mathcal{DC}[x] = x - \frac{1}{|\mathcal{V}\_c|}\mathbf{1}\mathbf{1}^Tx . $$
> This exactly corresponds to our HPF definition:
> $$HPF(h_{v,c}^{(l+1)}) = h_\{v,c}^{(l+1)} - LPF(h_{v,c}^{(l+1)}) = h_{v,c}^{(l+1)} - \frac{1}{|\mathcal{V}\_c|}\sum_{v\in\mathcal{V}\_c} h_{v,c}^{(l+1)}. $$
>
> When applying this to our fractal node features:
> $$f_c^{(l+1)} = \frac{1}{|\mathcal{V}\_c|}\sum_{v\in\mathcal{V}\_c} h_{v,c}^{(l+1)} + \omega_c^{(l)}(h_{v,c}^{(l+1)} - \frac{1}{|\mathcal{V}\_c|}\sum_{v\in\mathcal{V}\_c} h_{v,c}^{(l+1)}),$$
> this decomposition captures high-frequency components for below reasons.
>
> The term $(h_{v,c}^{(l+1)} - \frac{1}{|\mathcal{V}\_c|}\sum_{v\in\mathcal{V}\_c} h_{v,c}^{(l+1)})$ represents deviations from the mean, which by definition are the non-DC components in signal processing.
> These deviations capture local variations because $h_{v,c}^{(l+1)}$ are obtained through message passing, already encoding local neighborhood information, and the subtraction of mean removes global trends, leaving local changes.
> The difference between fractal node features and pooling features captures local variations because the message passing features $h_{v,c}^{(l+1)}$ already contain neighborhood information, subtracting the subgraph mean preserves relative differences between connected nodes, and the learnable parameter $\omega_c^{(l)}$ adaptively balances subgraph-level-global and local information.
>
> In discrete signal processing, high-frequency components are exactly those variations that deviate from the mean. Our equation preserves this principle through explicit mean subtraction while maintaining computational efficiency, capturing local characteristics through message-passing features, and allowing adaptive frequency balancing through $\omega_c^{(l)}$.
>
> Therefore, our implementation and theoretical formulation are consistent with signal processing principles.
>
> ---
>
> We appreciate your careful code review. However, there seems to be a misunderstanding about our implementation. While you noted that our code appears to compute HPF as $HPF(h_{v,c}^{(l+1)}) = h_c - LPF(h_{v,c}^{(l+1)})$ and suggested this might lead to $f_c^{(l+1)}=(1-\omega)\times\frac{1}{|\mathcal{V}\_c|}\sum_{v\in\mathcal{V}\_c} h_{v,c}^{(l+1)} + \omega \times h_c^{(l+1)}$, this is not an accurate interpretation of our implementation. Our code actually implements the exact formulation presented in the paper: $f_c^{(l+1)} = \frac{1}{|\mathcal{V}\_c|}\sum_{v\in\mathcal{V}\_c} h_{v,c}^{(l+1)} + \omega_c^{(l)}(h_{v,c}^{(l+1)} - \frac{1}{|\mathcal{V}\_c|}\sum_{v\in\mathcal{V}\_c} h_{v,c}^{(l+1)})$.
>
> We use this implementation because it directly aligns with classical signal processing principles where the high-frequency components are defined as deviations from the mean. In our code, we first compute the LPF (mean) component and then explicitly subtract it from the original signal to obtain the HPF component, following the standard signal processing practice of $\mathcal{HC}[x] = x - \mathcal{DC}[x]$. This implementation ensures we properly capture both subgraph-level-global structure (through the mean) and local variations (through deviations from the mean) while maintaining computational efficiency.

---

> ### Author Response · Authors · 2024-11-26
>
> ***W3. My primary concern remains unaddressed. The frequency analysis is not convincing to me unless you can address W2.***
>
> The frequency response analysis in `Figure 2` follows the established method in signal processing literature. Specifically, we compute the frequency response by treating our operators as linear filters and analyzing their Fourier-domain responses. For a given operator $A$ (whether it's our fractal node, GCN, self-attention, or mean pooling), we compute its spectral response as $\Lambda = \mathcal{F}A\mathcal{F}^{-1}$, where each row $\Lambda_i$ represents the response at the $i$-th frequency band. The normalized magnitude $\|\Lambda_i\|_2$ then gives us the spectral response intensity shown in Figure 2.
>
> In our `Figure 2`, our fractal nodes show balanced responses across frequencies ($\|\Lambda_i\|_2$ maintains significant values across the spectrum), validating our theoretical decomposition. Wheareas, mean pooling's response concentrates at low frequencies, confirming it acts as a pure low-pass filter. The higher high-frequency response of fractal nodes demonstrates our HPF's efficacy.
>
> These provide empirical validation of our theoretical background in discrete signal processing.
>
> ---
>
> ***W4. While my concern is partially addressed, the revised statement has given rise to a new concern. Specifically: 1) What specific information is being lost in the "pre-defined coarsened network"? 2) if it refer to high-frequency components of the subgraph, I disagree with your point particularly given that previous methods utilize transformers as the aggregator."***
>
> We apologize for rasing new concerns due to the lack of detailed explanation.
>
> To alleviate confusion, we have revised the paragraph in `Section 4.3` to $\color{blue}{blue}$ to compare how Coarformer and ANS-GT differ from our method.
> Regarding 1) of reviewer's comments, we intended to write this comparison in the sense that MLP-Mixer does not need to reconstruct these edges because "pre-defined coarsening graph" needs to reconstruct edges, but this seems to have caused confusion. For 2), we have revised this paragraph to remove the statement to our method for low/high frequency balance, as the reviewer's concerns raised in `Q2` should be alleviated.
>
>
> ---
>
> ***W5. My concern is partially addressed.***
>
> Regarding the reviewer's main concern, the response to W2 and W3, can you please elaborate on what the reviewer feels is still not addressed in our previous W5? We would like to do our best to address the reviewer's concerns.

---

> ### Author Response · Authors · 2024-11-26
>
> ***Q1. After reviewing and running your code, a critical issue concerning the expressive power of your method has surfaced. While I have no objections to incorporating additional features like RWPE to boost performance in graph classification tasks, it is imperative to note that the use of RWPE may mask the inherent expressiveness of your method. RWPE is already acknowledged for its expressiveness in addressing the CSL and SR25 problems. It's worth mentioning that your method fails to function effectively when RWPE is omitted. Notably, upon removing RWPE from your code, the best performance on CSL achieved was 0.3000. In my view, this significantly impacts the contribution of this work.***
>
> Thank you for your careful examination and conducting additional experiments, including running our code without positional encodings. We appreciate your thorough feedback which has helped us present a clearer and more accurate evaluation of our method's expressive power.
>
> We have addressed this concern by conducting new experiments that compare our method with baseline MPNNs without any positional encodings, ensuring a fair evaluation of its inherent expressiveness. The updated results in `Table 17` show improvements on three synthetic datasets designed to test model expressiveness.
>
> | MPNN | CSL | SR25 | EXP |
> | --- | --- | --- | --- |
> | GCN             | 10.00 | 6.67  | 52.17 |
> | GCN+FN$_M$      | 39.67 | 100.0 | 86.40 |
> | GINE            | 10.00 | 6.67  | 51.35 |
> | GINE+FN$_M$     | 47.33 | 100.0 | 95.58 |
> | GatedGCN        | 10.00 | 6.67  | 51.25 |
> | GatedGCN+FN$_M$ | 49.67 | 100.0 | 96.50 |
>
> Our new results show that even without positional encodings, fractal nodes improve the expressive power of base MPNNs. While base MPNNs achieve only 6.67-10% accuracy on CSL and SR25, our method reaches up to 49.67% on CSL and 100% on SR25. Similarly on EXP, our methods boost performance from around 51% to over 95%. These consistent improvements on 3 datasets show that our FN$_M$ method provides inherent structural and representational benefits independent of positional encodings.
>
> We also added an ablation study in `Appendix M` to clearly discuss the role of positional encodings on these three datasets, as mentioned by the reviewer.

---

> ### Author Response · Authors · 2024-11-27
>
> Dear Reviewer `yJ2G`,
>
> Again, we appreciate your concern regarding the graph signal processing (GSP) perspective. We want to clarify the intended scope and design choices of our method.
>
> Our method does not consider the internal connectivity structure of subgraphs and directly connects all nodes in a subgraph to its corresponding fractal node. This method choice places our method in the discrete signal processing (DSP) domain rather than GSP.
>
> This distinction is important because GSP fundamentally relies on understanding how signals propagate through the topology of a graph. In contrast, our method does not consider such signal propagation via internal connections of the sub-graph. Instead, we use mean pooling as a feature aggregation mechanism, which is naturally justified in the discrete Fourier transform, which corresponds to extracting the DC component of the signal.
>
> While one could potentially interpret our method through GSP, this would require additional assumptions that weren't part of our design considerations. Our direct connections to fractal nodes and feature-level aggregation make the DSP interpretation more appropriate and aligned with our method.
>
> We acknowledge that some of our terminology, such as the sentence  "capture the global structure of the subgraph," may have implied a GSP interpretation. We revised such expressions throughout the paper to reflect our method's scope better.
>
> We hope this clarification helps alleviate misconceptions about our method misapplying GSP or graph spectral theory principles. This clarification better aligns with Theorem 3.1 and more accurately represents the intended scope of our approach, which is fundamentally based on DSP principles.
>
> Best regards,
>
> Authors

---

> ### Author Response · Authors · 2024-11-30
>
> Dear Reviewer `yJ2G`,
>
> **As the discussion period has been extended until December 2nd**, I am writing to follow up on our previous response regarding the discrete signal processing (DSP) perspective of our method versus graph signal processing (GSP). We have carefully explained our design choice terms to avoid confusion and to more accuratly reflect the scope and theoretical perspective of our method.
>
> If you have had the opportunity to review our response, ***we would greatly appreciate your consideration in potentially revising your initial rating***, as we believe we have addressed your concerns about the theoretical framework and positioning of our work.
>
> We remain open to any additional feedback or questions you may have before the December 2nd deadline. Your careful attention to theoretical precision has helped us improve the clarity and accuracy of our presentation.
>
> Thank you for your continued engagement with our discussion.
>
> Best regards,
>
> The Authors

---

> > ### Comment · Reviewer_yJ2G · 2024-11-30
> >
> > I appreciate the effort the authors put into addressing our concerns during the rebuttal period, and most of the initial issues have been resolved. However, the original submission contained significant shortcomings, including issues with writing quality, reproducibility, and the discussion of complexity. While the rebuttal serves to clarify the existing content, it is not intended to supplement or introduce additional material missing from the initial submission. I have decided to slightly increase the score, primarily in recognition of the authors' efforts during the rebuttal period.

---

### Official Review · Reviewer_PuK2 · 2024-11-04

**Soundness:** 3
**Presentation:** 3
**Contribution:** 3
**Rating:** 6
**Confidence:** 3

**Summary:**

This paper presents an approach to enhance Graph Neural Networks (GNNs) by introducing the concept of so called fractal nodes. Most common GNN architectures are based on the Message Passing Framework (Message Passing Neural Networks (MPNNs)). Such architectures face challenges in balancing the incorporation of global information with local information propagation mechanisms. In particular, MPNNs struggle with problems like over-smoothing and over-squashing, which hinder their performance in capturing long-range dependencies. Here, Graph Transformers provide a mechanism to incorporate long-range interactions. However, they can be computationally expensive and often overlook the local structures that MPNNs excel at capturing. To overcome these problems in MPNNs, the authors find inspiration  in the fractal nature of many real-world networks, which exhibit self-similarity across different scales. The authors propose that this property can be leveraged to enhance information flow in GNNs: In order to do so, so called "Fractal Nodes" are introduced into given graphs: The proposed fractal nodes coexist with original nodes and each fractal node summarizes information from its corresponding subgraph while maintaining direct connections to the original nodes. This design allows for an aggregation of both local and global information.
Experiments demonstrate that GNNs enhanced with fractal nodes achieve performance comparable to or better than state-of-the-art graph Transformers on various graph-level-task datasets. It is also shown that this approach shows certain benefits when tackling oversquashing.

**Strengths:**

The paper is well structured. With its focus on the incorporation of long-range information distillation into message passing networks, it adresses a timely and important problem. The idea of aggregating local subgraph information into aggregate nodes is well founded. It is great to see that the proposed architecture is able to match or superseed transformer performance on graph level datasets. Beyond this, it is good to see that the authors did not only investigate their method solely in this setting, but in total investigate four different aspects of their proposed model in the experimental section. In particular the performance gain that the proposed fractal nodes bring in the synthetic expressivity setting of Section 5.2 is intriguing. The conducted ablation studies round off the paper well.

**Weaknesses:**

I would argue that a main weakness of the experimental section is the lack of (real world) node level tasks. I am aware that The TreeNeighboursMatch task is a node level task, and I think that it is great that this is included. However, it would be good to see how significant the inclusion of the proposed fractal nodes is for standard node classification benchmarks. Are long-range interactions a problem here and do the proposed fractal nodes alleviate the problem if it exists in this setting?

I am unsure how strong the connection to renormalization techniques is. From a heuristic perspective, I understand that a coarse graining nature is present in both the renormalization group in Physics and when summarizing subgraphs into single nodes. But are there any deeper connections between these two settings? If not, it might be good to make it clearer that this is only used as a heuristic comparison in the present paper.




It seems to be a missed opportunity that no (even preliminary) theoretical analysis of expressivity and mitigation of the oversquashing phenomenon was conducted. Could the authors add some details here?

**Questions:**

In principle, a different sub-graph partitioning could be used to determine how to select subgraphs and assign additional nodes.  Why is it sensible to use self-similarity as a criterion to select these subgraphs?

I have questions regarding the equation (2-4) encapsulating message passing with fractal nodes: As far as I can tell, from eq. (2), there is only messge passing _within_ each subgraph $G_c$, as there is a subscript $c$ present for all

$h^{(\ell)}_{u,c}$

aggregated in the message function $\Psi^{(\ell)}$. Is this just a problem in the notation (I assume that the statement $u \in \mathcal{N}_\nu$

 is supposed to mean that infact not only the intersection of $G_c$ and the neighbourhood $\mathcal{N}_\nu$ is relevant).
Furthermore: As far as I can tell, there is no message passing between the fractal nodes. This information mixing only (potentially) happens in the last layer via the MLP mixer (c.f. eq. (8)). Why not also consider  message passing on the coarse grained graph made up of all the fractal nodes?

How self-similar are the two graph structures (original graph and graph with fractal nodes)? It seems the METIS algorithm does not maximize self-similarity as its objective, but rather maximizes in-cluster connections while minimizing inter-cluster connections. This makes me wonder how adept the name fractal nodes truly is.

In Figure 5, why is there essentially no drop in accuracy until $r = 7$ with a stark drop (to zero) when $r = 8$? Could the authors comment on the origin of this  abrupt change?

Could you explain in more detail how the norm distribution experiment (at the beginning of Section 4.1) allows to draw conclusion about local and global information.

---

> ### Author Response · Authors · 2024-11-22
>
> We sincerely thank Reviewer `PuK2` for the positive review and attentive feedback, highlighting our strengths in
>
> 1. Clear presentation tackling long-range interaction in graphs
> 2. Innovative fractal node approach with theoretical foundations
> 3. Comprehensive validation on benchmark datasets
> 4. Clear demonstration of improved expressivity through synthetic experiments
>
> Below, we would like to respond point by point in the following:
>
> ---
>
> ***(W1) I would argue that a main weakness of the experimental section is the lack of (real world) node level tasks. I am aware that The TreeNeighboursMatch task is a node level task, and I think that it is great that this is included. However, it would be good to see how significant the inclusion of the proposed fractal nodes is for standard node classification benchmarks. Are long-range interactions a problem here and do the proposed fractal nodes alleviate the problem if it exists in this setting?***
>
> We thank this valuable feedback about evaluating our method on real-world node classification tasks. While TreeNeighboursMatch demonstrates our method's effectiveness on synthetic node-level tasks, we have conducted additional experiments on large-scale node classification datasets.
>
> We evaluate our method on 2 challenging large-scale graph datasets requiring long-range interactions: ogbn-arxiv (169,343 nodes, 1,166,243 edges) and ogbn-products (2,449,029 nodes, 61,859,140 edges). The results show that our fractal nodes improve performance on node classification.
>
> On ogbn-arxiv, GCN+FN achieves 73.03% accuracy and outperforms the base GCN and state-of-the-art Transformer-based models like Exphormer and SGFormer. Similarly, GraphSAGE+FN shows consistent improvements over the base model.
>
> On ogbn-products, GraphSAGE+FN$_M$ achieves state-of-the-art performance of 83.11% accuracy, outperforming the base GraphSAGE and most Transformer-based approaches. These results are particularly notable given that several Transformer-based models, such as GraphGPS and Exphormer, run out of memory on this large-scale dataset.
>
> These results indicate that fractal nodes improve MPNNs and capture long-range interactions in large-scale node classification tasks.
>
>
> | Model | OGBN-arxiv | OGBN-product |
> | --- |--- | --- |
> | #nodes                 | 169,343 | 2,449,029 |
> | #edges                 | 1,166,243 | 61,859,140 |
> | Model                  | Accuracy | Accuracy |
> | LINKX                  | 66.18 ± 0.33 | 71.59 ± 0.71 |
> | GraphGPS               | 70.97 ± 0.41 | OOM                          |
> | NAGphormer             | 70.13 ± 0.55 | 73.55 ± 0.21                 |
> | Exphormer              | 72.44 ± 0.28 | OOM                          |
> | NodeFormer             | 69.86 ± 0.25 | 72.93 ± 0.13                 |
> | DiffFormer             | 72.41 ± 0.40 | 74.16 ± 0.31                 |
> | PolyNormer             | 71.82 ± 0.23 | 82.97 ± 0.28  |
> | SGFormer               | 72.63 ± 0.13 | 74.16 ± 0.31 |
> | GCN                    | 71.74 ± 0.29 | 75.64 ± 0.21 |
> | GCN + **FN**           | $\color{red}{73.03 ± 0.37}$  | 81.29 ± 0.21 |
> | GCN + **FN$_M$**       | $\color{blue}{72.93 ± 0.35}$ | 81.33 ± 0.33 |
> | GraphSAGE              | 71.49 ± 0.27 | 78.29 ± 0.16 |
> | GraphSAGE + **FN**     | 72.70 ± 0.11 | $\color{blue}{83.07 ± 0.35}$ |
> | GraphSAGE + **FN$_M$**     | 72.54 ± 0.30 | $\color{red}{83.11 ± 0.07}$ |
>
> ---
>
> ***(W2) I am unsure how strong the connection to renormalization techniques is. From a heuristic perspective, I understand that a coarse graining nature is present in both the renormalization group in Physics and when summarizing subgraphs into single nodes. But are there any deeper connections between these two settings? If not, it might be good to make it clearer that this is only used as a heuristic comparison in the present paper.***
>
> Thank you for this insightful comment. While our fractal nodes method shares the concept of scale transformation with renormalization—where complex systems are studied at different scales—this is mainly a motivating analogy rather than a deep theoretical connection. The key inspiration we draw is the idea of summarizing local structures into representative units. However, our approach differs in that we maintain these units alongside the original nodes rather than replace them.
>
> We revised the caption of `Fig.1` to clarify this distinction and describe the connection to renormalization in `Section H`. We believe that this helps avoid potential misunderstandings of the useful intuition that renormalization techniques provided in developing our method.

---

> ### Author Response · Authors · 2024-11-22
>
> ***(W3) It seems to be a missed opportunity that no (even preliminary) theoretical analysis of expressivity and mitigation of the oversquashing phenomenon was conducted. Could the authors add some details here?***
>
> We appreciate your suggestion regarding the theoretical analysis. We have added a theoretical analysis in `Section 4.1` and `Appendix L` of the updated paper that explains why fractal nodes help mitigate over-squashing. So please refer to the updated paper, and here we leave an intuitive explanation for your understanding.
>
> The key insight comes from effective resistance theory[1]. Intuitively, effective resistance between two nodes decreases when there are shorter and more disjoint paths connecting them. For example, if a path of length 7 originally connected nodes $u$ and $v$, adding a new node that connects to u in 2 hops and directly to $v$ creates a new shorter path of length 3, thereby reducing the effective resistance between $u$ and $v$.
>
> We formally prove that the effective resistance in a graph with fractal nodes ($R_f$) is always less than or equal to the original effective resistance ($R$), i.e., $R_f(u,v) \leq R(u,v)$. This theoretical guarantee directly translates to improved signal propagation bounds, as we show that the signal difference between nodes decays exponentially with the inverse of effective resistance.
>
> This theoretical analysis provides justification for why fractal nodes are effective at mitigating over-squashing while maintaining the computational efficiency of MPNNs.
>
>
> > [1] Black, Mitchell, et al. "Understanding oversquashing in gnns through the lens of effective resistance." International Conference on Machine Learning. PMLR, 2023.
>
> ---
>
> ***(Q1) In principle, a different sub-graph partitioning could be used to determine how to select subgraphs and assign additional nodes. Why is it sensible to use self-similarity as a criterion to select these subgraphs?***
>
> The core of our approach lies not in the partitioning method itself but rather in how fractal nodes enforce feature self-similarity through their representation mechanism. While we use METIS as an implementation choice for initial graph partitioning, the feature self-similarity aspect comes from how fractal nodes represent and interact with their subgraphs - creating representations that capture local and global properties through LPF and HPF components. We can indeed view our method from both structural and feature self-similarity perspectives, and we have revised our `Introduction section` to reduce any confusion between these aspects.
>
> While our analysis in `Section 4.1` shows that partitioned subgraphs can have structural similarity with the whole graph, extending our method to use self-similarity as a partitioning criterion itself is an interesting direction but outside our current scope. However, this suggestion opens valuable future research directions, such as developing partitioning methods based on structural or feature-based self-similarity.
>
> We also have conducted an additional analysis using different subgraph partitioning methods in `Appendix I`. Our analysis shows that while METIS partitioning works optimally for graph-level benchmark datasets,  Louvain partitioning also performs well on large-scale datasets like ogbn. In other words, while different datasets may have their optimal partitioning methods, the key innovation of our approach is injecting feature self-similarity into subgraph nodes through fractal nodes.
>
> The empirical results demonstrate that even with METIS partitioning, the fractal node mechanism effectively captures and leverages self-similar properties in graph data, delivering strong performance on various tasks.

---

> ### Author Response · Authors · 2024-11-22
>
> ***(Q2) I have questions regarding the equation (2-4) encapsulating message passing with fractal nodes: As far as I can tell, from eq. (2), there is only messge passing within each subgraph $G_c$, as there is a subscript $c$ present for all $h_{v,c}^{(\ell)}$ aggregated in the message function $\psi^{(\ell)}$. Is this just a problem in the notation (I assume that the statement $u \in \mathcal{N}\_v$ is supposed to mean that infact not only the intersection of $\mathcal{G}\_c$ and the neighbourhood $\mathcal{N}_{v}$ relevant). Furthermore: As far as I can tell, there is no message passing between the fractal nodes. This information mixing only (potentially) happens in the last layer via the MLP mixer (c.f. eq. (8)). Why not also consider message passing on the coarse grained graph made up of all the fractal nodes?***
>
> Thank you for this insightful question about message passing mechanics in our model. You are correct that fractal node interactions occur primarily through the MLP-Mixer in the final layer. This design choice was deliberate, as it offers greater flexibility than defining explicit edge connectivity in a coarsened graph structure.
>
> We have experimentally evaluated the alternative approach of implementing message passing between fractal nodes across all layers. While this approach shows improvements in some cases compared to the base FN model (e.g., MolHiv accuracy improves from 0.7882 to 0.8025 for GINE), it consistently underperforms compared to our FN$_M$ design that uses the MLP-Mixer in the final layer. For instance, with GINE on PEPTIDES-FUNC, all-layer message passing achieves 0.6660 accuracy, while FN$_M$ reaches 0.7018.
>
> The empirical results across multiple datasets and model architectures (GCN, GINE, GatedGCN) demonstrate that our proposed design with final-layer mixing through MLP-Mixer consistently delivers superior performance. We added these comparative results to `Appendix E.6` to comprehensively analyze different architectural choices.
>
> We appreciate the reviewer's suggestion of exploring fractal node message passing, as it has helped us better validate our design decisions through thorough experimental comparison.
>
> | GCN | Peptides-func | Peptides-struct | MolHiv | MolTox |
> |---|---|---|---|---|
> | +FN | 0.6802 ± 0.0043 | 0.2530 ± 0.0004 | 0.7564 ± 0.0059 | 0.7670 ± 0.0073 |
> | +FN (all layers) | 0.6582 ± 0.0032 | 0.2531 ± 0.0008 | 0.7783 ± 0.0164    | 0.7600 ± 0.0037 |
> | +FN$_M$ (MLP-Mixer) | 0.6787 ± 0.0048    | 0.2464 ± 0.0014  | 0.7866 ± 0.0034  | 0.7882 ± 0.0041  |
>
> | GINE | Peptides-func | Peptides-struct | MolHiv | MolTox |
> |---|---|---|---|---|
> | +FN | 0.6815 ± 0.0059  | 0.2515 ± 0.0020  | 0.7882 ± 0.0050  | 0.7751 ± 0.0029 |
> | +FN (all layers) | 0.6660 ± 0.0067  | 0.2530 ± 0.0011  | 0.8025 ± 0.0100 | 0.7680 ± 0.0056  |
> | +FN$_M$ (MLP-Mixer) | 0.7018 ± 0.0074  | 0.2446 ± 0.0018  | 0.8127 ± 0.0076  | 0.7926 ± 0.0021 |
>
> | GatedGCN | Peptides-func | Peptides-struct | MolHiv | MolTox |
> |---|---|---|---|---|
> | +FN | 0.6778 ± 0.0056 | 0.2536 ± 0.0019 | 0.7967 ± 0.0098 | 0.7759 ± 0.0054 |
> | +FN (all layers) | 0.6658 ± 0.0048 | 0.2531 ± 0.0009 | 0.7898 ± 0.0065 | 0.7642 ± 0.0050 |
> | +FN$_M$ (MLP-Mixer) | 0.6950 ± 0.0047| 0.2453 ± 0.0014|0.8097 ± 0.0047|0.7922 ± 0.0054|

---

> ### Author Response · Authors · 2024-11-22
>
> ***(Q3) How self-similar are the two graph structures (original graph and graph with fractal nodes)? It seems the METIS algorithm does not maximize self-similarity as its objective, but rather maximizes in-cluster connections while minimizing inter-cluster connections. This makes me wonder how adept the name fractal nodes truly is.***
>
> We appreciate this insightful question about self-similarity in our method. The augmented graph is structurally different from the original graph due to the additional edges connecting fractal nodes to their subgraph nodes. We would like to clarify that self-similarity in our approach can be viewed from two perspectives: structural and feature-level self-similarity.
>
> From a structural perspective, While METIS does not explicitly optimize for structural self-similarity, our analysis in Section 4.1 and `Appendix C` demonstrates that the resulting subgraphs maintain similar distributional properties to the original graph through betweenness centrality analysis.
>
> From a feature-level perspective:
> 1. Each fractal node enforces feature self-similarity through:
>    - Adaptively combining low-frequency (LPF) and high-frequency (HPF) components of subgraph information
>    - Maintaining a shared latent space between fractal and original nodes
>    - Enabling smaller units (nodes) to reflect properties of larger units (subgraphs)
> 2. This feature-level self-similarity is the primary focus of our method, as it allows effective information propagation and representation learning.
>
> While METIS acts as an initialization tool for partitioning, the "fractal" terminology in our method primarily refers to how node representations capture and maintain feature self-similar properties during training. However, structural self-similarity naturally emerges as well. We have updated our paper, specifically the `Introduction` section, to clarify these distinct aspects and avoid confusion.
>
> Thank you for helping us improve the accuracy of our terminology.
>
> ---
>
> ***(Q4) In Figure 5, why is there essentially no drop in accuracy until r=7 with a stark drop (to zero) when r=8? Could the authors comment on the origin of this abrupt change?***
>
> As shown in Table below, the number of output classes grows exponentially with r, following $2^{(r-1)}$. When r=7, the model needs to classify into 128 classes, but at r=8 this jumps to 256 classes. This exponential scaling makes r=8 qualitatively different and substantially more challenging than r=7. The binary tree structure at r=8 has significantly more nodes and paths to consider than r=7. The fractal nodes help maintain good performance up through r=7 by effectively propagating information in the tree structure. Despite our architectural advantage, the fundamental difficulty of suddenly classifying 256 different classes at r=8 becomes too difficult compared to the capacity of the base MPNN models that fractal nodes are applied to: GCN, GINE, and GatedGCN models.
>
> | Dataset | #Graphs | #Class |
> |---|---|---|
> | TreeNeighbourMatch (r=2) | 96 | 4 |
> | TreeNeighbourMatch (r=3) | 32,000 | 8 |
> | TreeNeighbourMatch (r=4) | 64,000 | 16 |
> | TreeNeighbourMatch (r=5) | 128,000 | 32 |
> | TreeNeighbourMatch (r=6) | 256,000 | 64 |
> | TreeNeighbourMatch (r=7) | 512,000 | 128 |
> | TreeNeighbourMatch (r=8) | 640,000 | 256 |
>
> ---
>
> ***(Q5) Could you explain in more detail how the norm distribution experiment (at the beginning of Section 4.1) allows to draw conclusion about local and global information?***
>
> We have strengthened our analysis with a frequency response analysis. Rather than relying on L2 norm distributions, we now analyze how different methods process and preserve information across frequency scales.
>
> As shown in new `Fig.2`, our frequency response analysis reveals distinct patterns in how each method handles information: self-attention shows strong responses across all frequencies but potentially over-emphasizes global information. In contrast, mean pooling shows minimal response in low frequencies, indicating significant information loss. In contrast, fractal nodes show a well-balanced frequency response profile, with strong low-frequency components capturing subgraph-level global context and high-frequency components preserving local details. This balanced response maintains both global and local information.
>
> Note that we have also added theoretical analysis in addition to frequency response analysis to enhance the content of `Section 4.1`, and we would greatly appreciate it if you could provide more details on this in `Appendix L`.

---

> > ### Comment · Reviewer_PuK2 · 2024-11-23
> > **Thanks!**
> >
> > Thank you for your response and the work that went into preparing it. I am happy to maintain my current score.

---

> ### Author Response · Authors · 2024-11-27
>
> Dear Reviewer `PuK2`,
>
> Thank you for your thorough review and feedback. We deeply appreciate how your comments have helped us improve our paper. With the extended rebuttal period, we would like to highlight several key improvements made in response to your review:
>
> 1. One of the most significant improvements to our paper, thanks to your review, is the theoretical content regarding the mitigation of over-squashing. We have added important theoretical analysis in `Section 4.1` and `Appendix L`. ***We noticed that we previously misplaced our response to your W3 as a response to Q5, and have now corrected this.*** We would be grateful if you could review this corrected response.
>
> 2. As you suggested, we have demonstrated the importance of our method on node classification benchmarks, particularly showing its effectiveness on large-scale benchmarks such as ogbn-arxiv and ogbn-products where long-range interactions are crucial. `Tables 15` and `Table 16` show that *our method effectively enhances MPNN capacity while maintaining computational efficiency* -- providing a practical alternative to graph Transformers, which *aligns with the architectural strengths you acknowledged in your initial review.*
>
> 3. Your insightful comments helped us justify the design of our FN$_M$ in response to your Q2, which we have detailed in Appendix E.6.
>
> 4. Thanks to your feedback, we have improved our terminology and enhanced the paper's presentation, particularly in clarifying Section 4.1.
>
> Given these substantial additions and clarifications, we would be grateful if you would reconsider increasing the score. Of course, if you have any remaining questions or concerns, we are happy to address them during this extended rebuttal period.
>
> Thank you again for your valuable feedback in helping us improve our work.
>
> Best regards,
>
> The Authors

---

> ### Author Response · Authors · 2024-11-30
>
> Dear Reviewer `Puk2`,
>
> We appreciate your decision to maintain the current score, and thank you for your feedback. **However, we would like to remind you that the discussion period has been extended until December 2nd.**
>
> In our previous response, we outlined several substantial improvements made to the paper based on your valuable feedback, including:
> - Enhanced theoretical analysis of over-squashing mitigation in `Section 4.1` and `Appendix L`
> - New evaluation on node classification benchmarks
> - Detailed justification of our FN$_M$ design in `Appendix E.6`
> - Improved term and presentation clarity
>
> Given these significant improvements and the extended discussion period until December 2nd, ***we would be grateful if you would consider giving us another opportunity to further strengthen the paper and potentially reconsider the rating.*** We are dedicated to addressing any additional concerns you may have.
>
> Thank you for your continued engagement in helping us improve our work.
>
> Best regards,
>
> The Authors

---

### Author Response · Authors · 2024-11-22

Dear reviewers,

We sincerely appreciate your attentive feedback and comments. We have updated our paper, with all changes and additions highlighted in $\color{red}{\text{red}}$ in the updated PDF. These major changes include:

- Theoretical enhancements:
  1. `Section 4.1` : Added comprehensive theoretical analysis in (highlighted in red), including frequency response analysis to better explain how fractal nodes work
  2. `Appendix L`: Added detailed theoretical analysis with proofs of added Theorems in `Section 4.1`

- Additional empirical studies:
  1. `Appendix E.5`: Extended sensitivity analysis on the number of fractal nodes
  2. `Appendix E.6`: Comparative study of all-layer fractal node message passing
  3. `Appendix G`: Thorough analysis of subgraph size ratio distribution
  4. `Appendix I`: Comprehensive evaluation using different partitioning algorithms
  5. `Appendix J`: Extensive scalability analysis on both synthetic and real-world graphs
  6. `Appendix K`: New large-scale node classification experiments in

- Conceptual Clarifications:
  1. `Appendix H`: Added explanation connections to renormalization techniques
  2. `Appendix L`: Enhanced theoretical framework linking our approach to existing methods

**We encourage you to review these updates, which may enhance your understanding of our responses to your questions and concerns.**
So please refer to the $\color{red}{\text{red}}$ highlighted sections in our updated paper to track our updates in response to your feedback.

Thank you for helping us improve the quality and clarity of our work.

Best regards,

The Authors

---

### Author Response · Authors · 2024-12-02
**Gentle Reminder**

Dear All Reviewers and Chairs,

The discussion is due in two days and we have significantly improved the paper. In particular, we would appreciate it if Reviewer HGoY and Reviewer Vhu4 could review our latest rebuttal and provide their feedback. We are eager to assist them in better understanding our work.

Best,

Authors

---

### Meta-Review · Area_Chair_HHmq · 2024-12-21

**Metareview:**

**(a) Scientific Claims and Findings:**
The paper titled "Fractal-Inspired Message Passing Neural Networks with Fractal Nodes" introduces a novel concept termed 'fractal nodes' to enhance Graph Neural Networks (GNNs). Drawing inspiration from the fractal nature of real-world networks, the authors propose a message-passing scheme that captures both local and global structural information. By creating fractal nodes that coexist with original nodes, the method enforces feature self-similarity, allowing fractal nodes to adaptively summarize subgraph information. This approach aims to alleviate the over-squashing problem by providing direct shortcuts for information to traverse long distances within the graph. Empirical evaluations demonstrate that the proposed method achieves performance comparable to or better than graph Transformers while maintaining the computational efficiency of traditional Message Passing Neural Networks (MPNNs).

**(b) Strengths:**
* Innovative Approach: The introduction of fractal nodes balances local and global information processing in GNNs, addressing limitations in both MPNNs and graph Transformers.
* Addressing Over-Squashing: By providing direct shortcuts for long-distance information passage, the method effectively tackles the over-squashing problem, enhancing the model's ability to capture long-range dependencies.
* Empirical Validation: The proposed architecture demonstrates competitive or superior performance compared to Graph Transformers, coupled with the computational efficiency characteristic of MPNNs.

**(c) Weaknesses:**
* Theoretical Foundation: While the concept of fractal nodes is intriguing, the paper initially lacked a rigorous theoretical analysis detailing how and why this approach effectively balances local and global information processing. Section 4.1 and Appendix L in the updated manuscript now provide insights into why the proposed method mitigates over-squashing. The analysis argues via the effective resistance, which is a proxy measure for over-squashing. Yet, it does not directly relate to the generalization performance of simple GNNs.
* Scalability Analysis: The original submission did not provide an in-depth examination of the model's scalability, particularly concerning large-scale graphs common in real-world applications. An analysis for ogbn-arxiv was added to the appendix, which suggests that training time and memory requirements of the proposed method are similar to GCNs. However, the time to identify fractal nodes seems to have been omitted from the analysis.
* Reproducibility: Reviewer yJ2G raised concerns regarding the reproducibility of the results, which were addressed during the rebuttal.
* Self-similarity and ablation: Reviewer yJ2G was not fully satisfied by the offered connection between self-similarity and generalization. Is the identification of fractal nodes really relevant for performance improvements? Or could also the introduction of other additional nodes (e.g. with some hierarchical meaning like community structure or just random) and the separate message-passing lead to similar performance gains? An ablation could provide insights into the relevance of self-similarity in this context.
* Comparative Evaluation: A more comprehensive comparison with existing state-of-the-art models, beyond graph Transformers, would strengthen the paper by contextualizing its contributions within the broader landscape of GNN research. Reviewer Vhu4 found performance improvements of the proposed method in the experiments of the main paper to be not particularly significant.

**(d) Reasons for Rejection:**
After thorough evaluation, the decision to reject the paper is based on the following considerations:
1. Reviewers were not sufficiently convinced by the rebuttal, which led to significant changes that warrant a thorough review of the updated manuscript.
2. Scalability Analysis: The scaleability analysis should be extended to cover the cost of identifying fractal nodes.
3. Relevance of self-similarity: An open point is whether the self-similarity is relevant for performance gains or just the addition of virtual nodes that are related to a higher level organisation of the graph (which is not a new idea on its own) and learning related separate message passing functions.

**Additional Comments On Reviewer Discussion:**

Most reviewers engaged in long discussions with the authors. Many issues have been addressed during the rebuttal, including a lack of theoretical analysis (Reviewer HGoY), problems with reproducibility (Reviewer yJ2G) and understanding of the impact of self-similarity on generalization (Reviewer yJ2G). Reviewer Vhu4 found performance improvements of the proposed method in the experiments of the main paper to be not particularly significant.

I believe that the many additional contributions (including an added theoretical analysis) warrant another round of thorough reviews.

---

### Decision · Program_Chairs · 2025-01-22

Reject